# SHARPNESS-AWARE BLACK-BOX OPTIMIZATION

**Feiyang Ye**[1,2,*]**, Yueming Lyu**[3,4,*]**, Xuehao Wang**[1]**,**
**Masashi Sugiyama**[5,6]**, Yu Zhang**[1,†]**, Ivor W. Tsang**[2,3,4,7]

[1]Department of Computer Science and Engineering, Southern University of Science and Technology
[2]Australian Artificial Intelligence Institute, University of Technology Sydney
[3]Centre for Frontier AI Research, Agency for Science, Technology and Research, Singapore
[4]Institute of High Performance Computing, Agency for Science, Technology and Research, Singapore
[5]RIKEN Center for Advanced Intelligence Project
[6]Graduate School of Frontier Sciences, The University of Tokyo
[7]College of Computing and Data Science, Nanyang Technological University
{feiyang.ye.uts,xuehaowangfi,yu.zhang.ust}@gmail.com
sugi@k.u-tokyo.ac.jp  {Lyu_Yueming,Ivor_Tsang}@cfar.a-star.edu.sg

## ABSTRACT

Black-box optimization algorithms have been widely used in various machine learning problems, including reinforcement learning and prompt fine-tuning. However, directly optimizing the training loss value, as commonly done in existing black-box optimization methods, could lead to suboptimal model quality and generalization performance. To address those problems in black-box optimization, we propose a novel Sharpness-Aware Black-box Optimization (SABO) algorithm, which applies a sharpness-aware minimization strategy to improve the model generalization. Specifically, the proposed SABO method first reparameterizes the objective function by its expectation over a Gaussian distribution. Then it iteratively updates the parameterized distribution by approximated stochastic gradients of the maximum objective value within a small neighborhood around the current solution in the Gaussian distribution space. Theoretically, we prove the convergence rate and generalization bound of the proposed SABO algorithm. Empirically, extensive experiments on the black-box prompt fine-tuning tasks demonstrate the effectiveness of the proposed SABO method in improving model generalization performance.

## 1 INTRODUCTION

Black-box optimization involves optimizing one objective function by using function queries only. In this work, we study the black-box optimization problem (Jones et al., 1998), which is formulated as

$$\min_{\boldsymbol{x}} F(\boldsymbol{x}), \text{ s.t. } \boldsymbol{x} \in \mathcal{X}, \tag{1}$$

where $\mathcal{X} \subseteq \mathbb{R}^d$, and $d$ represents the parameter dimension. The objective function $F : \mathbb{R}^d \to \mathbb{R}$, which satisfies $F(\boldsymbol{x}) \geq -\infty$ (lower bounded), can only be queried to obtain function values and we cannot get the gradient of $F$ w.r.t. $\boldsymbol{x}$. In this work, we focus on the online setting for black-box optimization, where different from the offline setting (Chen et al., 2022; Qi et al., 2022), we do not have a prior dataset containing the variable $\boldsymbol{x}$ and its corresponding objective value.

Black-box optimization has drawn intensive attention in a wide range of applications, such as deep reinforcement learning (Salimans et al., 2017; Conti et al., 2018), black-box adversarial attacks of deep neural networks (Ilyas et al., 2018; Dong et al., 2019), etc. Recently, black-box optimization has shown increasing power on real-world natural language processing tasks, especially with the emergence of large language models (LLMs). Since a common practice is to release LLMs as a service and allow users to access it through their inference APIs. In such a scenario, called Languaged-Model-as-a-Service (LMaaS) (Sun et al., 2022b;a; 2023), users cannot access or tune model parameters but can only tune their prompts without model backpropagation to accomplish language tasks of interest, which directly increase the demand of black-box optimization methods.

---

[*]Equal contribution.
[†]Corresponding author.

Although black-box optimization algorithms have been successfully applied to various learning tasks, most existing works directly optimize the training loss value (Sun et al., 2022b), which may lead to suboptimal model quality and generalization performance. Since the training loss landscape is complex and has many local minima with different generalization abilities (Zhang et al., 2021), the learned model may suffer from the overfitting problem, causing poor generalization performance (Foret et al., 2021). Hence, reducing the performance gap between training and testing is an important research topic in deep learning (Neyshabur et al., 2017). Recently, there have been many works exploring the close relationship between loss geometry and generalization performance, and it has been observed that flat minima often imply better generalization (Dziugaite & Roy, 2017; Chatterji et al., 2019; Petzka et al., 2021). This inspires us to design a black-box optimization algorithm to improve the model generalization by finding the flat minima.

Sharpness-aware minimization (SAM) (Foret et al., 2021) is a state-of-the-art method to seek flat minima in white-box cases by solving a min-max optimization problem. SAM minimizes the maximum objective value within a small neighborhood around current solution. Since SAM considers the geometry of the Euclidean parameter space, it uses the Euclidean ball to define the neighborhood. In SAM, each update consists of two forward-backward computations: one for computing the perturbation and the other for computing the actual update direction. SAM has been proven to perform better than SGD and its variants (Kwon et al., 2021; Zhuang et al., 2022; Zhao et al., 2022; Kim et al., 2022; Jiang et al., 2023; Liu et al., 2022) yield significant performance gains in various fields such as computer vision and natural language processing (Bahri et al., 2022; Foret et al., 2021). However, SAM and its variants rely on the availability of true gradients or stochastic gradients w.r.t. the variable $x$, so they are inapplicable to black-box optimization.

To take advantage of SAM to improve the generalization performance of black-box optimization, we propose a **S**harpness-**A**ware **B**lack-box **O**ptimization (**SABO**) algorithm. Specifically, SABO first reparameterizes the objective function via its expectation over a Gaussian distribution, which can help to optimize the objective by only accessing the function value (Wierstra et al., 2014; Lyu & Tsang, 2021). Then the SABO method seeks to identify the robust minimum region over the space of Gaussian distributions, which is different from SAM that finds the flat minimum over the parameter space. To achieve that, the SABO method iteratively updates the parameterized distribution via a search direction obtained by approximated stochastic gradients for the maximum objective value within a small neighborhood around the current solution in the space of Gaussian distributions. Theoretically, we analyze the convergence rate and provide a generalization error bound for the proposed SABO algorithm. Empirically, we verify the convergence result of the proposed algorithm on the synthetic problems, and extensive experimental results on a black-box prompt fine-tuning problem demonstrate the effectiveness of the proposed SABO method. Our contributions are summarized as follows.

- We propose the SABO algorithm for black-box optimization. To the best of our knowledge, we are the first to design a stochastic gradient approximation algorithm to improve the model generalization in black-box optimization by using the sharpness-aware minimization strategy.

- Theoretically, we prove that the proposed SABO algorithm possesses a convergence rate $\mathcal{O}(\frac{\log T}{T})$ in a full-batch function query setting and $\mathcal{O}(\frac{1}{\sqrt{T}})$ in a mini-batch function query setting, respectively. Moreover, we provide a generalization error analysis for the proposed SABO method.

- Empirically, we verify the convergence result of the SABO algorithm on the synthetic numerical problems. Moreover, extensive experiments on black-box prompt fine-tuning tasks demonstrate the effectiveness of the proposed SABO method in improving the model generalization performance.

**Notation and Symbols.** We denote by $\| \cdot \|_2$ and $\| \cdot \|_\infty$ the $l_2$ norm and the $l_\infty$ norm for vectors, respectively. $\| \cdot \|_{\mathrm{F}}$ denotes the Frobenius norm for matrices. $\mathcal{S}^+$ denotes the set of positive semi-definite matrices. For a square matrix $\mathbf{X}$, $\mathrm{diag}(\mathbf{X})$ represents a vector with diagonal entries in $\mathbf{X}$, and if $\mathbf{x}$ is a vector, $\mathrm{diag}(\mathbf{x})$ represents a diagonal matrix with $\mathbf{x}$ as its diagonal entries. We define $\|X\|_Y := \sqrt{\langle X, YX \rangle}$ for a positive semi-definite matrix $Y \in \mathcal{S}^+$ or a non-negative vector $Y$, where $\langle \cdot, \cdot \rangle$ denotes the inner product under the Frobenius norm for matrices and inner product under the $l_2$ norm for vectors. $\frac{X}{Y}$ denotes the elementwise division operation when $X$ and $Y$ are vectors (for the diagonal matrix), and the elementwise division operation for diagonal elements in $X$ and $Y$ when they are diagonal matrices.

## 2 BACKGROUND

**Stochastic Gradient Approximation**   The stochastic gradient approximation method (Wierstra et al., 2014; Lyu & Tsang, 2021; Ye et al., 2024) is a representative strategy for solving black-box optimization problems, which instead of maintaining a population of searching points, iteratively updates a search distribution by stochastic gradient approximation. The general procedure of stochastic gradient approximation methods is to first generate a batch of sample points by a parameterized search distribution. Then the sample points allow the algorithm to capture the local structure of the fitness function and appropriately estimate the stochastic gradient to update the distribution.

Specifically, the stochastic gradient approximation method reparameterizes $F(\boldsymbol{x})$ as

$$J(\boldsymbol{\theta}) = \mathbb{E}_{p_{\boldsymbol{\theta}}(\boldsymbol{x})}[F(\boldsymbol{x})] = \int F(\boldsymbol{x})p(\boldsymbol{x};\boldsymbol{\theta})\mathrm{d}\boldsymbol{x}, \tag{2}$$

where $\boldsymbol{\theta}$ denotes the parameters of density $p(\boldsymbol{x};\boldsymbol{\theta})$ or $p_{\boldsymbol{\theta}}$ and $F(\boldsymbol{x})$ is also referred to as the fitness function for $\boldsymbol{x}$. Based on this definition, we can obtain the Monte Carlo estimation of the search gradient as

$$\bar{\nabla}_{\boldsymbol{\theta}} J(\boldsymbol{\theta}) = \frac{1}{N} \sum_{j=1}^{N} F(\boldsymbol{x}_j) \nabla_{\boldsymbol{\theta}} \log p_{\boldsymbol{\theta}}(\boldsymbol{x}_j), \tag{3}$$

where $\boldsymbol{x}_j$ denotes the $j$-th sample and $N$ denotes the number of samples. Therefore, the stochastic gradient $\bar{\nabla}_{\boldsymbol{\theta}} J(\boldsymbol{\theta})$ provides a search direction in the space of search distributions.

**Sharpness-Aware Minimization**   SAM (Foret et al., 2021) attempts to improve generalization by finding flat minima. This is achieved by minimizing the worst-case loss within some perturbation radius. Mathematically, it is formulated as the following minimax optimization problem:

$$\min_{\boldsymbol{x} \in \mathcal{X}} \max_{\|\boldsymbol{\epsilon}\|_2 \leq \rho^2} F(\boldsymbol{x} + \boldsymbol{\epsilon}), \tag{4}$$

where $F : \mathbb{R}^d \to \mathbb{R}$ is the objective function, $\boldsymbol{x}$ denotes variables that can represent model parameters, $\rho > 0$ is a positive constant, and $\boldsymbol{\epsilon}$ is the perturbation whose magnitude is bounded by $\rho^2$. By taking the first-order approximation of $F(\boldsymbol{x}+\boldsymbol{\epsilon})$ over $F(\boldsymbol{x})$, a solution of $\boldsymbol{\epsilon}$ for the maximization subproblem can be obtained as

$$\boldsymbol{\epsilon}(\boldsymbol{x}) = \frac{\rho^2 \nabla F(\boldsymbol{x})}{\|\nabla F(\boldsymbol{x})\|_2}. \tag{5}$$

Then problem (4) can be solved by performing the gradient descent method for the minimization subproblem as $\boldsymbol{x}_{t+1} = \boldsymbol{x}_t - \beta_t \nabla_{\boldsymbol{x}} F(\boldsymbol{x}_t + \boldsymbol{\epsilon}(\boldsymbol{x}_t))$, where $\beta_t$ represents the step size in the $t$-th iteration. Note that this gradient implicitly depends on the Hessian of $F(\boldsymbol{x})$ because $\boldsymbol{\epsilon}(\boldsymbol{x})$ is a function of $\boldsymbol{x}$. To accelerate the computation, a common approach in SAM-based methods (Foret et al., 2021; Kim et al., 2022; Jiang et al., 2023) is to apply a first-order gradient approximation, so we obtain the update rule for SAM as $\boldsymbol{x}_{t+1} = \boldsymbol{x}_t - \beta_t \nabla_{\boldsymbol{x}} F(\boldsymbol{x})|_{\boldsymbol{x}=\boldsymbol{x}_t+\boldsymbol{\epsilon}_t}$, where $\boldsymbol{\epsilon}_t = \boldsymbol{\epsilon}(\boldsymbol{x}_t)$ is viewed as a constant w.r.t. $\boldsymbol{x}$.

## 3 METHODOLOGY

In this section, we introduce the proposed SABO algorithm. Firstly, we formulate the sharpness-aware black-box optimization as a min-max optimization problem and solve it in Section 3.1, and in Section 3.2, we derive the update formula of parameters in the search distribution. The detailed derivations in this section are put in Appendix A.

### 3.1 SHARPNESS-AWARE BLACK-BOX OPTIMIZATION

Suppose we are given a training set $\mathcal{D}$ with i.i.d. samples $\{(X_i, y_i)\}$. The main objective is defined as

$$F(\boldsymbol{x};\mathcal{D}) = \frac{1}{|\mathcal{D}|} \sum_{(X_i, y_i) \in \mathcal{D}} l(\boldsymbol{x};(X_i, y_i)), \tag{6}$$

where $\boldsymbol{x}$ denotes model parameters, $|\mathcal{D}|$ denotes the number of data in the dataset $\mathcal{D}$, and $l(\boldsymbol{x}; (X, y))$ denotes the loss function (e.g., the cross-entropy loss for classification). To simplify the notation, we define $F(\boldsymbol{x}) := F(\boldsymbol{x}; \mathcal{D})$.

In black-box optimization, we aim at minimizing the objective function $F(\boldsymbol{x})$, with only function queries. Due to the lack of gradient information, we first apply the stochastic gradient approximation method (Wierstra et al., 2014; Lyu & Tsang, 2021). We denote by $\boldsymbol{\theta}$ the parameters of the search distribution $p_{\boldsymbol{\theta}}$ and define the expected fitness of $F(\boldsymbol{x})$ under the parametric search distribution $p_{\boldsymbol{\theta}}(\boldsymbol{x})$ as $J(\boldsymbol{\theta}) = \mathbb{E}_{p_{\boldsymbol{\theta}}(\boldsymbol{x})}[F(\boldsymbol{x})]$. Then the optimal parameter $\boldsymbol{\theta}$ can be found by minimizing the reparameterized objective $J(\boldsymbol{\theta})$.

Inspired by SAM (Foret et al., 2021), we attempt to improve generalization by finding flat minima of $\boldsymbol{\theta}$. However, for the reparameterized objective $J(\boldsymbol{\theta})$, the geometry of the corresponding distribution space is not Euclidean but a statistical manifold, where the distance between two probability distributions is defined by some statistical distance, e.g., Kullback-Leibler (KL) divergence. Therefore, instead of restricting the perturbation in an Euclidean ball, we restrict the perturbation distribution to be inside a small neighborhood of the unperturbed distribution w.r.t. the KL divergence (Amari, 2016). The proposed optimization problem for black-box optimization is formulated as

$$\min_{\boldsymbol{\theta}} \max_{\boldsymbol{\delta} \in \mathcal{C}(\boldsymbol{\theta})} J(\boldsymbol{\theta} + \boldsymbol{\delta}), \tag{7}$$

where $\mathcal{C}(\boldsymbol{\theta}) = \{\boldsymbol{\delta} \mid \mathrm{KL}(p_{\boldsymbol{\theta}+\boldsymbol{\delta}} \| p_{\boldsymbol{\theta}}) \leq \rho^2\}$, $\rho$ is a positive constant, and $J(\boldsymbol{\theta} + \boldsymbol{\delta}) = \mathbb{E}_{x \sim p_{\boldsymbol{\theta}+\boldsymbol{\delta}}}[F(\boldsymbol{x})]$. Note that $\mathcal{C}(\boldsymbol{\theta})$ defines the neighborhood around a given distribution $p_{\boldsymbol{\theta}}$ in the distribution space, which is different from the neighborhood of SAM that is defined in the parameter space. Here $\rho$ defines the size of the neighborhood.

Generally, problem (7) is applicable to any family of distribution $p_{\boldsymbol{\theta}}$. For computational consideration, the search distribution is assumed to be a Gaussian distribution, i.e., $p_{\boldsymbol{\theta}}(\boldsymbol{x}) = \mathcal{N}(\boldsymbol{x} \mid \boldsymbol{\mu}, \boldsymbol{\Sigma})$ where $\boldsymbol{\mu}$ denotes the mean and $\boldsymbol{\Sigma}$ denotes the covariance matrix, and correspondingly $\boldsymbol{\theta}$ includes $\boldsymbol{\mu}$ and $\boldsymbol{\Sigma}$, i.e., $\boldsymbol{\theta} = \{\boldsymbol{\mu}, \boldsymbol{\Sigma}\}$. In this work, we assume that the covariance matrix $\boldsymbol{\Sigma}$ is a diagonal matrix. For a perturbation $\boldsymbol{\delta} = \{\boldsymbol{\delta}_{\boldsymbol{\mu}}, \boldsymbol{\delta}_{\boldsymbol{\Sigma}}\}$ where $\boldsymbol{\delta}_{\boldsymbol{\Sigma}}$ is a digaonal matrix, the perturbed distribution is a Gaussian distribution $p_{\boldsymbol{\theta}+\boldsymbol{\delta}}(\boldsymbol{x}) = \mathcal{N}(\boldsymbol{x} \mid \boldsymbol{\mu} + \boldsymbol{\delta}_{\boldsymbol{\mu}}, \boldsymbol{\Sigma} + \boldsymbol{\delta}_{\boldsymbol{\Sigma}})$.

We need to solve problem (7) to derive the update formulation for $\boldsymbol{\theta}$. By using the first-order Taylor expansion, the maximization subproblem can be approximated as a quadratically constrained linear programming problem:

$$\max_{\boldsymbol{\delta} \in \mathcal{C}(\boldsymbol{\theta})} J(\boldsymbol{\theta} + \boldsymbol{\delta}) \approx \max_{\boldsymbol{\delta} \in \mathcal{C}(\boldsymbol{\theta})} \langle \nabla_{\boldsymbol{\theta}} J(\boldsymbol{\theta}), \boldsymbol{\delta} \rangle, \tag{8}$$

The corresponding Lagrangian of problem (8) is

$$\mathcal{L}(\boldsymbol{\delta}, \lambda) = -\langle \nabla_{\boldsymbol{\theta}} J(\boldsymbol{\theta}), \boldsymbol{\delta} \rangle + \lambda(\mathrm{KL}(p_{\boldsymbol{\theta}+\boldsymbol{\delta}} \| p_{\boldsymbol{\theta}}) - \rho^2), \tag{9}$$

where $\lambda$ is the Lagrange multiplier. We can see that problem (9) is convex with respect to $\boldsymbol{\delta}$. Therefore, by setting the derivatives w.r.t. $\boldsymbol{\delta}_{\boldsymbol{\mu}}$ and $\boldsymbol{\delta}_{\boldsymbol{\Sigma}}$ to zero, we can obtain $\boldsymbol{\delta}$ as

$$\boldsymbol{\delta}_{\boldsymbol{\mu}}(\boldsymbol{\theta}) = \frac{1}{\lambda} \boldsymbol{\Sigma} \nabla_{\boldsymbol{\mu}} J(\boldsymbol{\theta}), \quad \boldsymbol{\delta}_{\boldsymbol{\Sigma}}(\boldsymbol{\theta}) = \frac{2 \boldsymbol{\Sigma} \nabla_{\boldsymbol{\Sigma}} J(\boldsymbol{\theta})}{\lambda \boldsymbol{\Sigma}^{-1} - 2 \nabla_{\boldsymbol{\Sigma}} J(\boldsymbol{\theta})}. \tag{10}$$

As shown in Eq. (10), to calculate the perturbation, we need to calculate the inverse covariance matrix, i.e., $\boldsymbol{\Sigma}^{-1}$, which is computationally expensive for high-dimensional problems. Therefore, assuming that $\boldsymbol{\Sigma}$ is a diagonal matrix can significantly reduce the computation cost. Then by plugging $\boldsymbol{\delta}_{\boldsymbol{\mu}}(\boldsymbol{\theta})$ and $\boldsymbol{\delta}_{\boldsymbol{\Sigma}}(\boldsymbol{\theta})$ into the neighborhood constraint, i.e., $\boldsymbol{\delta} \in \mathcal{C}(\boldsymbol{\theta})$, we can determine the optimal $\lambda$ as

$$\lambda = \frac{1}{\rho} \sqrt{\|\boldsymbol{\Sigma} \nabla_{\boldsymbol{\Sigma}} J(\boldsymbol{\theta})\|_{\mathrm{F}}^2 + 0.5 \|\boldsymbol{\Sigma}^{\frac{1}{2}} \nabla_{\boldsymbol{\mu}} J(\boldsymbol{\theta})\|_2^2}. \tag{11}$$

Based on Eqs. (10) and (11), we obtain the approximated closed-form solution $\boldsymbol{\delta}(\boldsymbol{\theta})$ for a given $\boldsymbol{\theta}$. With $\boldsymbol{\delta}(\boldsymbol{\theta})$, the minimization subproblem of problem (7) can be reformulated as

$$\min_{\boldsymbol{\theta}} J(\boldsymbol{\theta} + \boldsymbol{\delta}(\boldsymbol{\theta})). \tag{12}$$

To solve problem (12), in the $t$-th iteration, we add a regularization term $\frac{1}{\beta_t} \mathrm{KL}(p_{\boldsymbol{\theta}} \| p_{\boldsymbol{\theta}_t})$ to problem (12) to enforce $\boldsymbol{\theta}$ to be close to $\boldsymbol{\theta}_t$, and obtain $\boldsymbol{\theta}_{t+1}$ by solving the following problem as

$$\boldsymbol{\theta}_{t+1} = \arg\min_{\boldsymbol{\theta}} J(\boldsymbol{\theta} + \boldsymbol{\delta}(\boldsymbol{\theta_t})) - J(\boldsymbol{\theta}_t) + \frac{1}{\beta_t} \mathrm{KL}(p_{\boldsymbol{\theta}} \| p_{\boldsymbol{\theta}_t}). \tag{13}$$

Following standard SAM-based methods (Foret et al., 2021), we treat $\delta(\boldsymbol{\theta}_t)$ as a constant $\delta_t$ instead of a function of $\boldsymbol{\theta}_t$ to accelerate the computation. Then by using the first-order Taylor expansion, problem (13) can be approximated as

$$\boldsymbol{\theta}_{t+1} = \arg\min_{\boldsymbol{\theta}} \langle \boldsymbol{\theta} - \boldsymbol{\theta}_t, \nabla_{\boldsymbol{\theta}} J(\boldsymbol{\theta}_t + \boldsymbol{\delta}_t) \rangle + \frac{1}{\beta_t} \mathrm{KL}(p_{\boldsymbol{\theta}} \| p_{\boldsymbol{\theta}_t}). \tag{14}$$

By solving problem (14), we can obtain the update formulations for $\boldsymbol{\mu}$ and $\boldsymbol{\Sigma}$ in the $t$-th iteration as

$$\boldsymbol{\mu}_{t+1} = \boldsymbol{\mu}_t - \beta_t \boldsymbol{\Sigma}_t \nabla_{\boldsymbol{\mu}} J(\boldsymbol{\theta}_t + \boldsymbol{\delta}_t), \quad \boldsymbol{\Sigma}_{t+1}^{-1} = \boldsymbol{\Sigma}_t^{-1} + 2\beta_t \nabla_{\boldsymbol{\Sigma}} J(\boldsymbol{\theta}_t + \boldsymbol{\delta}_t), \tag{15}$$

where $\nabla_{\boldsymbol{\mu}} J(\boldsymbol{\theta}_t + \boldsymbol{\delta}_t)$ and $\nabla_{\boldsymbol{\Sigma}} J(\boldsymbol{\theta}_t + \boldsymbol{\delta}_t)$ denote the derivative of $J(\boldsymbol{\theta})$ w.r.t. $\boldsymbol{\mu}$ and $\boldsymbol{\Sigma}$ at $\boldsymbol{\mu} = \boldsymbol{\theta}_t + \boldsymbol{\delta}_{\boldsymbol{\mu}_t}$ and $\boldsymbol{\Sigma} = \boldsymbol{\Sigma}_t + \boldsymbol{\delta}_{\boldsymbol{\Sigma}_t}$, respectively.

## 3.2 UPDATE FORMULATIONS FOR SABO

The gradients of the reparameterized objective $J(\boldsymbol{\theta})$ w.r.t. $\boldsymbol{\mu}$ and $\boldsymbol{\Sigma}$ rely on the expectations of the black-box function and can be obtained with only function queries (Wierstra et al., 2014) (see Theorem A.1 in Appendix A.4). Hence, we estimate them by Monte Carlo sampling. Specifically, the stochastic approximation of the gradients $\nabla_{\boldsymbol{\mu}} J(\boldsymbol{\theta}_t)$ and $\nabla_{\boldsymbol{\Sigma}} J(\boldsymbol{\theta}_t)$ are given as

$$\boldsymbol{g}_t' = \frac{1}{N} \sum_{j=1}^{N} \boldsymbol{\Sigma}_t^{-1}(\boldsymbol{x}_j' - \boldsymbol{\mu}_t)\big(F(\boldsymbol{x}_j') - F(\boldsymbol{\mu}_t)\big), \tag{16}$$

$$\boldsymbol{G}_t' = \frac{1}{2N} \sum_{j=1}^{N} \mathrm{diag}\Big[\boldsymbol{\Sigma}_t^{-1}\big[\mathrm{diag}\big((\boldsymbol{x}_j' - \boldsymbol{\mu}_t)(\boldsymbol{x}_j' - \boldsymbol{\mu}_t)^{\top}\boldsymbol{\Sigma}_t^{-1} - \boldsymbol{I}\big)(F(\boldsymbol{x}_j') - F(\boldsymbol{\mu}_t))\big]\Big], \tag{17}$$

where $\boldsymbol{x}_j'$ denotes the $j$-th sample sampled from the distribution $\mathcal{N}(\boldsymbol{x} \mid \boldsymbol{\mu}_t, \boldsymbol{\Sigma}_t)$. Note that $\boldsymbol{g}_t'$ is an unbiased estimator for the gradient $\nabla_{\boldsymbol{\mu}} J(\boldsymbol{\theta})$ as proved in Lemma C.5, and inspired by Lyu & Tsang (2021), we subtract $F(\boldsymbol{\mu}_t)$ to improve the computational stability while keeping them as unbiased estimations. To avoid the scaling problem, in practice, we can employ a monotonic transformation strategy and the details can be found in Appendix F.

Then according to Eq. (10), we obtain the perturbation $\boldsymbol{\delta}_t$ in the $t$-th iteration as

$$\boldsymbol{\delta}_t = \left\{ \frac{1}{\lambda} \boldsymbol{\Sigma}_t \boldsymbol{g}_t', \frac{2\boldsymbol{\Sigma}_t \boldsymbol{G}_t'}{\lambda \boldsymbol{\Sigma}_t^{-1} - 2\boldsymbol{G}_t'} \right\}, \tag{18}$$

where $\lambda$ is approximated by $\frac{1}{\rho}\sqrt{\|\boldsymbol{\Sigma}_t \boldsymbol{G}_t'\|_{\mathrm{F}}^2 + 0.5\|\boldsymbol{\Sigma}_t^{\frac{1}{2}} \boldsymbol{g}_t'\|_2^2}$. Similarly, the gradients $\nabla_{\boldsymbol{\mu}} J(\boldsymbol{\theta}_t + \boldsymbol{\delta}_t)$ and $\nabla_{\boldsymbol{\Sigma}} J(\boldsymbol{\theta}_t + \boldsymbol{\delta}_t)$ can be approximated as follows:

$$\boldsymbol{g}_t = \frac{1}{N} \sum_{j=1}^{N} \widehat{\boldsymbol{\Sigma}}_t^{-1}(\boldsymbol{x}_j - \widehat{\boldsymbol{\mu}}_t)\big(F(\boldsymbol{x}_j) - F(\widehat{\boldsymbol{\mu}}_t)\big), \tag{19}$$

$$\boldsymbol{G}_t = \frac{1}{2N} \sum_{j=1}^{N} \mathrm{diag}\Big[\widehat{\boldsymbol{\Sigma}}_t^{-1}\big[\mathrm{diag}\big((\boldsymbol{x}_j - \widehat{\boldsymbol{\mu}}_t)(\boldsymbol{x}_j - \widehat{\boldsymbol{\mu}}_t)^{\top}\widehat{\boldsymbol{\Sigma}}_t^{-1} - \boldsymbol{I}\big)(F(\boldsymbol{x}_j) - F(\widehat{\boldsymbol{\mu}}_t))\big]\Big], \tag{20}$$

where $\widehat{\boldsymbol{\Sigma}}_t = \boldsymbol{\Sigma}_t + \boldsymbol{\delta}_{\boldsymbol{\Sigma}_t}$, $\widehat{\boldsymbol{\mu}}_t = \boldsymbol{\mu}_t + \boldsymbol{\delta}_{\boldsymbol{\mu}_t}$, and $\boldsymbol{x}_j$ denotes the $j$-th sample sampled from the distribution $\mathcal{N}(\boldsymbol{x} \mid \widehat{\boldsymbol{\mu}}_t, \widehat{\boldsymbol{\Sigma}}_t)$. Then the updated formulations for $\boldsymbol{\mu}$ and $\boldsymbol{\Sigma}$ are rewritten as

$$\boldsymbol{\mu}_{t+1} = \boldsymbol{\mu}_t - \beta_t \boldsymbol{\Sigma}_t \boldsymbol{g}_t, \quad \boldsymbol{\Sigma}_{t+1}^{-1} = \boldsymbol{\Sigma}_t^{-1} + 2\beta_t \boldsymbol{G}_t. \tag{21}$$

The entire algorithm is shown in Algorithm 1. To avoid the scaling problem, in practice, we can employ monotonic transformation for the objective, more details can be found in Appendix F. For the proposed SABO method, the computation cost per iteration is of order $\mathcal{O}(Nd)$.

**Mini-batch SABO.** In Algorithm 1, we assume full access to the objective function $F(\boldsymbol{x}; \mathcal{D})$, while in practice, a full-batch function query might be costly. Therefore, we can perform a mini-batch function query. Specifically, in each iteration, we query the expected fitness by a mini-batch of data and approximate $F(\boldsymbol{x}; \mathcal{D})$ by

$$F(\boldsymbol{x}; \mathcal{B}) = \frac{1}{|\mathcal{B}|} \sum_{(X,y) \in \mathcal{B}} l(\boldsymbol{x}; (X, y)), \tag{22}$$

where $|\mathcal{B}|$ denotes the number of data in the mini-batch $\mathcal{B}$. The corresponding expected fitness of $F(\boldsymbol{x}; \mathcal{B})$ under the distribution $p_{\boldsymbol{\theta}}(\boldsymbol{x})$ is formulated as $J(\boldsymbol{\theta}; \mathcal{B}) = \mathbb{E}_{p_{\boldsymbol{\theta}}(\boldsymbol{x})}[F(\boldsymbol{x}; \mathcal{B})]$. Then similar to the full-batch function query setting, we can approximate the gradients $\nabla_{\boldsymbol{\mu}} J(\boldsymbol{\theta}_t, \mathcal{B})$ and $\nabla_{\boldsymbol{\Sigma}} J(\boldsymbol{\theta}_t, \mathcal{B})$, where the detailed formulations can be found in Appendix B. The entire SABO algorithm with mini-batch function queries is shown in Algorithm 2 in Appendix B.

---

**Algorithm 1** SABO

---

**Require:** Neighborhood size $\rho$, learning rate $\beta_t$
1: Initialized $\boldsymbol{\theta}_0 = (\boldsymbol{\mu}_0, \boldsymbol{\Sigma}_0)$ ;
2: **for** $t = 0$ to $T - 1$ **do**
3:     Take i.i.d. samples $\boldsymbol{z}'_j \sim \mathcal{N}(0, I)$ and set $\boldsymbol{x}'_j = \boldsymbol{\mu}_t + \boldsymbol{\Sigma}_t^{\frac{1}{2}} \boldsymbol{z}'_j$ for $j \in \{1, \dots, N\}$;
4:     Query the batch observations $\{F(\boldsymbol{x}'_1), \dots, F(\boldsymbol{x}'_N)\}$;
5:     Compute the gradient $\boldsymbol{g}'_t$ via Eq. (16) and compute the gradient $\boldsymbol{G}'_t$ via Eq. (17);
6:     Compute $\lambda = \frac{1}{\rho}\sqrt{\|\boldsymbol{\Sigma}_t \boldsymbol{G}'_t\|_F^2 + 0.5\|\boldsymbol{\Sigma}_t^{\frac{1}{2}} \boldsymbol{g}'_t\|_2^2}$;
7:     Compute $\boldsymbol{\delta}_{\boldsymbol{\mu}_t}$ and $\boldsymbol{\delta}_{\boldsymbol{\Sigma}_t}$ via Eq. (18);
8:     Take i.i.d. samples $\boldsymbol{z}_j \sim \mathcal{N}(0, I)$ for $j \in \{1, \dots, N\}$;
9:     Set $\boldsymbol{x}_j = \boldsymbol{\mu}_t + \boldsymbol{\delta}_{\boldsymbol{\mu}_t} + (\boldsymbol{\Sigma}_t + \boldsymbol{\delta}_{\boldsymbol{\Sigma}_t})^{\frac{1}{2}} \boldsymbol{z}_j$ for $j \in \{1, \dots, N\}$;
10:    Query the batch observations $\{F(\boldsymbol{x}_1), \dots, F(\boldsymbol{x}_N)\}$;
11:    Compute the gradient $\boldsymbol{g}_t$ via Eq. (19) and compute the gradient $\boldsymbol{G}_t$ via Eq. (20);
12:    Set $\boldsymbol{\mu}_{t+1} = \boldsymbol{\mu}_t - \beta_t \boldsymbol{\Sigma}_t \boldsymbol{g}_t$ and set $\boldsymbol{\Sigma}_{t+1}^{-1} = \boldsymbol{\Sigma}_t^{-1} + 2\beta_t \boldsymbol{G}_t$;
13: **end for**
14: **return** $\boldsymbol{\theta}_T = (\boldsymbol{\mu}_T, \boldsymbol{\Sigma}_T)$.

---

## 4 ANALYSIS

In this section, we provide comprehensive theoretical analyses for the proposed SABO method with all the detailed proofs in Appendix D.

### 4.1 CONVERGENCE ANALYSIS OF SABO

Firstly, we make an assumption for the reparameterized objective function.

**Assumption 4.1** *The function $F(\boldsymbol{x})$ is $L_F$-Lipschitz w.r.t. $\boldsymbol{x}$. The function $J(\boldsymbol{\theta})$ satisfies that $\nabla_{\boldsymbol{\mu}} J(\boldsymbol{\theta})$ is $L$-Lipschitz w.r.t. $\boldsymbol{\theta} = \{\boldsymbol{\mu}, \boldsymbol{\Sigma}\} \in \Theta$, where $\Theta := \{\boldsymbol{\mu}, \boldsymbol{\Sigma} \mid \boldsymbol{\mu} \in \mathbb{R}^d, \boldsymbol{\Sigma} \in \mathcal{S}^+\}$.*

Note that the proposed SABO algorithm approximates the gradients of the reparameterized objective function. It is necessary to study the relation between the optimal solutions of the original objective functions $F(\boldsymbol{x})$ and $J(\boldsymbol{\theta})$, and we put the results in the following proposition.

**Proposition 4.2** *(Lyu & Tsang, 2021) Suppose $p_{\boldsymbol{\theta}}(\boldsymbol{x})$ is a Gaussian distribution with $\boldsymbol{\theta} = \{\boldsymbol{\mu}, \boldsymbol{\Sigma}\}$ and $F(\boldsymbol{x})$ is a convex function. Let $J(\boldsymbol{\theta}) = \mathbb{E}_{p_{\boldsymbol{\theta}}}[F(\boldsymbol{x})]$, and $J(\boldsymbol{\mu}^*, \boldsymbol{0}) := F(\boldsymbol{\mu}^*)$. Then we have*

$$F(\boldsymbol{\mu}) - F(\boldsymbol{\mu}^*) \leq J(\boldsymbol{\mu}, \boldsymbol{\Sigma}) - J(\boldsymbol{\mu}^*, \boldsymbol{0}), \tag{23}$$

*where $\boldsymbol{0}$ denotes a zero matrix with appropriate size.*

The convexity assumption of the objective function in Theorem 4.3 has been widely adopted in the area of stochastic gradient approximation black-box optimization (Beyer, 2014; Wierstra et al., 2014; Lyu & Tsang, 2021; Ye, 2023). Since the Gaussian-smooth approximation function is always an upper bound of the true target function in convex cases, i.e., $F(\mu) \leq \mathbb{E}_{\mathcal{N}(\mu, \Sigma)}[F(x)]$. When $\boldsymbol{\mu}^*$ is an optimal solution of minimization problem $\min_{\boldsymbol{x}} F(\boldsymbol{x})$, Proposition 4.2 implies that the difference between the objective value at $\boldsymbol{\mu}$ and the optimal objective value of the original problem is upper-bounded by that of the expected objective function. Then for Algorithm 1, the following theorem captures the convergence of $\boldsymbol{\mu}$ for a convex objective function.

**Theorem 4.3** *Suppose that $F(\boldsymbol{x})$ is a convex function, $J(\boldsymbol{\theta})$ is $c$-strongly convex w.r.t. $\boldsymbol{\mu}$, the gradient estimator $\boldsymbol{G}_t$ (w.r.t. the covariance matrix) is positive semi-definite matrix such that $\xi \boldsymbol{I} \preceq \boldsymbol{G}_t \preceq \frac{c\boldsymbol{I}}{4}$ with $\xi \geq 0$, $\boldsymbol{\Sigma}_0 \in \mathcal{S}^+$, and $\boldsymbol{\Sigma}_0 \preceq R\boldsymbol{I}$ where $R > 0$. Suppose the sequence $\{\boldsymbol{\mu}_t\}$ generated by Algorithm 1 satisfies that the distance between the sequence $\{\boldsymbol{\mu}_t\}$ and the optimal solution of $F(\boldsymbol{x})$ is bounded, i.e., $\|\boldsymbol{\mu}_t - \boldsymbol{\mu}^*\| \leq D$, $\|\nabla_{\boldsymbol{\Sigma}=\boldsymbol{\Sigma}_t} J(\boldsymbol{\theta})\|_F \leq H$, $\beta_t = \mathcal{O}(1)$, and $\rho < \frac{\sqrt{d}}{2}$ satisfies $\rho = \mathcal{O}(\frac{1}{\sqrt{T}})$, then with Assumption 4.1, we have*

$$\frac{1}{T}\sum_{t=0}^{T-1} \mathbb{E}\left[J(\boldsymbol{\mu}_{t+1}, \boldsymbol{\Sigma}_t) - J(\boldsymbol{\mu}^*, 0)\right] = \mathcal{O}\left(\frac{\log T}{T}\right). \tag{24}$$

Based on Theorem 4.3 and Proposition 4.2, when $\beta_t = \mathcal{O}(1)$ and $\rho = \mathcal{O}(\frac{1}{\sqrt{T}})$, we have

$$\frac{1}{T} \sum\nolimits_{t=0}^{T-1} \mathbb{E}\left[F(\boldsymbol{\mu}_{t+1}) - F(\boldsymbol{\mu}^*)\right] = \mathcal{O}(\frac{\log T}{T}). \tag{25}$$

Therefore, the proposed SABO algorithm with full-batch function query possesses a convergence rate $\mathcal{O}(\frac{\log T}{T})$ for the convex objective function. Additionally, in Theorem 4.3, when $\beta = \mathcal{O}(1)$ and $\rho = \mathcal{O}(1)$, the proposed SABO algorithm still maintains a convergence rate $\mathcal{O}(T^{-\frac{1}{2}})$. The detailed discussion is provided in Remark D.1.

For the mini-batch setting, we make an additional assumption for the objective function $F(\boldsymbol{x}; \mathcal{D})$.

**Assumption 4.4** *It is assumed that the datasets $\mathcal{D}$ and $\mathcal{B}$ are i.i.d. sampled from a data distribution $P(X, y)$, and the variance of the mini-batch estimation of the objective function is bounded, i.e., $\|F(\boldsymbol{x}; \mathcal{B}) - \mathbb{E}F(\boldsymbol{x}; \mathcal{B})\|_2^2 \leq \varepsilon_{\mathrm{B}}^2$ and $\|F(\boldsymbol{x}; \mathcal{D}) - \mathbb{E}F(\boldsymbol{x}; \mathcal{D})\|_2^2 \leq \varepsilon_{\mathrm{D}}^2$.*

Note that for the standard stochastic gradient descent (SGD) method (Shamir & Zhang, 2013), the unbiased estimation and bounded variance assumptions were made for the approximated gradient. However, in black-box optimization, the gradient of the objective function $F(\boldsymbol{x})$ w.r.t. $\boldsymbol{x}$ is unavailable. Hence we can only make assumptions for the batch estimations $F(\boldsymbol{x}; \mathcal{B})$ and $F(\boldsymbol{x}; \mathcal{D})$.

Then we have the following result for the mini-batch estimation.

**Proposition 4.5** *Suppose Assumption 4.4 holds, then we have $\mathbb{E}F(\boldsymbol{x}; \mathcal{B}) = \mathbb{E}F(\boldsymbol{x}; \mathcal{D})$, and $\|F(\boldsymbol{x}; \mathcal{B}) - F(\boldsymbol{x}; \mathcal{D})\|_2^2 \leq \varepsilon^2$, where $\varepsilon^2 = 2(\varepsilon_{\mathrm{B}}^2 + \varepsilon_{\mathrm{D}}^2)$.*

Then with Assumption 4.4, the following theorem shows the convergence of $\boldsymbol{\mu}$ for Algorithm 2.

**Theorem 4.6** *Suppose that $F(\boldsymbol{x})$ is a convex function, $J(\boldsymbol{\theta})$ is $c$-strongly convex w.r.t. $\boldsymbol{\mu}$, the gradient estimator $\boldsymbol{G}_t$ (w.r.t. the covariance matrix) is positive semi-definite matrix such that $\xi \boldsymbol{I} \preceq \boldsymbol{G}_t \preceq \frac{c\boldsymbol{I}}{4}$ with $\xi \geq 0$, $\boldsymbol{\Sigma}_0 \in \mathcal{S}^+$, and $\boldsymbol{\Sigma}_0 \preceq R\boldsymbol{I}$ where $R > 0$. Suppose the sequence $\{\boldsymbol{\mu}_t\}$ generated by Algorithm 2 satisfies that the distance between the sequence $\{\boldsymbol{\mu}_t\}$ and the optimal solution of $F(\boldsymbol{x})$ is bounded, i.e., $\|\boldsymbol{\mu}_t - \boldsymbol{\mu}^*\| \leq D$, $\|\nabla_{\boldsymbol{\Sigma} = \boldsymbol{\Sigma}_t} J(\boldsymbol{\theta})\|_F \leq H$, $\beta_t = \mathcal{O}(1)$, and $\rho < \frac{\sqrt{d}}{2}$ satisfies $\rho = \mathcal{O}(1)$. Then with Assumptions 4.1 and 4.4, we have*

$$\frac{1}{T} \sum\nolimits_{t=0}^{T-1} \mathbb{E}\left[J(\boldsymbol{\mu}_{t+1}, \boldsymbol{\Sigma}_t) - J(\boldsymbol{\mu}^*, 0)\right] = \mathcal{O}\left(\frac{1}{\sqrt{T}}\right). \tag{26}$$

Based on Theorem 4.6 and Proposition 4.2, the proposed SABO algorithm with mini-batch function query possesses a convergence rate $\mathcal{O}(T^{-\frac{1}{2}})$.

**Remark 4.7** *Note that in Theorem 4.3 and Theorem 4.6, the convergence results do not require the objective function $F(\boldsymbol{x})$ to be strongly convex or differentiable. Hence, the convergence holds for a non-smooth convex function $F(\boldsymbol{x})$ (as long as $J(\boldsymbol{\theta})$ being a $c$-strongly convex function w.r.t. $\boldsymbol{\mu}$).*

## 4.2 GENERALIZATION ERROR ANALYSIS

In this subsection, we analyze the generalization bound of the proposed SABO algorithm. Specifically, we bound the expectation of the objective function over the Gaussian perturbation.

We denote by $(X, y)$ a data pair drawn from a data distribution $P(X, y)$ and by $F(\boldsymbol{x}; (X, y))$ the corresponding loss of parameter $\boldsymbol{x}$ on $(X, y)$. So we have $F(\boldsymbol{x}; (X, y)) = l(\boldsymbol{x}; (X, y))$. We define the population loss over the data distribution $P(X, y)$ as $\mathbb{E}_{P(X,y)}[F(\boldsymbol{x}; (X, y))]$, and the empirical loss over a dataset $\mathcal{S}$, which consists of $M$ i.i.d. samples drawn from $P(X, y)$, as $F(\boldsymbol{x}; \mathcal{S}) = \frac{1}{M} \sum_{i=1}^M l(\boldsymbol{x}; (X_i, y_i))$. Then we have following result.

**Theorem 4.8** *Let the loss function $F(\boldsymbol{x}; (X, y))$ be a convex function w.r.t. $\boldsymbol{x}$, then for any $\boldsymbol{\mu} \in \mathbb{R}^d$, with probability at least $1 - \kappa$, we have*

$$\mathbb{E}_{P(X,y)}[F(\boldsymbol{\mu}; (X, y))] \leq \max_{\boldsymbol{\delta} \in \mathcal{C}(\boldsymbol{\theta})} \mathbb{E}_{p_{\boldsymbol{\theta}+\boldsymbol{\delta}}}\left[F(\boldsymbol{x}; S)\right] + \sqrt{\frac{\rho^2 + \log(\frac{M}{\kappa})}{2(M-1)}}, \tag{27}$$

*where $p_{\boldsymbol{\theta}} := \mathcal{N}(\boldsymbol{\mu}, \boldsymbol{\Sigma})$, $\mathcal{C}(\boldsymbol{\theta}) = \{\boldsymbol{\delta} \mid \mathrm{KL}(p_{\boldsymbol{\theta}+\boldsymbol{\delta}} \| p_{\boldsymbol{\theta}}) \leq \rho^2\}$, and $\mathcal{S}$ denotes the training set that consists of $M$ i.i.d. samples drawn from data distribution $P(X, y)$.*

Theorem 4.8 provides a generalization bound for the proposed SABO algorithm. Compared with the generalization bound of SAM presented in Appendix A.1 of Foret et al. (2021), Theorem 4.8 has an asymptotically identical order in the complexity term. However, the expected generalization loss on the right-hand side of Eq. (27) is different in that we reparameterize the objective function and have perturbation of $\boldsymbol{\theta}$ in its neighborhood of a statistical manifold, i.e., $\boldsymbol{\delta} \in \mathcal{C}(\boldsymbol{\theta})$, while SAM bounds the generalization loss averaged over a spherical Gaussian perturbation on parameters.

## 5 RELATED WORKS

**Black-Box Optimization.** Many methods have been proposed for black-box optimization, including Bayesian optimization (BO) methods (Srinivas et al., 2010; Gardner et al., 2017; Nayebi et al., 2019), stochastic optimization methods such as evolution strategies (ES) (Hansen, 2006; Wierstra et al., 2014; Lyu & Tsang, 2021), and genetic algorithms (GA) (Srinivas & Patnaik, 1994; Mirjalili, 2019). Among those methods, BO achieves good performance for low-dimensional problems, but it often fails to handle high-dimensional problems through a global surrogate model, as shown in (Eriksson et al., 2019) and (Nguyen et al., 2022). As a result, TuRBO (Eriksson et al., 2019) and GIBO (Nguyen et al., 2022) try to address this problem with the local BO approach. (Ziomek & Ammar, 2023) further showed that decomposition is important for alleviating the high-dimensional problems in BO. Although BO is not our main focus, we further compare our method with these local Bayesian optimization methods on the black-box prompt fine-tuning problem, and the corresponding exploration is shown in Appendix H.1. GA method is computationally expensive for machine learning problems and usually lacks convergence analysis. The stochastic optimization methods such as CMA-ES (Hansen, 2006) and INGO (Lyu & Tsang, 2021) can scale up to higher-dimensional problems compared with BO. Hence we mainly consider stochastic optimization methods as baseline methods in our experiments.

**Sharpness-Aware Minimization.** SAM has been widely studied for improving the model generalization. Among previous works on SAM (Kwon et al., 2021; Zhuang et al., 2022; Zhao et al., 2022; Jiang et al., 2023; Tahmasebi et al., 2024), the most relevant method to our approach is the FSAM method (Kim et al., 2022) which also finds the worst-case objective function via a statistical manifold instead of the Euclidean space. However, the loss function of the model studied in FSAM is a predictive distribution conditional on both model parameters and data, while in our case, we consider the parameter as a Gaussian distribution. The bSAM method (Möllenhoff & Khan, 2022) builds a connection between the SAM objective and Bayes objective by Fenchel biconjugate of the loss function. Möllenhoff & Khan (2022) shares a similarity with our work in developing SAM w.r.t. the expected loss. However, The bSAM method relies on the derivation of a convex lower bound of the expected loss by the Fenchel biconjugate and the perturbation is still w.r.t. each point inside the expected loss as standard SAM. Hence FSAM and bSAM are different from the proposed SABO method. Additionally, like other variants of SAM, FSAM and bSAM are both inapplicable to black-box optimization. The STABLEOPT method (Bogunovic et al., 2018) proposes a SAM-like optimization formulation in the Bayesian optimization area. They aim to improve the robustness w.r.t. the adversarial perturbation of the return point by a GP-based optimization. Their method relies on a GP surrogate model that is expensive for training and inference. In addition, it is challenging for the proposed adversarial robust GP-based optimization to handle high-dimensional problems. In contrast, our work aims to improve the generalization property in high-dimensional black-box optimization.

## 6 EMPIRICAL STUDY

In this section, we empirically evaluate the proposed SABO method, and compare it with four representative black-box methods, i.e., CMA-ES (Hansen, 2006), MMES (He et al., 2020), BES (Gao & Sener, 2022), and INGO (Lyu & Tsang, 2021). All the experiments are conducted on a single NVIDIA GeForce RTX 3090 GPU.

## 6.1 SYNTHETIC PROBLEMS

To verify the convergent results of the proposed SABO method in Section 4. We compare the proposed SABO method with baseline methods on minimizing four $d$-dimensional synthetic benchmark test functions, i.e., ellipsoid function, $l_{\frac{1}{2}}$-ellipsoid function, different powers function, and Levy function. All the test functions are listed in Appendix G.1.

The results are evaluated by calculating the Euclidean distance between the solution $x$ and the optimal solution $x^*$, i.e., $\mathcal{E} = \|x - x^*\|_2$. We then assess the baseline methods using varying dimensions, i.e., $d \in \{200, 500, 1000\}$. *Due to the page limitation, the implementation details and more detailed experimental results are put in Appendix G.1.*

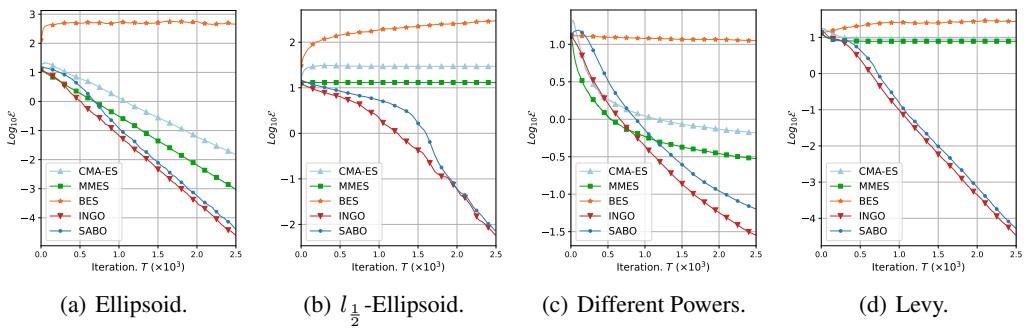

(a) Ellipsoid.     (b) $l_{\frac{1}{2}}$-Ellipsoid.     (c) Different Powers.     (d) Levy.

Figure 1: Results on the four test functions with problem dimension $d = 500$ and $N = 50$.

**Result.** Figure 1 shows the results on four $d$-dimensional synthetic problems with $d = 500$ and population size $N = 50$. The proposed SABO method approximately achieves a linear convergence rate similar to the INGO method. This is reasonable as these two methods have the same theoretical convergent rate, i.e., $\mathcal{O}(\frac{\log T}{T})$, according to Theorem 4.3 and Theorem 5 in Lyu & Tsang (2021). Since the SABO method perturbs the main objective in each iteration, its practical convergence speed is slightly slower than INGO. The CMA-ES method and MMES methods can converge on the ellipsoid problem and the different powers problem, but they do not converge as fast as SABO and INGO methods. Moreover, CMA-ES and MMES methods fail on the $l_{\frac{1}{2}}$-ellipsoid problem and Levy problem. The BES method fails on all test problems. This shows that it could be challenging for BES to optimize non-smooth or high-dimensional test functions without adaptively updating mean and covariance. These results demonstrate the superiority of the SABO method in optimizing high-dimensional problems, and verify our theoretical convergence results.

## 6.2 BLACK-BOX PROMPT FINE-TUNING

Black-box prompt fine-tuning of large language models (Ding et al., 2023; Sun et al., 2022b;a; 2023) is a promising direction to achieve expertise models efficiently for downstream tasks. In such an LMaaS setting, we cannot access the model parameter and can only tune their prompts without backpropagation. We evaluate the proposed SABO method in improving generalization performance on the black-box prompt fine-tuning task.

**Datasets.** We conduct experiments on six language understanding benchmark datasets: *SST-2* (Socher et al., 2013) and *Yelp polarity* (Zhang et al., 2015) for sentiment analysis, *AG's News* (Zhang et al., 2015) for topic classification, *MRPC* (Dolan & Brockett, 2005) for paraphrase, *RTE* (Wang et al., 2018) and *SNLI* (Bowman et al., 2015) for natural language inference. Each dataset contains a classification task. The statistics of six datasets are summarized in Table 1 of (Sun et al., 2022b). By following (Sun et al., 2022b), the testing accuracy is used to measure the performance of all the methods on the *SST-2*, *AG's News*, *RTE*, *SNLI*, and *Yelp P.* datasets, and the F1 score is used to measure the performance on the *MRPC* datasets.

**Implementation Details.** Following Sun et al. (2022b), we employ a fixed randomly initialized matrix $A \in \mathbb{R}^{d \times D}$ to project a vector $v \in \mathbb{R}^d$ onto the token embedding space $\mathbb{R}^D$. Then we

Table 1: Performance (%) on *SST-2*, *AG's News*, *MRPC*, *RTE*, *SNLI* and *Yelp P.* datasets. We report the mean and standard deviation over 3 random seeds. The best result across all groups is highlighted in **bold** and the best result in each group is marked with underlined.

| Methods | SST-2 | AG's News | MRPC | RTE | SNLI | Yelp P. |
|---|---|---|---|---|---|---|
| Zero-shot | 79.82 | 76.96 | 67.40 | 51.62 | 38.82 | 89.64 |
| | | | *Dimension $d = 200$* | | | |
| CMA-ES | $85.74_{\pm 0.35}$ | $82.09_{\pm 0.56}$ | $74.98_{\pm 2.16}$ | $51.02_{\pm 2.14}$ | $34.27_{\pm 1.18}$ | $90.57_{\pm 0.05}$ |
| MMES | $83.98_{\pm 0.78}$ | $80.52_{\pm 0.99}$ | $76.54_{\pm 4.34}$ | $48.50_{\pm 0.45}$ | $40.39_{\pm 1.83}$ | $90.94_{\pm 0.36}$ |
| BES | $83.52_{\pm 0.11}$ | $75.44_{\pm 0.31}$ | $79.23_{\pm 0.20}$ | $53.07_{\pm 0.29}$ | $38.73_{\pm 0.17}$ | $89.65_{\pm 0.01}$ |
| INGO | $83.57_{\pm 0.11}$ | $76.47_{\pm 0.03}$ | $78.87_{\pm 0.20}$ | $53.07_{\pm 0.00}$ | $38.86_{\pm 0.06}$ | $89.84_{\pm 0.04}$ |
| **SABO** | $\underline{87.88}_{\pm 0.53}$ | $\underline{82.22}_{\pm 0.41}$ | $\underline{79.35}_{\pm 0.12}$ | $\mathbf{53.67}_{\pm 0.17}$ | $\underline{40.72}_{\pm 0.15}$ | $\underline{91.50}_{\pm 0.13}$ |
| | | | *Dimension $d = 500$* | | | |
| CMA-ES | $86.12_{\pm 0.59}$ | $82.50_{\pm 0.23}$ | $77.10_{\pm 1.90}$ | $52.71_{\pm 0.51}$ | $41.34_{\pm 1.49}$ | $91.19_{\pm 0.44}$ |
| MMES | $85.28_{\pm 0.94}$ | $81.67_{\pm 0.80}$ | $77.31_{\pm 1.24}$ | $48.74_{\pm 0.59}$ | $42.07_{\pm 2.62}$ | $91.39_{\pm 0.24}$ |
| BES | $83.56_{\pm 0.05}$ | $75.93_{\pm 0.17}$ | $79.21_{\pm 0.09}$ | $52.95_{\pm 0.17}$ | $38.64_{\pm 0.28}$ | $89.62_{\pm 0.07}$ |
| INGO | $84.29_{\pm 0.34}$ | $76.54_{\pm 0.20}$ | $79.09_{\pm 0.15}$ | $53.19_{\pm 0.17}$ | $38.91_{\pm 0.10}$ | $89.90_{\pm 0.13}$ |
| **SABO** | $\underline{87.31}_{\pm 0.38}$ | $\underline{82.65}_{\pm 0.59}$ | $\underline{79.62}_{\pm 0.07}$ | $\underline{53.55}_{\pm 0.17}$ | $\mathbf{42.29}_{\pm 2.48}$ | $\underline{91.83}_{\pm 0.16}$ |
| | | | *Dimension $d = 1000$* | | | |
| CMA-ES | $86.85_{\pm 0.57}$ | $82.21_{\pm 0.36}$ | $78.98_{\pm 0.17}$ | $52.35_{\pm 0.17}$ | $38.40_{\pm 1.83}$ | $90.46_{\pm 0.62}$ |
| MMES | $84.98_{\pm 0.52}$ | $80.86_{\pm 1.95}$ | $76.43_{\pm 0.82}$ | $49.22_{\pm 1.23}$ | $39.82_{\pm 3.43}$ | $91.63_{\pm 0.20}$ |
| BES | $83.11_{\pm 0.11}$ | $75.66_{\pm 0.09}$ | $79.09_{\pm 0.08}$ | $53.19_{\pm 0.17}$ | $38.57_{\pm 0.13}$ | $89.61_{\pm 0.04}$ |
| INGO | $84.36_{\pm 0.23}$ | $76.35_{\pm 0.14}$ | $78.97_{\pm 0.08}$ | $53.07_{\pm 0.29}$ | $39.05_{\pm 0.06}$ | $89.95_{\pm 0.08}$ |
| **SABO** | $\mathbf{87.96}_{\pm 0.83}$ | $\mathbf{82.77}_{\pm 0.41}$ | $\mathbf{79.68}_{\pm 0.23}$ | $\underline{53.31}_{\pm 0.17}$ | $\underline{40.32}_{\pm 0.27}$ | $\mathbf{91.96}_{\pm 0.41}$ |

optimize the vector $v \in \mathbb{R}^d$ instead of directly optimizing the prompt $p \in \mathbb{R}^D$. The pre-trained RoBERTa$_{\text{LARGE}}$ model (Liu et al., 2019) is used as the backbone model. The matrix $A$ is sampled from the normal distribution as described in Sun et al. (2022a), i.e., $\mathcal{N}(0, \frac{\sigma_e}{\sqrt{d}})$, where $\sigma_e$ is the standard deviation of word embeddings in RoBERTa$_{\text{LARGE}}$. The templates and label words in Table 1 of Sun et al. (2022b) are used to conduct the zero-shot baseline.

For CMA-ES, MMES, BES, INGO, and SABO methods, we employ the cross-entropy loss of training data as the black-box objective for six datasets and optimize the vector $v$ with 100 iterations. The Gaussian distributions are initialized as $\mu_0 = 0$ and $\Sigma_0 = I$, and the population size $N$ is set to 100. We perform a grid search for hyperparameters of INGO, SABO, and BES methods. Specifically, we search the learning rate $\beta$ over $\{0.1, 0.5, 1, 5\}$ for INGO, SABO, and BES, the neighborhood size $\rho$ over $\{10, 50, 100, 500\}$ for SABO, and the spacing $c$ over $\{0.1, 1, 10\}$ for BES. Additionally, we evaluate the performance of all methods on different dimensions of $v$, specifically $d \in \{200, 500, 1000\}$. All the experiments are performed in three independent runs, and the experimental results of mean objective $\pm$ std are reported.

**Results.** Table 1 presents experimental results on these six benchmark datasets for three different dimensions of the vector $v$. We can see that the SABO method consistently outperforms all baselines in terms of testing classification accuracy or testing F1 scores across different settings, highlighting its effectiveness in improving generalization performance. Notably, even in the high-dimensional setting (i.e., $d = 1000$), our method maintains good performance. Moreover, we can see that when $d = 1000$, SABO achieves the best performance on *SST-2*, *AG's News*, *MRPC*, and *Yelp P.* datasets. The SABO method also achieves the best performance on *RTE* and *SNLI* datasets with $d = 200$ and $d = 500$, respectively.

## 7 CONCLUSION

In this work, we have introduced SABO, a novel black-box optimization algorithm that improves generalization by utilizing a sharpness-aware minimization strategy. Theoretically, we provide a convergence guarantee for the proposed SABO algorithm in both full-batch function query and mini-batch function query settings. Additionally, we prove the generalization bound for the proposed method. Empirical studies on synthetic numerical problems verify the convergence properties of the proposed method. Moreover, extensive experimental results on black-box prompt fine-tuning problems demonstrate the effectiveness of the proposed SABO method in improving the generalization performance.

ACKNOWLEDGEMENTS

This work is supported by National Key R&D Program of China 2022ZD0160300 and NSFC key grant under grant no. 62136005. Dr. Yueming Lyu is supported by Career Development Fund no. C243512014 and no. C233312007 of the Agency for Science, Technology and Research. Prof. Sugiyama was supported by the Institute for AI and Beyond, UTokyo and by a grant from Apple, Inc. Any views, opinions, findings, and conclusions or recommendations expressed in this material are those of the authors and should not be interpreted as reflecting the views, policies or position, either expressed or implied, of Apple Inc.

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

# A    ADDITIONAL MATERIAL FOR SECTION 3

## A.1    DETERMINE THE PERTURBATION $\delta$

The Lagrangian of problem (8) is

$$\mathcal{L}(\boldsymbol{\delta}, \lambda) = -\langle \nabla_{\boldsymbol{\theta}} J(\boldsymbol{\theta}), \boldsymbol{\delta} \rangle + \lambda(\mathrm{KL}(p_{\boldsymbol{\theta}+\boldsymbol{\delta}} \| p_{\boldsymbol{\theta}}) - \rho^2) \tag{28}$$

$$= -\boldsymbol{\delta}_{\boldsymbol{\mu}}^{\top} \nabla_{\boldsymbol{\mu}} J(\boldsymbol{\theta}_t) - \mathrm{tr}(\boldsymbol{\delta}_{\boldsymbol{\Sigma}} \nabla_{\boldsymbol{\Sigma}} J(\boldsymbol{\theta}_t)) \tag{29}$$
$$+ \frac{\lambda}{2} \left[ \mathrm{tr}(\boldsymbol{\Sigma}^{-1}(\boldsymbol{\Sigma} + \boldsymbol{\delta}_{\boldsymbol{\Sigma}})) + \boldsymbol{\delta}_{\boldsymbol{\mu}}^{\top} \boldsymbol{\Sigma}^{-1} \boldsymbol{\delta}_{\boldsymbol{\mu}} + \log \frac{|\boldsymbol{\Sigma}|}{|(\boldsymbol{\Sigma} + \boldsymbol{\delta}_{\boldsymbol{\Sigma}})|} - d \right] - \lambda \rho^2.$$

Taking the derivative $\boldsymbol{\delta}_{\boldsymbol{\mu}}$ and $\boldsymbol{\delta}_{\boldsymbol{\Sigma}}$ and setting them to zero, we can obtain that

$$\nabla_{\boldsymbol{\mu}} J(\boldsymbol{\theta}) - \lambda \boldsymbol{\Sigma}^{-1} \boldsymbol{\delta}_{\boldsymbol{\mu}} = 0, \tag{30}$$

$$\nabla_{\boldsymbol{\Sigma}} J(\boldsymbol{\theta}) - \frac{\lambda}{2} [(\boldsymbol{\Sigma} + \boldsymbol{\delta}_{\boldsymbol{\Sigma}})^{-1} - \boldsymbol{\Sigma}^{-1}] = 0. \tag{31}$$

Note that $\boldsymbol{\Sigma}$ is a diagonal matrix. Therefore, we can achieve that

$$\boldsymbol{\delta}_{\boldsymbol{\mu}}(\boldsymbol{\theta}) = \frac{1}{\lambda} \boldsymbol{\Sigma} \nabla_{\boldsymbol{\mu}} J(\boldsymbol{\theta}), \tag{32}$$

$$\boldsymbol{\delta}_{\boldsymbol{\Sigma}}(\boldsymbol{\theta}) = \frac{2 \boldsymbol{\Sigma} \nabla_{\boldsymbol{\Sigma}} J(\boldsymbol{\theta})}{\lambda \boldsymbol{\Sigma}^{-1} - 2 \nabla_{\boldsymbol{\Sigma}} J(\boldsymbol{\theta})}. \tag{33}$$

## A.2    DETERMINE THE OPTIMAL $\lambda$

Note that we have

$$\mathrm{KL}(p_{\boldsymbol{\theta}+\boldsymbol{\delta}} \| p_{\boldsymbol{\theta}}) = \frac{1}{2} \left[ \mathrm{tr}(\boldsymbol{\Sigma}^{-1}(\boldsymbol{\Sigma} + \boldsymbol{\delta}_{\boldsymbol{\Sigma}})) + \boldsymbol{\delta}_{\boldsymbol{\mu}}^{\top} \boldsymbol{\Sigma}^{-1} \boldsymbol{\delta}_{\boldsymbol{\mu}} + \log \frac{|\boldsymbol{\Sigma}|}{|(\boldsymbol{\Sigma} + \boldsymbol{\delta}_{\boldsymbol{\Sigma}})|} - d \right] \tag{34}$$

$$= \frac{1}{2} \left[ \mathrm{tr}(I + \frac{2}{\lambda} \nabla_{\boldsymbol{\Sigma}} J(\boldsymbol{\theta})(\boldsymbol{\Sigma} + \boldsymbol{\delta}_{\boldsymbol{\Sigma}})) + \boldsymbol{\delta}_{\boldsymbol{\mu}}^{\top} \boldsymbol{\Sigma}^{-1} \boldsymbol{\delta}_{\boldsymbol{\mu}} + \log \frac{|\boldsymbol{\Sigma}|}{|(\boldsymbol{\Sigma} + \boldsymbol{\delta}_{\boldsymbol{\Sigma}})|} - d \right] \tag{35}$$

$$= \frac{1}{2} \left[ \mathrm{tr}(\frac{2}{\lambda} \nabla_{\boldsymbol{\Sigma}} J(\boldsymbol{\theta}) \boldsymbol{\delta}_{\boldsymbol{\Sigma}}) + \boldsymbol{\delta}_{\boldsymbol{\mu}}^{\top} \boldsymbol{\Sigma}^{-1} \boldsymbol{\delta}_{\boldsymbol{\mu}} + Q \right], \tag{36}$$

where

$$Q = |\frac{2}{\lambda} \boldsymbol{\Sigma} \nabla_{\boldsymbol{\Sigma}} J(\boldsymbol{\theta})| + \log(I - \frac{2}{\lambda} \boldsymbol{\Sigma} \nabla_{\boldsymbol{\Sigma}} J(\boldsymbol{\theta})). \tag{37}$$

Since $\boldsymbol{\Sigma}$ and $\nabla_{\boldsymbol{\Sigma}} J(\boldsymbol{\theta})$ are both diagonal matrix, we denote $\mathrm{diag}(\boldsymbol{\Sigma} \nabla_{\boldsymbol{\Sigma}} J(\boldsymbol{\theta})) = (v^1, \dots, v^d)$, then we have

$$Q = \log(\prod_{i=1}^{d}(1 - \frac{2}{\lambda} v^i)) + \sum_{i=1}^{d} \frac{2}{\lambda} v^i = \sum_{i=1}^{d} \left( \log(1 - \frac{2}{\lambda} v^i) + \frac{2}{\lambda} v^i \right) = \sum_{i=1}^{d} -\frac{2}{\lambda^2}(v^i)^2 - \mathcal{O}(\frac{1}{\lambda^3}(v^i)^3). \tag{38}$$

We denote $\mathrm{diag}(\nabla_{\boldsymbol{\Sigma}} J(\boldsymbol{\theta})) = (\hat{v}^1, \dots, \hat{v}^d)$, then we have $\hat{v}^i \sigma^i = v^i$ and

$$\mathrm{tr}(\frac{2}{\lambda} \nabla_{\boldsymbol{\Sigma}} J(\boldsymbol{\theta}) \boldsymbol{\delta}_{\boldsymbol{\Sigma}}) = \sum_{i=1}^{d} \frac{2}{\lambda} \hat{v}^i (\frac{1}{(\sigma^i)^{-1} - \frac{2}{\lambda} \hat{v}^i} - \sigma^i) = \sum_{i=1}^{d} \frac{2}{\lambda} \hat{v}^i \sigma^i (\frac{2}{\lambda} \sigma^i \hat{v}^i) + \mathcal{O}(\frac{4}{\lambda^2}(\sigma^i \hat{v}^i)^2). \tag{39}$$

Then substituting $\boldsymbol{\delta}_{\boldsymbol{\mu}}(\boldsymbol{\theta})$ and $\boldsymbol{\delta}_{\boldsymbol{\Sigma}}(\boldsymbol{\theta})$ into the inequality $\mathrm{KL}(p_{\boldsymbol{\theta}+\boldsymbol{\delta}} \| p_{\boldsymbol{\theta}_t}) \le \rho^2$, we can obtain that

$$\mathrm{KL}(p_{\boldsymbol{\theta}+\boldsymbol{\delta}} \| p_{\boldsymbol{\theta}}) = \frac{1}{2} \left[ \frac{2}{\lambda^2} \| \boldsymbol{\Sigma} \nabla_{\boldsymbol{\Sigma}} J(\boldsymbol{\theta}) \|_{\mathrm{F}}^2 + \frac{1}{\lambda^2} \| \boldsymbol{\Sigma}^{\frac{1}{2}} \nabla_{\boldsymbol{\mu}} J(\boldsymbol{\theta}) \|_2^2 \right] + \epsilon \le \rho^2, \tag{40}$$

where $\epsilon = \mathcal{O}(\frac{4(\sigma^i \hat{v}^i)^2}{\lambda^2}) = \mathcal{O}(\frac{1}{\lambda^2})$. Let the equality holds and solve Eq. (40), we have

$$\lambda \approx \frac{1}{\rho} \sqrt{\| \boldsymbol{\Sigma} \nabla_{\boldsymbol{\Sigma}} J(\boldsymbol{\theta}) \|_{\mathrm{F}}^2 + 0.5 \| \boldsymbol{\Sigma}^{\frac{1}{2}} \nabla_{\boldsymbol{\mu}} J(\boldsymbol{\theta}) \|_2^2}. \tag{41}$$

### A.3 UPDATE RULE FOR $\theta_t$

Note that we have

$$\langle \boldsymbol{\theta} - \boldsymbol{\theta}_t, \nabla_{\boldsymbol{\theta}} J(\boldsymbol{\theta}_t + \boldsymbol{\delta}_t) \rangle + \frac{1}{\beta_t} \mathrm{KL}(p_{\boldsymbol{\theta}} \| p_{\boldsymbol{\theta}_t})$$
$$= (\boldsymbol{\mu} - \boldsymbol{\mu}_t)^\top \nabla_{\boldsymbol{\mu}} J(\boldsymbol{\theta}_t + \boldsymbol{\delta}_t) + \mathrm{tr}((\boldsymbol{\Sigma} - \boldsymbol{\Sigma}_t) \nabla_{\boldsymbol{\Sigma}} J(\boldsymbol{\theta}_t + \boldsymbol{\delta}_t))$$
$$+ \frac{1}{2\beta_t} \left[ \mathrm{tr}(\boldsymbol{\Sigma}_t^{-1} \boldsymbol{\Sigma}) + (\boldsymbol{\mu} - \boldsymbol{\mu}_t)^\top \boldsymbol{\Sigma}_t^{-1} (\boldsymbol{\mu} - \boldsymbol{\mu}_t) + \log \frac{|\boldsymbol{\Sigma}_t|}{|\boldsymbol{\Sigma}|} - d \right],$$

where $\nabla_{\boldsymbol{\mu}} J(\boldsymbol{\theta}_t + \boldsymbol{\delta}_t)$ and $\nabla_{\boldsymbol{\Sigma}} J(\boldsymbol{\theta}_t + \boldsymbol{\delta}_t)$ denotes the derivative w.r.t. $\boldsymbol{\mu}$ and $\boldsymbol{\Sigma}$ taking at $\boldsymbol{\mu} = \boldsymbol{\mu}_t + \boldsymbol{\delta}_{\boldsymbol{\mu}_t}$ and $\boldsymbol{\Sigma} = \boldsymbol{\Sigma}_t + \boldsymbol{\delta}_{\boldsymbol{\Sigma}_t}$, respectively. We can see the above problem is convex with respect to $\boldsymbol{\mu}$ and $\boldsymbol{\Sigma}$. Taking the derivative w.r.t. $\boldsymbol{\mu}$ and $\boldsymbol{\Sigma}$ and setting them to zero, we can obtain that

$$\nabla_{\boldsymbol{\mu}} J(\boldsymbol{\theta}_t + \boldsymbol{\delta}_t) + \frac{1}{\beta_t} \boldsymbol{\Sigma}_t^{-1} (\boldsymbol{\mu} - \boldsymbol{\mu}_t) = 0, \tag{42}$$

$$\nabla_{\boldsymbol{\Sigma}} J(\boldsymbol{\theta}_t + \boldsymbol{\delta}_t) + \frac{1}{2\beta_t} [\boldsymbol{\Sigma}_t^{-1} - \boldsymbol{\Sigma}^{-1}] = 0. \tag{43}$$

Therefore, we obtain the update rule for $\boldsymbol{\theta}_t$ as

$$\boldsymbol{\mu}_{t+1} = \boldsymbol{\mu}_t - \beta_t \boldsymbol{\Sigma}_t \nabla_{\boldsymbol{\mu}} J(\boldsymbol{\theta}_t + \boldsymbol{\delta}_t), \tag{44}$$
$$\boldsymbol{\Sigma}_{t+1}^{-1} = \boldsymbol{\Sigma}_t^{-1} + 2\beta_t \nabla_{\boldsymbol{\Sigma}} J(\boldsymbol{\theta}_t + \boldsymbol{\delta}_t). \tag{45}$$

### A.4 THE GRADIENTS OF $J(\theta)$

To obtain the gradients of the reparameterized objective $J(\boldsymbol{\theta})$ w.r.t. $\boldsymbol{\mu}$ and $\boldsymbol{\Sigma}$, we use the following theorem to show that only function queries are needed.

**Theorem A.1** *(Wierstra et al., 2014) The gradient of the expectation of an integrable function $F(\boldsymbol{x})$ under a Gaussian distribution $p_{\boldsymbol{\theta}} := \mathcal{N}(\boldsymbol{\mu}, \boldsymbol{\Sigma})$ with respect to the mean $\boldsymbol{\mu}$ and the covariance $\boldsymbol{\Sigma}$ can be expressed as*

$$\nabla_{\boldsymbol{\mu}} J(\boldsymbol{\theta}) = \mathbb{E}_{p_{\boldsymbol{\theta}}} [\boldsymbol{\Sigma}^{-1} (\boldsymbol{x} - \boldsymbol{\mu}) F(\boldsymbol{x})], \tag{46}$$

$$\nabla_{\boldsymbol{\Sigma}} J(\boldsymbol{\theta}) = \frac{1}{2} \mathbb{E}_{p_{\boldsymbol{\theta}}} [(\boldsymbol{\Sigma}^{-1} (\boldsymbol{x} - \boldsymbol{\mu})(\boldsymbol{x} - \boldsymbol{\mu})^\top \boldsymbol{\Sigma}^{-1} - \boldsymbol{\Sigma}^{-1}) F(\boldsymbol{x})]. \tag{47}$$

## B MINI-BATCH SABO

For the proposed SABO with a mini-batch function query, the stochastic approximation of the gradients $\nabla_{\boldsymbol{\mu}} J(\boldsymbol{\theta}_t)$ and $\nabla_{\boldsymbol{\Sigma}} J(\boldsymbol{\theta}_t)$ using Monte Carlo sampling are given as

$$\boldsymbol{g}_t' = \frac{1}{NM} \sum_{j=1}^N \sum_{i=1}^M \boldsymbol{\Sigma}_t^{-1} (\boldsymbol{x}_j' - \boldsymbol{\mu}_t) \big( F(\boldsymbol{x}_j'; (X_i', y_i')) - F(\boldsymbol{\mu}_t; (X_i', y_i')) \big), \tag{48}$$

$$\boldsymbol{G}_t' = \frac{1}{2NM} \sum_{j=1}^N \sum_{i=1}^M \mathrm{diag} \Big[ \boldsymbol{\Sigma}_t^{-1} \big[ \mathrm{diag} \big( (\boldsymbol{x}_j' - \boldsymbol{\mu}_t)(\boldsymbol{x}_j' - \boldsymbol{\mu}_t)^\top \boldsymbol{\Sigma}_t^{-1} - \boldsymbol{I} \big)$$
$$\times (F(\boldsymbol{x}_j'; (X_i', y_i')) - F(\boldsymbol{\mu}_t; (X_i', y_i'))) \big] \Big], \tag{49}$$

where $\boldsymbol{x}_j'$ denotes the $j$-th sample sampled from the distribution $\mathcal{N}(\boldsymbol{x} \mid \boldsymbol{\mu}_t, \boldsymbol{\Sigma}_t)$, and $(X_i', y_i')$ denotes the $i$-th data in the mini-batch dataset $\mathcal{B}'$. Then by setting $\boldsymbol{\delta}_t = \{ \frac{1}{\lambda} \boldsymbol{\Sigma}_t \boldsymbol{g}_t', \frac{2\boldsymbol{\Sigma}_t \boldsymbol{G}_t'}{\lambda \boldsymbol{\Sigma}_t^{-1} - 2\boldsymbol{G}_t'} \}$, the gradients

---

**Algorithm 2** Mini-batch SABO

---

**Require:** Neighborhood size $\rho$, learning rate $\beta_t$, batch size $M$
1: Initialized $\boldsymbol{\theta}_0 = (\boldsymbol{\mu}_0, \boldsymbol{\Sigma}_0)$ ;
2: **for** $t = 0$ to $T - 1$ **do**
3:     Take i.i.d. samples $\boldsymbol{z}_j' \sim \mathcal{N}(0, I)$ and set $\boldsymbol{x}_j' = \boldsymbol{\mu}_t + \boldsymbol{\Sigma}_t^{\frac{1}{2}} \boldsymbol{z}_j'$ for $j \in \{1, \ldots, N\}$;
4:     Sample a batch of data $\mathcal{B}'$;
5:     Query the batch observations $\{F(\boldsymbol{x}_1'; \mathcal{B}'), \ldots, F(\boldsymbol{x}_N'; \mathcal{B}')\}$;
6:     Set the gradient $\boldsymbol{g}_t'$ via Eq. (48) and set the gradient $\boldsymbol{G}_t'$ via Eq. (49);
7:     Compute $\lambda = \frac{1}{\rho}\sqrt{\|\boldsymbol{\Sigma}_t \boldsymbol{G}_t'\|_{\mathrm{F}}^2 + 0.5\|\boldsymbol{\Sigma}_t^{\frac{1}{2}} \boldsymbol{g}_t'\|_2^2}$;
8:     Compute $\boldsymbol{\delta}_{\boldsymbol{\mu}_t} = \frac{1}{\lambda} \boldsymbol{\Sigma}_t \boldsymbol{g}_t'$ and $\boldsymbol{\delta}_{\boldsymbol{\Sigma}_t} = \frac{2\boldsymbol{\Sigma}_t \boldsymbol{G}_t'}{\lambda \boldsymbol{\Sigma}_t^{-1} - 2\boldsymbol{G}_t'}$;
9:     Take i.i.d. samples $\boldsymbol{z}_j \sim \mathcal{N}(0, I)$ for $j \in \{1, \ldots, N\}$;
10:     Set $\boldsymbol{x}_j = \boldsymbol{\mu}_t + \boldsymbol{\delta}_{\boldsymbol{\mu}_t} + (\boldsymbol{\Sigma}_t + \boldsymbol{\delta}_{\boldsymbol{\Sigma}_t})^{\frac{1}{2}} \boldsymbol{z}_j$ for $j \in \{1, \ldots, N\}$;
11:     Sample a batch of data $\mathcal{B}$;
12:     Query the batch observations $\{F(\boldsymbol{x}_1; \mathcal{B}), \ldots, F(\boldsymbol{x}_N; \mathcal{B})\}$;
13:     Set the gradient $\boldsymbol{g}_t$ via Eq. (50) and set the gradient $\boldsymbol{G}_t$ via Eq. (51);
14:     Set $\boldsymbol{\mu}_{t+1} = \boldsymbol{\mu}_t - \beta_t \boldsymbol{\Sigma}_t \boldsymbol{g}_t$;
15:     Set $\boldsymbol{\Sigma}_{t+1}^{-1} = \boldsymbol{\Sigma}_t^{-1} + 2\beta_t \boldsymbol{G}_t$;
16: **end for**
17: **return** $\boldsymbol{\theta}_T = (\boldsymbol{\mu}_T, \boldsymbol{\Sigma}_T)$.

---

$\nabla_{\boldsymbol{\mu}} J(\boldsymbol{\theta}_t + \boldsymbol{\delta}_t)$ and $\nabla_{\boldsymbol{\Sigma}} J(\boldsymbol{\theta}_t + \boldsymbol{\delta}_t)$ can also be approximated as

$$\boldsymbol{g}_t = \frac{1}{NM} \sum_{j=1}^{N} \sum_{i=1}^{M} \widehat{\boldsymbol{\Sigma}}_t^{-1} (\boldsymbol{x}_j - \widehat{\boldsymbol{\mu}}_t) \big(F(\boldsymbol{x}_j; (X_i, y_i)) - F(\widehat{\boldsymbol{\mu}}_t; (X_i, y_i))\big), \tag{50}$$

$$\begin{aligned}
\boldsymbol{G}_t = \frac{1}{2NM} \sum_{j=1}^{N} \sum_{i=1}^{M} \mathrm{diag}\Big[ \widehat{\boldsymbol{\Sigma}}_t^{-1} \big[ \mathrm{diag}\big( (\boldsymbol{x}_j - \widehat{\boldsymbol{\mu}}_t)(\boldsymbol{x}_j - \widehat{\boldsymbol{\mu}}_t)^{\top} \widehat{\boldsymbol{\Sigma}}_t^{-1} - \boldsymbol{I} \big) \\
\times \big(F(\boldsymbol{x}_j; (X_i, y_i)) - F(\widehat{\boldsymbol{\mu}}_t; (X_i, y_i))\big) \big] \Big],
\end{aligned} \tag{51}$$

where $\widehat{\boldsymbol{\Sigma}}_t = \boldsymbol{\Sigma}_t + \boldsymbol{\delta}_{\boldsymbol{\Sigma}_t}$, $\widehat{\boldsymbol{\mu}}_t = \boldsymbol{\mu}_t + \boldsymbol{\delta}_{\boldsymbol{\mu}_t}$, $\boldsymbol{x}_j$ denotes the $j$-th sample sampled from the distribution $\mathcal{N}(\boldsymbol{x} \mid \widehat{\boldsymbol{\mu}}_t, \widehat{\boldsymbol{\Sigma}}_t)$, and $(X_i, y_i)$ denotes the $i$-th data in the mini-batch dataset $\mathcal{B}$. Then we can update $\boldsymbol{\theta}_t$ by Eq. (21). The entire algorithm of SABO with a mini-batch function query is shown in Algorithm 2.

## C TECHNICAL LEMMAS

In this section, we introduce the following technical lemmas for analysis. The proof of all technical lemmas is put in Appendix E.

**Lemma C.1** *Suppose $\boldsymbol{\Sigma}$ and $\hat{\boldsymbol{\Sigma}}$ are two $d$-dimensional diagonal matrix and $\boldsymbol{z}$ is a $d$-dimensional vector, then we have $\|\boldsymbol{\Sigma} \boldsymbol{z}\| \leq \|\boldsymbol{\Sigma}\|_{\mathrm{F}} \|\boldsymbol{z}\|$ and $\|\boldsymbol{\Sigma} \hat{\boldsymbol{\Sigma}}\|_{\mathrm{F}} \leq \|\boldsymbol{\Sigma}\|_{\mathrm{F}} \|\hat{\boldsymbol{\Sigma}}\|_{\mathrm{F}}$.*

**Lemma C.2** *Given a convex function $F(\boldsymbol{x})$, for Gaussian distribution with parameters $\boldsymbol{\theta} := \{\boldsymbol{\mu}, \boldsymbol{\Sigma}^{\frac{1}{2}}\}$, let $J(\boldsymbol{\theta}) := \mathbb{E}_{p(\boldsymbol{x}; \boldsymbol{\theta})}[F(\boldsymbol{x})]$. Then $J(\boldsymbol{\theta})$ is a convex function with respect to $\boldsymbol{\theta}$.*

**Lemma C.3** *Suppose that the gradient $\boldsymbol{G}_t$ are positive semi-definite matrix and satisfies $\xi \boldsymbol{I} \preceq \boldsymbol{G}_t \preceq b\boldsymbol{I}$. Then for algorithm 1 and 2, we have the following results.*

(a) *The (diagonal) covariance matrix $\boldsymbol{\Sigma}_T$ satisfies $\frac{1}{2b \sum_{t=1}^{T} \beta_t \boldsymbol{I} + \boldsymbol{\Sigma}_0^{-1}} \preceq \boldsymbol{\Sigma}_T \preceq \frac{1}{2\xi \sum_{t=1}^{T} \beta_t \boldsymbol{I} + \boldsymbol{\Sigma}_0^{-1}}$.*

(b) *The Frobenius norm for the covariance matrix $\boldsymbol{\Sigma}_t$ satisfies $\|\boldsymbol{\Sigma}_t\|_{\mathrm{F}} \leq \frac{\sqrt{d}}{2\xi \sum_{k=1}^{t} \beta_k}$.*

**Lemma C.4** *For given $\boldsymbol{\theta}_t$, denote the approximated gradients of $\nabla_{\boldsymbol{\mu}} J(\boldsymbol{\theta}_t)$ and $\nabla_{\boldsymbol{\Sigma}} J(\boldsymbol{\theta}_t)$ by $\boldsymbol{g}'_t$ and $\boldsymbol{G}'_t$, respectively. Then for Algorithm 1 and Algorithm 2, if $\rho \leq \frac{\sqrt{d}}{2}$, then the perturbation satisfies, $\|\boldsymbol{\delta}_{\boldsymbol{\mu}_t}\| \leq \rho\sqrt{2}\|\boldsymbol{\Sigma}_t^{\frac{1}{2}}\|_{\mathrm{F}}$ and $\|\boldsymbol{\delta}_{\boldsymbol{\Sigma}_t}\|_{\mathrm{F}} \leq 2\rho r\|\boldsymbol{\Sigma}_t\|_{\mathrm{F}}$, where $r$ is a positive constant.*

**Lemma C.5** *In Algorithm 1, suppose the gradient estimator $\boldsymbol{g}'_t$ in $t$-th iteration as*

$$\boldsymbol{g}'_t = \boldsymbol{\Sigma}_t^{-\frac{1}{2}}\boldsymbol{z}\big(F(\boldsymbol{\mu}_t + \boldsymbol{\Sigma}_t^{\frac{1}{2}}\boldsymbol{z}) - F(\boldsymbol{\mu}_t)\big), \tag{52}$$

*where $\boldsymbol{z} \sim \mathcal{N}(0, I)$. Then $\boldsymbol{g}'_t$ is an unbiased estimator of the gradient $\nabla_{\boldsymbol{\mu}}\mathbb{E}_{p_{\boldsymbol{\theta}_t}}[F(\boldsymbol{x})]$.*

**Lemma C.6** *In Algorithm 1, suppose the gradient estimator $\boldsymbol{g}_t$ in $t$-th iteration as*

$$\boldsymbol{g}_t = \widehat{\boldsymbol{\Sigma}}_t^{-\frac{1}{2}}\boldsymbol{z}\big(F(\widehat{\boldsymbol{\mu}}_t + \widehat{\boldsymbol{\Sigma}}_t^{\frac{1}{2}}\boldsymbol{z}) - F(\widehat{\boldsymbol{\mu}}_t)\big), \tag{53}$$

*where $\boldsymbol{z} \sim \mathcal{N}(0, I)$. Suppose that Assumption 4.1 holds, $\rho < \frac{\sqrt{d}}{2}$, the gradient $\boldsymbol{G}_t$ is positive semi-definite matrix and satisfies $\xi\boldsymbol{I} \preceq \boldsymbol{G}_t \preceq b\boldsymbol{I}$ and $\boldsymbol{\Sigma}_0 \preceq R\boldsymbol{I}$, where $\xi, b, R \geq 0$. Then we have*

$$\mathbb{E}[\|\boldsymbol{\Sigma}_t^{\frac{1}{2}}\boldsymbol{g}_t\|^2] \leq \frac{L_F^2(1 + 2\rho r')(d+4)^2}{2\xi(\sum_{k=1}^t \beta_k)}, \tag{54}$$

*where $r'$ is a positive constant.*

**Lemma C.7** *In Algorithm 2, suppose assumption 4.4 holds and the gradient estimator $\boldsymbol{g}'_t$ in $t$-th iteration as*

$$\boldsymbol{g}'_t = \boldsymbol{\Sigma}_t^{-\frac{1}{2}}\boldsymbol{z}\big(F(\boldsymbol{\mu}_t + \boldsymbol{\Sigma}_t^{\frac{1}{2}}\boldsymbol{z}; \mathcal{B}) - F(\boldsymbol{\mu}_t; \mathcal{B})\big), \tag{55}$$

*where $\boldsymbol{z} \sim \mathcal{N}(0, I)$. Then $\boldsymbol{g}'_t$ is an unbiased estimator of the gradient $\nabla_{\boldsymbol{\mu}}\mathbb{E}_{p_{\boldsymbol{\theta}_t}}[F(\boldsymbol{x})]$.*

**Lemma C.8** *In Algorithm 2, suppose the gradient estimator $\boldsymbol{g}_t$ in $t$-th iteration as*

$$\boldsymbol{g}_t = \widehat{\boldsymbol{\Sigma}}_t^{-\frac{1}{2}}\boldsymbol{z}\big(F(\widehat{\boldsymbol{\mu}}_t + \widehat{\boldsymbol{\Sigma}}_t^{\frac{1}{2}}\boldsymbol{z}; \mathcal{B}) - F(\widehat{\boldsymbol{\mu}}_t; \mathcal{B})\big), \tag{56}$$

*where $\boldsymbol{z} \sim \mathcal{N}(0, I)$. Suppose that Assumption 4.1 and Assumption 4.4 hold, $\rho < \frac{\sqrt{d}}{2}$, the gradient $\boldsymbol{G}_t$ is positive semi-definite matrix and satisfies $\xi\boldsymbol{I} \preceq \boldsymbol{G}_t \preceq b\boldsymbol{I}$ and $\boldsymbol{\Sigma}_0 \preceq R\boldsymbol{I}$, where $\xi, b, R \geq 0$. Then we have*

$$\mathbb{E}[\|\boldsymbol{\Sigma}_t^{\frac{1}{2}}\boldsymbol{g}_t\|^2] \leq \frac{L_F^2(1 + 2\rho r')(d+4)^2}{6\xi(\sum_{k=1}^t \beta_k)} + \frac{2(d+4)\varepsilon^2}{3}, \tag{57}$$

*where $r'$ is a positive constant, $\varepsilon^2 = 2\varepsilon_{\mathrm{B}}^2 + 2\varepsilon_{\mathrm{D}}^2$.*

## D PROOF OF THE RESULT IN SECTION 4

In this section, we provide the proof of the result in Section 4.

### D.1 PROOF OF THE PROPOSITION 4.2

Note that $F(\boldsymbol{x})$ is a convex function, we have

$$F(\boldsymbol{\mu}) = F(\mathbb{E}_{\boldsymbol{x} \sim \mathcal{N}(\boldsymbol{\mu}, \boldsymbol{\Sigma})}[\boldsymbol{x}]) \leq \mathbb{E}_{\boldsymbol{x} \sim \mathcal{N}(\boldsymbol{\mu}, \boldsymbol{\Sigma})}[F(\boldsymbol{x})] = J(\boldsymbol{\mu}, \boldsymbol{\Sigma}). \tag{58}$$

Since $F(\boldsymbol{\mu}^*) = J(\boldsymbol{\mu}^*, \mathbf{0})$, it follows that

$$F(\boldsymbol{\mu}) - F(\boldsymbol{\mu}^*) \leq J(\boldsymbol{\mu}, \boldsymbol{\Sigma}) - J(\boldsymbol{\mu}^*, \mathbf{0}), \tag{59}$$

where we reach the conclusion.

## D.2 PROOF OF THEOREM 4.3

The update rule of $\boldsymbol{\mu}$ can be represented as $\boldsymbol{\mu}_{t+1} = \boldsymbol{\mu}_t - \beta_t \boldsymbol{\Sigma}_t \boldsymbol{g}_t$. Since $F(\boldsymbol{x})$ is convex function, we have $J(\boldsymbol{\theta})$ is convex w.r.t. $\boldsymbol{\theta} = \{\boldsymbol{\mu}, \boldsymbol{\Sigma}^{\frac{1}{2}}\}$ by Lemma C.2, together with $J(\boldsymbol{\theta})$ is $c$-strongly convex w.r.t. $\boldsymbol{\mu}$ we obtain

$$J(\boldsymbol{\theta}_t) \leq J(\boldsymbol{\mu}^*, 0) + \nabla_{\boldsymbol{\mu}} J(\boldsymbol{\theta}_t)^\top (\boldsymbol{\mu}_t - \boldsymbol{\mu}^*) + \nabla_{\boldsymbol{\Sigma}^{\frac{1}{2}}} J(\boldsymbol{\theta}_t)^\top \boldsymbol{\Sigma}_t^{\frac{1}{2}} - \frac{c}{2} \|\boldsymbol{\mu}_t - \boldsymbol{\mu}^*\|^2. \tag{60}$$

Note that $\nabla_{\boldsymbol{\Sigma}^{\frac{1}{2}}} J(\boldsymbol{\theta}_t) = \boldsymbol{\Sigma}_t^{\frac{1}{2}} \nabla_{\boldsymbol{\Sigma}} J(\boldsymbol{\theta}_t) + \nabla_{\boldsymbol{\Sigma}} J(\boldsymbol{\theta}_t) \boldsymbol{\Sigma}_t^{\frac{1}{2}}$, we have

$$J(\boldsymbol{\theta}_t) \leq J(\boldsymbol{\mu}^*, 0) + \nabla_{\boldsymbol{\mu}} J(\boldsymbol{\theta}_t)^\top (\boldsymbol{\mu}_t - \boldsymbol{\mu}^*) + 2\nabla_{\boldsymbol{\Sigma}} J(\boldsymbol{\theta}_t) \boldsymbol{\Sigma}_t - \frac{c}{2} \|\boldsymbol{\mu}_t - \boldsymbol{\mu}^*\|^2. \tag{61}$$

Let $A_t = J(\boldsymbol{\mu}_t, \boldsymbol{\Sigma}_t) - J(\boldsymbol{\mu}^*, 0)$, we have

$$\beta_t \mathbb{E}[A_t] \leq \beta_t \mathbb{E}_{\boldsymbol{z}}[\nabla_{\boldsymbol{\mu}} J(\boldsymbol{\theta}_t)^\top (\boldsymbol{\mu}_t - \boldsymbol{\mu}^*)] + 2\beta_t \mathbb{E}_{\boldsymbol{z}}[\nabla_{\boldsymbol{\Sigma}} J(\boldsymbol{\theta}_t)^\top \boldsymbol{\Sigma}_t] - \frac{c\beta_t}{2} \|\boldsymbol{\mu}_t - \boldsymbol{\mu}^*\|^2. \tag{62}$$

Note that

$$\|\boldsymbol{\mu}_t - \boldsymbol{\mu}^*\|_{\boldsymbol{\Sigma}_t^{-1}}^2 - \|\boldsymbol{\mu}_{t+1} - \boldsymbol{\mu}^*\|_{\boldsymbol{\Sigma}_t^{-1}}^2$$

$$= \|\boldsymbol{\mu}_t - \boldsymbol{\mu}^*\|_{\boldsymbol{\Sigma}_t^{-1}}^2 - \|\boldsymbol{\mu}_t - \beta_t \boldsymbol{\Sigma}_t \boldsymbol{g}_t - \boldsymbol{\mu}^*\|_{\boldsymbol{\Sigma}_t^{-1}}^2 \tag{63}$$

$$= \|\boldsymbol{\mu}_t - \boldsymbol{\mu}^*\|_{\boldsymbol{\Sigma}_t^{-1}}^2 - \left( \|\boldsymbol{\mu}_t - \boldsymbol{\mu}^*\|_{\boldsymbol{\Sigma}_t^{-1}}^2 - 2\beta_t \langle \boldsymbol{\mu}_t - \boldsymbol{\mu}^*, \boldsymbol{g}_t \rangle + \beta_t^2 \langle \boldsymbol{\Sigma}_t \boldsymbol{g}_t, \boldsymbol{g}_t \rangle \right) \tag{64}$$

$$= 2\beta_t \boldsymbol{g}_t^\top (\boldsymbol{\mu}_t - \boldsymbol{\mu}^*) - \beta_t^2 (\boldsymbol{\Sigma}_t \boldsymbol{g}_t)^\top \boldsymbol{g}_t. \tag{65}$$

Therefore we have

$$\boldsymbol{g}_t^\top (\boldsymbol{\mu}_t - \boldsymbol{\mu}^*) = \frac{1}{2\beta_t} \left( \|\boldsymbol{\mu}_t - \boldsymbol{\mu}^*\|_{\boldsymbol{\Sigma}_t^{-1}}^2 - \|\boldsymbol{\mu}_{t+1} - \boldsymbol{\mu}^*\|_{\boldsymbol{\Sigma}_t^{-1}}^2 \right) + \frac{\beta_t}{2} (\boldsymbol{\Sigma}_t \boldsymbol{g}_t)^\top \boldsymbol{g}_t. \tag{66}$$

According to Lemma C.5, we have $\mathbb{E}\boldsymbol{g}_t = \nabla_{\boldsymbol{\mu}} J(\boldsymbol{\theta}_t + \boldsymbol{\delta}_t)$. Then we have

$$\mathbb{E}\left[ (\nabla_{\boldsymbol{\mu}} J(\boldsymbol{\theta}_t) - \boldsymbol{g}_t)^\top (\boldsymbol{\mu}_t - \boldsymbol{\mu}^*) \right]$$

$$= \mathbb{E}\left[ (\nabla_{\boldsymbol{\mu}} J(\boldsymbol{\theta}_t) - \nabla_{\boldsymbol{\mu}} J(\boldsymbol{\theta}_t + \boldsymbol{\delta}_t))^\top (\boldsymbol{\mu}_t - \boldsymbol{\mu}^*) \right] + \mathbb{E}\left[ (\nabla_{\boldsymbol{\mu}} J(\boldsymbol{\theta}_t + \boldsymbol{\delta}_t) - \boldsymbol{g}_t)^\top (\boldsymbol{\mu}_t - \boldsymbol{\mu}^*) \right] \tag{67}$$

$$\leq \|\boldsymbol{\mu}_t - \boldsymbol{\mu}^*\| \|\nabla_{\boldsymbol{\mu}} J(\boldsymbol{\theta}_t) - \nabla_{\boldsymbol{\mu}} J(\boldsymbol{\theta}_t + \boldsymbol{\delta}_t)\| \tag{68}$$

$$\leq DL\|\boldsymbol{\delta}_t\| \tag{69}$$

$$\leq DL\rho_t U_t, \tag{70}$$

where the first inequality is due to the Cauchy-Schwarz inequality, the second inequality is due to $\|\boldsymbol{\mu}_t - \boldsymbol{\mu}^*\| \leq D$ and Assumption 4.1, the last inequality is due to Lemma C.4, and $U_t = \max(2\sqrt{2}\|\boldsymbol{\Sigma}_t^{\frac{1}{2}}\|_{\mathrm{F}}, 4r\|\boldsymbol{\Sigma}_t\|_{\mathrm{F}})$. Then we have

$$\mathbb{E}_{\boldsymbol{z}}\left[ \nabla_{\boldsymbol{\mu}} J(\boldsymbol{\theta}_t)^\top (\boldsymbol{\mu}_t - \boldsymbol{\mu}^*) \right] \tag{71}$$

$$= \mathbb{E}_{\boldsymbol{z}}\left[ \boldsymbol{g}_t^\top (\boldsymbol{\mu}_t - \boldsymbol{\mu}^*) + (\nabla_{\boldsymbol{\mu}} J(\boldsymbol{\theta}_t) - \boldsymbol{g}_t)^\top (\boldsymbol{\mu}_t - \boldsymbol{\mu}^*) \right] \tag{72}$$

$$\leq \frac{1}{2\beta_t} \mathbb{E}_{\boldsymbol{z}}[\|\boldsymbol{\mu}_t - \boldsymbol{\mu}^*\|_{\boldsymbol{\Sigma}_t^{-1}}^2 - \|\boldsymbol{\mu}_{t+1} - \boldsymbol{\mu}^*\|_{\boldsymbol{\Sigma}_t^{-1}}^2] + \beta_t (\boldsymbol{\Sigma}_t \boldsymbol{g}_t)^\top \boldsymbol{g}_t] + DL\rho_t U_t, \tag{73}$$

where the inequality is due to Eq. (66), Eq. (70), and $\beta_t \geq 0$. Note that

$$\mathbb{E}\left[ \nabla_{\boldsymbol{\Sigma}} J(\boldsymbol{\theta}_t)^\top \boldsymbol{\Sigma}_t \right] \leq \mathbb{E}\left[ \|\nabla_{\boldsymbol{\Sigma}} J(\boldsymbol{\theta}_t)\| \right] \|\boldsymbol{\Sigma}_t\|_{\mathrm{F}} \leq H\|\boldsymbol{\Sigma}_t\|_{\mathrm{F}}, \tag{74}$$

where the first inequality is due to Lemma C.1 and the second inequality is due to the Lipschitz continuous assumption of the function $J(\boldsymbol{\theta})$. Then substituting Eq. (73) and Eq. (74) into Eq. (62) and multiplying $\beta_t$ on both sides of the inequality, we have

$$\beta_t \mathbb{E}[A_t] \leq \frac{1}{2} \mathbb{E}[\|\boldsymbol{\mu}_t - \boldsymbol{\mu}^*\|_{\boldsymbol{\Sigma}_t^{-1}}^2 - \|\boldsymbol{\mu}_{t+1} - \boldsymbol{\mu}^*\|_{\boldsymbol{\Sigma}_t^{-1}}^2] - \frac{c\beta_t}{2} \|\boldsymbol{\mu}_t - \boldsymbol{\mu}^*\|^2 + 2\beta_t^2 H\|\boldsymbol{\Sigma}_t\|_{\mathrm{F}}$$

$$+ DL\rho_t U_t + \beta_t^2 \|\boldsymbol{\Sigma}_t^{\frac{1}{2}}\|_{\mathrm{F}} \mathbb{E}\|\boldsymbol{\Sigma}_t^{\frac{1}{2}} \boldsymbol{g}_t\|^2, \tag{75}$$

We further obtain that

$$\sum_{t=0}^{T-1}\left[\frac{1}{2}\mathbb{E}[\|\boldsymbol{\mu}_t-\boldsymbol{\mu}^*\|_{\boldsymbol{\Sigma}_t^{-1}}^2-\|\boldsymbol{\mu}_{t+1}-\boldsymbol{\mu}^*\|_{\boldsymbol{\Sigma}_t^{-1}}^2]-\frac{c\beta_t}{2}\|\boldsymbol{\mu}_t-\boldsymbol{\mu}^*\|^2\right] \tag{76}$$

$$\leq \frac{1}{2}\sum_{t=0}^{T-1}\left[\|\boldsymbol{\mu}_t-\boldsymbol{\mu}^*\|_{\boldsymbol{\Sigma}_t^{-1}}^2-\|\boldsymbol{\mu}_{t-1}-\boldsymbol{\mu}^*\|_{\boldsymbol{\Sigma}_{t-1}^{-1}}^2-\frac{c\beta_t}{2}\|\boldsymbol{\mu}_t-\boldsymbol{\mu}^*\|^2\right] \tag{77}$$

$$+\frac{1}{2}\left[\|\boldsymbol{\mu}_0-\boldsymbol{\mu}^*\|_{\boldsymbol{\Sigma}_0^{-1}}^2-\|\boldsymbol{\mu}_T-\boldsymbol{\mu}^*\|_{\boldsymbol{\Sigma}_{T-1}^{-1}}^2\right] \tag{78}$$

$$\leq \frac{1}{2}\sum_{t=0}^{T-1}\left[\|\boldsymbol{\mu}_t-\boldsymbol{\mu}^*\|_{2\beta_t\boldsymbol{G}_t}^2-\frac{c\beta_t}{2}\|\boldsymbol{\mu}_t-\boldsymbol{\mu}^*\|^2\right]+\|\boldsymbol{\Sigma}_0^{-1}\|_{\mathrm{F}}D^2 \tag{79}$$

$$\leq \frac{1}{2}\sum_{t=0}^{T-1}\left[\frac{c\beta_t}{2}\|\boldsymbol{\mu}_t-\boldsymbol{\mu}^*\|^2-\frac{c\beta_t}{2}\|\boldsymbol{\mu}_t-\boldsymbol{\mu}^*\|^2\right]+\|\boldsymbol{\Sigma}_0^{-1}\|_{\mathrm{F}}D^2 \tag{80}$$

$$=\|\boldsymbol{\Sigma}_0^{-1}\|_{\mathrm{F}}D^2, \tag{81}$$

where the second inequality is due to the update rule of $\boldsymbol{\Sigma}_t$ and $\|\boldsymbol{\mu}_t-\boldsymbol{\mu}^*\|\leq D$, and the third inequality is due to Cauchy-Schwarz inequality and $\boldsymbol{G}_t\preceq\frac{c}{4}\boldsymbol{I}$. Then we have

$$\sum_{t=0}^{T-1}\beta_t\mathbb{E}[A_t]\leq \|\boldsymbol{\Sigma}_0^{-1}\|_{\mathrm{F}}D^2+\sum_{t=0}^{T-1}\left(2H\beta_t^2\|\boldsymbol{\Sigma}_t\|_{\mathrm{F}}+DL\rho_tU_t+\beta_t^2\|\boldsymbol{\Sigma}_t^{\frac{1}{2}}\|_{\mathrm{F}}\mathbb{E}\|\boldsymbol{\Sigma}_t^{\frac{1}{2}}\boldsymbol{g}_t\|^2\right) \tag{82}$$

$$\leq \|\boldsymbol{\Sigma}_0^{-1}\|_{\mathrm{F}}D^2+\sum_{t=0}^{T-1}\left(\frac{2H\beta_t^2\sqrt{d}}{2\xi\sum_{k=1}^t\beta_k}+DL\rho_tU_t+\frac{L_F^2R^{\frac{1}{2}}(1+2\rho r')(d+4)^2\beta_t^2}{2\xi(\sum_{k=1}^t\beta_k)}\right), \tag{83}$$

where the first inequality is due to Eq. (81) and Eq. (75), and the second inequality is due to Lemma C.6 and $\|\boldsymbol{\Sigma}_t^{\frac{1}{2}}\|_{\mathrm{F}}\leq R^{\frac{1}{2}}$. According to C.3 (b), we have $U_t\leq\max(\frac{2\sqrt{2}d^{\frac{1}{4}}}{(2\xi\sum_{k=1}^t\beta_k)^{\frac{1}{2}}},\frac{4r\sqrt{d}}{2\xi\sum_{k=1}^t\beta_k})$. Therefore, there exists a constant $t^*$, when $t>t^*-1$, $U_t\leq\frac{2\sqrt{2}d^{\frac{1}{4}}}{(2\xi\sum_{k=1}^t\beta_k)^{\frac{1}{2}}}$. Denote $\sum_{t=0}^{t^*-1}\frac{4r\rho_t\sqrt{d}}{2\xi\sum_{k=1}^t\beta_k}$ by a constant $\Gamma$. Then if $T>t^*$, we have

$$\frac{1}{T}\sum_{t=0}^{T-1}\mathbb{E}[A_t]\leq \frac{\|\boldsymbol{\Sigma}_0^{-1}\|_{\mathrm{F}}D^2}{T\beta_t}+\frac{1}{T}\sum_{t=0}^{T-1}\left(\frac{2H\beta_t\sqrt{d}}{2\xi\sum_{k=1}^t\beta_k}+\frac{L_F^2R^{\frac{1}{2}}(1+2\rho r')(d+4)^2\beta_t}{2\xi\sum_{k=1}^t\beta_k}\right) \\ +\frac{1}{T}\sum_{t=t*}^{T-1}\left(\frac{2\sqrt{2}DL\rho_td^{\frac{1}{4}}}{\beta_t(2\xi\sum_{k=1}^t\beta_k)^{\frac{1}{2}}}\right)+\frac{DL\Gamma}{T\beta_t}, \tag{84}$$

Let $\beta_t=\beta$ and $\rho_t=\frac{\rho_0}{\sqrt{t}}$, we can obtain that

$$\frac{1}{T}\sum_{t=0}^{T-1}\mathbb{E}[A_t]\leq \frac{\|\boldsymbol{\Sigma}_0^{-1}\|_{\mathrm{F}}D^2}{T\beta}+\frac{1}{T}\sum_{t=0}^{T-1}\left(\frac{C}{2\xi t}\right)+\frac{1}{T}\sum_{t=t*}^{T-1}\left(\frac{2\sqrt{2}DL\rho_0d^{\frac{1}{4}}}{\sqrt{2\xi}\beta^{\frac{3}{2}}t}\right)+\frac{DL\Gamma}{T\beta}, \tag{85}$$

where $C=2H\sqrt{d}+L_F^2R^{\frac{1}{2}}(1+2\rho r')(d+4)^2$. Since we have $\sum_{t=1}^T\frac{1}{t}\leq 1+\log(T)$, we obtain

$$\frac{1}{T}\sum_{t=0}^{T-1}\mathbb{E}\left[J(\boldsymbol{\mu}_{t+1},\boldsymbol{\Sigma}_t)-J(\boldsymbol{\mu}^*,0)\right]=\frac{1}{T}\sum_{t=0}^{T-1}\mathbb{E}[A_t]=\mathcal{O}\left(\frac{1}{T}+\frac{\log T}{T}\right), \tag{86}$$

where we reach the conclusion.

**Remark D.1** *In Theorem 4.3, if we set $\beta=\mathcal{O}(1)$ and $\rho=\mathcal{O}(\frac{1}{\sqrt{T}})$, then the third term in right-hand side of Eq. (84) has a convergence rate of $\mathcal{O}(\frac{\log T}{T})$. If we set $\beta=\mathcal{O}(1)$ and $\rho=\mathcal{O}(1)$, then*

*the third term in right-hand side of Eq. (84) has a convergence rate of $\mathcal{O}(\frac{1}{\sqrt{T}})$, since we have $\sum_{t=1}^{T} \frac{1}{\sqrt{t}} \leq 2\sqrt{T}$. Then we obtain*

$$\frac{1}{T}\sum_{t=0}^{T-1}\mathbb{E}\left[J(\boldsymbol{\mu}_{t+1}, \boldsymbol{\Sigma}_t) - J(\boldsymbol{\mu}^*, 0)\right] = \frac{1}{T}\sum_{t=0}^{T-1}\mathbb{E}[A_t] = \mathcal{O}\left(\frac{1}{T} + \frac{\log T}{T} + \frac{1}{\sqrt{T}}\right). \quad (87)$$

*This implies $\frac{1}{T}\sum_{t=0}^{T-1}\mathbb{E}\left[J(\boldsymbol{\mu}_{t+1}, \boldsymbol{\Sigma}_t) - J(\boldsymbol{\mu}^*, 0)\right] = \mathcal{O}\left(\frac{1}{\sqrt{T}}\right).$*

### D.3 PROOF OF THE PROPOSITION 4.5

Since the datasets $\mathcal{D}$ and $\mathcal{B}$ are i.i.d. sampled from one data distribution $P(X, y)$. We have $\mathbb{E}_{(X,y)\sim P}[F(\boldsymbol{x}; (X, y))] = \mathbb{E}[F(\boldsymbol{x}; \mathcal{B})] = \mathbb{E}[F(\boldsymbol{x}; \mathcal{D})]$. We obtain that

$$\|F(\boldsymbol{x}; \mathcal{B}) - F(\boldsymbol{x}; \mathcal{D})\|_2^2 \leq 2\|F(\boldsymbol{x}; \mathcal{B}) - \mathbb{E}F(\boldsymbol{x}; \mathcal{B})\|_2^2 + 2\|F(\boldsymbol{x}; \mathcal{D}) - \mathbb{E}F(\boldsymbol{x}; \mathcal{D})\|_2^2 = 2(\varepsilon_{\mathrm{B}}^2 + \varepsilon_{\mathrm{D}}^2) \quad (88)$$

### D.4 PROOF OF THEOREM 4.6

Note that for SABO with a mini-batch function query, the update rule of $\boldsymbol{\mu}$ can be represented as $\boldsymbol{\mu}_{t+1} = \boldsymbol{\mu}_t - \beta_t \boldsymbol{\Sigma}_t \boldsymbol{g}_t$. Let $B_t = J(\boldsymbol{\mu}_t, \boldsymbol{\Sigma}_t) - J(\boldsymbol{\mu}^*, 0)$, then by using Eq. (61), we have

$$\beta_t \mathbb{E}[B_t] \leq \beta_t \mathbb{E}[\nabla_{\boldsymbol{\mu}} J(\boldsymbol{\theta}_t)^\top (\boldsymbol{\mu}_t - \boldsymbol{\mu}^*)] + 2\beta_t \mathbb{E}[\nabla_{\boldsymbol{\Sigma}} J(\boldsymbol{\theta}_t)^\top \boldsymbol{\Sigma}_t] - \frac{c\beta_t}{2}\|\boldsymbol{\mu}_t - \boldsymbol{\mu}^*\|^2 \quad (89)$$

$$\leq \beta_t \mathbb{E}[\nabla_{\boldsymbol{\mu}} J(\boldsymbol{\theta}_t)^\top (\boldsymbol{\mu}_t - \boldsymbol{\mu}^*)] + 2\beta_t H\|\boldsymbol{\Sigma}_t\|_{\mathrm{F}} - \frac{c\beta_t}{2}\|\boldsymbol{\mu}_t - \boldsymbol{\mu}^*\|^2, \quad (90)$$

where the second inequality is due to Lemma C.1 and Lipschitz continuous assumption of the function $J(\boldsymbol{\theta})$. Note that

$$\|\boldsymbol{\mu}_t - \boldsymbol{\mu}^*\|_{\boldsymbol{\Sigma}_t^{-1}}^2 - \|\boldsymbol{\mu}_{t+1} - \boldsymbol{\mu}^*\|_{\boldsymbol{\Sigma}_t^{-1}}^2$$

$$= \|\boldsymbol{\mu}_t - \boldsymbol{\mu}^*\|_{\boldsymbol{\Sigma}_t^{-1}}^2 - \|\boldsymbol{\mu}_t - \beta_t \boldsymbol{\Sigma}_t \boldsymbol{g}_t - \boldsymbol{\mu}^*\|_{\boldsymbol{\Sigma}_t^{-1}}^2 \quad (91)$$

$$= \|\boldsymbol{\mu}_t - \boldsymbol{\mu}^*\|_{\boldsymbol{\Sigma}_t^{-1}}^2 - \left(\|\boldsymbol{\mu}_t - \boldsymbol{\mu}^*\|_{\boldsymbol{\Sigma}_t^{-1}}^2 - 2\beta_t \langle \boldsymbol{\mu}_t - \boldsymbol{\mu}^*, \boldsymbol{g}_t \rangle + \beta_t^2 \langle \boldsymbol{\Sigma}_t \boldsymbol{g}_t, \boldsymbol{g}_t \rangle\right) \quad (92)$$

$$= 2\beta_t \boldsymbol{g}_t^\top (\boldsymbol{\mu}_t - \boldsymbol{\mu}^*) - \beta_t^2 (\boldsymbol{\Sigma}_t \boldsymbol{g}_t)^\top \boldsymbol{g}_t. \quad (93)$$

It follows that

$$\boldsymbol{g}_t^\top (\boldsymbol{\mu}_t - \boldsymbol{\mu}^*) = \frac{1}{2\beta_t}\left(\|\boldsymbol{\mu}_t - \boldsymbol{\mu}^*\|_{\boldsymbol{\Sigma}_t^{-1}}^2 - \|\boldsymbol{\mu}_{t+1} - \boldsymbol{\mu}^*\|_{\boldsymbol{\Sigma}_t^{-1}}^2\right) + \frac{\beta_t}{2}(\boldsymbol{\Sigma}_t \boldsymbol{g}_t)^\top \boldsymbol{g}_t \quad (94)$$

$$\leq \frac{1}{2\beta_t}\left(\|\boldsymbol{\mu}_t - \boldsymbol{\mu}^*\|_{\boldsymbol{\Sigma}_t^{-1}}^2 - \|\boldsymbol{\mu}_{t+1} - \boldsymbol{\mu}^*\|_{\boldsymbol{\Sigma}_t^{-1}}^2\right) + \beta_t \|\boldsymbol{\Sigma}_t^{\frac{1}{2}}\|_{\mathrm{F}}\|\boldsymbol{\Sigma}_t^{\frac{1}{2}}\boldsymbol{g}_t\|^2, \quad (95)$$

where the inequality is due to $\beta_t \geq 0$ and Lemma C.1. According to Lemma C.7, we have $\mathbb{E}\boldsymbol{g}_t = \nabla_{\boldsymbol{\mu}} J(\boldsymbol{\theta}_t + \boldsymbol{\delta}_t)$. Therefore

$$\mathbb{E}\left[(\nabla_{\boldsymbol{\mu}} J(\boldsymbol{\theta}_t) - \boldsymbol{g}_t)^\top (\boldsymbol{\mu}_t - \boldsymbol{\mu}^*)\right]$$

$$= \mathbb{E}_{\boldsymbol{z}}\left[(\nabla_{\boldsymbol{\mu}} J(\boldsymbol{\theta}_t) - \nabla_{\boldsymbol{\mu}} J(\boldsymbol{\theta}_t + \boldsymbol{\delta}_t))^\top (\boldsymbol{\mu}_t - \boldsymbol{\mu}^*)\right] + \mathbb{E}_{\boldsymbol{z}}\left[(\nabla_{\boldsymbol{\mu}} J(\boldsymbol{\theta}_t + \boldsymbol{\delta}_t) - \boldsymbol{g}_t)^\top (\boldsymbol{\mu}_t - \boldsymbol{\mu}^*)\right] \quad (96)$$

$$\leq \|\boldsymbol{\mu}_t - \boldsymbol{\mu}^*\|\|\nabla_{\boldsymbol{\mu}} J(\boldsymbol{\theta}_t) - \nabla_{\boldsymbol{\mu}} J(\boldsymbol{\theta}_t + \boldsymbol{\delta}_t)\| \quad (97)$$

$$\leq DL\|\boldsymbol{\delta}_t\| \quad (98)$$

$$\leq DL\rho_t U_t, \quad (99)$$

where the first inequality is due to the Cauchy-Schwarz inequality, the second inequality is due to $\|\boldsymbol{\mu}_t - \boldsymbol{\mu}^*\| \leq D$ and smoothness assumption of the function $J(\boldsymbol{\theta})$, the last inequality is due to Lemma C.4, and $U_t = \max(2\sqrt{2}\|\boldsymbol{\Sigma}_t^{\frac{1}{2}}\|_{\mathrm{F}}, 4r\|\boldsymbol{\Sigma}_t\|_{\mathrm{F}})$. Then we have

$$\mathbb{E}_{\boldsymbol{z}}\left[\nabla_{\boldsymbol{\mu}} J(\boldsymbol{\theta}_t)^\top (\boldsymbol{\mu}_t - \boldsymbol{\mu}^*)\right] \quad (100)$$

$$= \mathbb{E}_{\boldsymbol{z}}\left[\boldsymbol{g}_t^\top (\boldsymbol{\mu}_t - \boldsymbol{\mu}^*) + (\nabla_{\boldsymbol{\mu}} J(\boldsymbol{\theta}_t) - \boldsymbol{g}_t)^\top (\boldsymbol{\mu}_t - \boldsymbol{\mu}^*)\right] \quad (101)$$

$$\leq \frac{1}{2\beta_t}\mathbb{E}_{\boldsymbol{z}}[\|\boldsymbol{\mu}_t - \boldsymbol{\mu}^*\|_{\boldsymbol{\Sigma}_t^{-1}}^2 - \|\boldsymbol{\mu}_{t+1} - \boldsymbol{\mu}^*\|_{\boldsymbol{\Sigma}_t^{-1}}^2] + \beta_t \|\boldsymbol{\Sigma}_t^{\frac{1}{2}}\|_{\mathrm{F}}\mathbb{E}\|\boldsymbol{\Sigma}_t^{\frac{1}{2}}\boldsymbol{g}_t\|^2 + DL\rho_t U_t, \quad (102)$$

where the inequality is due to Eq. (95) and Eq. (99). Then substituting Eq. (102) into Eq. (90) and multiplying $\beta_t$ on both sides of the inequality, we have

$$
\begin{aligned}
\beta_t \mathbb{E}[B_t] \leq & \frac{1}{2} \mathbb{E}[\|\boldsymbol{\mu}_t - \boldsymbol{\mu}^*\|_{\boldsymbol{\Sigma}_t^{-1}}^2 - \|\boldsymbol{\mu}_{t+1} - \boldsymbol{\mu}^*\|_{\boldsymbol{\Sigma}_t^{-1}}^2] - \frac{c\beta_t}{2} \|\boldsymbol{\mu}_t - \boldsymbol{\mu}^*\|^2 + 2\beta_t^2 H \|\boldsymbol{\Sigma}_t\|_{\mathrm{F}} \\
& + DL\rho_t U_t + \beta_t^2 \|\boldsymbol{\Sigma}_t^{\frac{1}{2}}\|_{\mathrm{F}} \mathbb{E} \|\boldsymbol{\Sigma}_t^{\frac{1}{2}} \boldsymbol{g}_t\|^2,
\end{aligned}
\tag{103}
$$

We further obtain that

$$
\sum_{t=0}^{T-1} \left[ \frac{1}{2} \mathbb{E}[\|\boldsymbol{\mu}_t - \boldsymbol{\mu}^*\|_{\boldsymbol{\Sigma}_t^{-1}}^2 - \|\boldsymbol{\mu}_{t+1} - \boldsymbol{\mu}^*\|_{\boldsymbol{\Sigma}_t^{-1}}^2] - \frac{c\beta_t}{2} \|\boldsymbol{\mu}_t - \boldsymbol{\mu}^*\|^2 \right]
\tag{104}
$$

$$
\leq \frac{1}{2} \sum_{t=0}^{T-1} \left[ \|\boldsymbol{\mu}_t - \boldsymbol{\mu}^*\|_{\boldsymbol{\Sigma}_t^{-1}}^2 - \|\boldsymbol{\mu}_{t-1} - \boldsymbol{\mu}^*\|_{\boldsymbol{\Sigma}_{t-1}^{-1}}^2 - \frac{c\beta_t}{2} \|\boldsymbol{\mu}_t - \boldsymbol{\mu}^*\|^2 \right]
\tag{105}
$$

$$
+ \frac{1}{2} \left[ \|\boldsymbol{\mu}_0 - \boldsymbol{\mu}^*\|_{\boldsymbol{\Sigma}_0^{-1}}^2 - \|\boldsymbol{\mu}_T - \boldsymbol{\mu}^*\|_{\boldsymbol{\Sigma}_{T-1}^{-1}}^2 \right]
\tag{106}
$$

$$
\leq \frac{1}{2} \sum_{t=0}^{T-1} \left[ \|\boldsymbol{\mu}_t - \boldsymbol{\mu}^*\|_{2\beta_t \boldsymbol{G}_t}^2 - \frac{c\beta_t}{2} \|\boldsymbol{\mu}_t - \boldsymbol{\mu}^*\|^2 \right] + \|\boldsymbol{\Sigma}_0^{-1}\|_{\mathrm{F}} D^2
\tag{107}
$$

$$
\leq \frac{1}{2} \sum_{t=0}^{T-1} \left[ \frac{c\beta_t}{2} \|\boldsymbol{\mu}_t - \boldsymbol{\mu}^*\|^2 - \frac{c\beta_t}{2} \|\boldsymbol{\mu}_t - \boldsymbol{\mu}^*\|^2 \right] + \|\boldsymbol{\Sigma}_0^{-1}\|_{\mathrm{F}} D^2
\tag{108}
$$

$$
= \|\boldsymbol{\Sigma}_0^{-1}\|_{\mathrm{F}} D^2,
\tag{109}
$$

where the second inequality is due to the update rule of $\boldsymbol{\Sigma}_t$ and $\|\boldsymbol{\mu}_t - \boldsymbol{\mu}^*\| \leq D$, and the third inequality is due to Cauchy-Schwarz inequality and $\boldsymbol{G}_t \preceq \frac{c}{4} \boldsymbol{I}$. Then we have

$$
\begin{aligned}
\sum_{t=0}^{T-1} \beta_t \mathbb{E}[B_t] \leq & \|\boldsymbol{\Sigma}_0^{-1}\|_{\mathrm{F}} D^2 + \sum_{t=0}^{T-1} \left( 2H\beta_t^2 \|\boldsymbol{\Sigma}_t\|_{\mathrm{F}} + DL\rho_t U_t + \beta_t^2 \|\boldsymbol{\Sigma}_t^{\frac{1}{2}}\|_{\mathrm{F}} \mathbb{E} \|\boldsymbol{\Sigma}_t^{\frac{1}{2}} \boldsymbol{g}_t\|^2 \right) \\
\end{aligned}
\tag{110}
$$

$$
\begin{aligned}
\leq & \|\boldsymbol{\Sigma}_0^{-1}\|_{\mathrm{F}} D^2 + \sum_{t=0}^{T-1} \left( \frac{2H\beta_t^2 \sqrt{d}}{2\xi \sum_{k=1}^t \beta_k} + DL\rho_t U_t + \frac{2\beta_t^2 \|\boldsymbol{\Sigma}_t^{\frac{1}{2}}\|_{\mathrm{F}} (d+4)\varepsilon^2}{3} \right. \\
& \left. + \frac{L_F^2 (1 + 2\rho r')(d+4)^2 \beta_t^2 \|\boldsymbol{\Sigma}_t^{\frac{1}{2}}\|_{\mathrm{F}}}{6\xi (\sum_{k=1}^t \beta_k)} \right)
\end{aligned}
\tag{111}
$$

$$
\begin{aligned}
\leq & \|\boldsymbol{\Sigma}_0^{-1}\|_{\mathrm{F}} D^2 + \sum_{t=0}^{T-1} \left( \frac{2H\beta_t^2 \sqrt{d}}{2\xi \sum_{k=1}^t \beta_k} + DL\rho_t U_t + \frac{2\beta_t^2 (d+4)d^{\frac{1}{4}}\varepsilon^2}{3(2\xi \sum_{k=1}^t \beta_k)^{\frac{1}{2}}} \right. \\
& \left. + \frac{L_F^2 (1 + 2\rho r')(d+4)^2 \beta_t^2 R^{\frac{1}{2}}}{6\xi (\sum_{k=1}^t \beta_k)} \right),
\end{aligned}
\tag{112}
$$

where the first inequality is due to Eq. (103) and Eq. (109), the second inequality is due to Lemma C.3 (b) and Lemma C.8, and the third inequality is due to $\|\boldsymbol{\Sigma}_t^{\frac{1}{2}}\|_{\mathrm{F}} \leq R^{\frac{1}{2}}$ and Lemma C.3 (b). According to C.3 (b), we have $U_t \leq \max(\frac{2\sqrt{2}d^{\frac{1}{4}}}{(2\xi \sum_{k=1}^t \beta_k)^{\frac{1}{2}}}, \frac{4r\sqrt{d}}{2\xi \sum_{k=1}^t \beta_k})$. Therefore, there exists a constant $t^*$, when $t > t^* - 1$, $U_t \leq \frac{2\sqrt{2}d^{\frac{1}{4}}}{(2\xi \sum_{k=1}^t \beta_k)^{\frac{1}{2}}}$. Denote $\sum_{t=0}^{t^*-1} \frac{4r\rho_t \sqrt{d}}{2\xi \sum_{k=1}^t \beta_k}$ by a constant $\Gamma$. Then if $T > t^*$, we have

$$
\begin{aligned}
\frac{1}{T} \sum_{t=0}^{T-1} \mathbb{E}[B_t] \leq & \frac{\|\boldsymbol{\Sigma}_0^{-1}\|_{\mathrm{F}} D^2}{T\beta_t} + \frac{1}{T} \sum_{t=0}^{T-1} \left( \frac{2H\beta_t \sqrt{d}}{2\xi \sum_{k=1}^t \beta_k} + \frac{L_F^2 R^{\frac{1}{2}} (1 + 2\rho r')(d+4)^2 \beta_t}{6\xi \sum_{k=1}^t \beta_k} \right) \\
& + \frac{1}{T} \sum_{t=t*}^{T-1} \left( \frac{2\sqrt{2}DL\rho_t d^{\frac{1}{4}}}{\beta_t (2\xi \sum_{k=1}^t \beta_k)^{\frac{1}{2}}} \right) + \frac{1}{T} \sum_{t=0}^{T-1} \left( \frac{2\beta_t (d+4)d^{\frac{1}{4}}\varepsilon^2}{3(2\xi \sum_{k=1}^t \beta_k)^{\frac{1}{2}}} \right) + \frac{DL\Gamma}{T\beta_t},
\end{aligned}
\tag{113}
$$

Let $\beta_t = \beta$ and $\rho_t = \rho$, we have

$$
\begin{aligned}
\frac{1}{T} \sum_{t=0}^{T-1} \mathbb{E}[B_t] \leq & \frac{\|\mathbf{\Sigma}_0^{-1}\|_{\mathrm{F}} D^2}{T\beta} + \frac{1}{T} \sum_{t=0}^{T-1} \left( \frac{2H\sqrt{d}}{2\xi t} + \frac{L_F^2 R^{\frac{1}{2}}(1+2\rho r')(d+4)^2}{6\xi t} \right) \\
& + \frac{1}{T} \sum_{t=t*}^{T-1} \left( \frac{2\sqrt{2}DL\rho d^{\frac{1}{4}}}{\sqrt{2\xi}\beta^{\frac{3}{2}}\sqrt{t}} \right) + \frac{1}{T} \sum_{t=0}^{T-1} \left( \frac{2(d+4)d^{\frac{1}{4}}\varepsilon^2}{3\sqrt{2\xi}\sqrt{t}} \right) + \frac{DL\Gamma}{T\beta}.
\end{aligned}
\tag{114}
$$

Since we have $\sum_{t=1}^{T} \frac{1}{t} \leq 1 + \log(T)$ and $\sum_{t=1}^{T} \frac{1}{\sqrt{t}} \leq 2\sqrt{T}$, we obtain

$$
\frac{1}{T} \sum_{t=0}^{T-1} \mathbb{E}\left[ J(\boldsymbol{\mu}_{t+1}, \mathbf{\Sigma}_t) - J(\boldsymbol{\mu}^*, 0) \right] = \frac{1}{T} \sum_{t=0}^{T-1} \mathbb{E}[B_t] = \mathcal{O}\left( \frac{1}{T} + \frac{\log T}{T} + \frac{1}{\sqrt{T}} \right),
\tag{115}
$$

where we reach the conclusion.

### D.5 PROOF OF THEOREM 4.8

Using PAC-Bayesian bound (McAllester, 1999) and following (Dziugaite & Roy, 2017), for any prior distribution $p$ and any posterior distribution $q$ that may be dependent on finite dataset $\mathcal{S}$, where data set $S$ with $M$ i.i.d. samples drawn from data distribution $P(X, y)$, we have

$$
\forall q, \quad \mathbb{E}_{q(\boldsymbol{x})}\left[ \mathbb{E}_{P(X,y)}[F(\boldsymbol{x}; (X, y))] \right] \leq \mathbb{E}_{q(\boldsymbol{x})}\left[ F(\boldsymbol{x}; \mathcal{S}) \right] + \sqrt{\frac{\mathrm{KL}(q\|p) + \log(\frac{M}{\kappa})}{2(M-1)}}.
\tag{116}
$$

Set the posterior distribution as $q = p_{\boldsymbol{\theta}} := \mathcal{N}(\boldsymbol{\mu}, \mathbf{\Sigma})$. Denote the set $\mathcal{M}(\boldsymbol{\theta}) = \{\boldsymbol{\delta} \mid \mathrm{KL}(p_{\boldsymbol{\theta}+\boldsymbol{\delta}}\|p_{\boldsymbol{\theta}}) + \mathrm{KL}(p_{\boldsymbol{\delta}}\|p_{\boldsymbol{\theta}+\boldsymbol{\delta}}) \leq \rho^2\}$. We can choose $p \in \mathcal{M}(\boldsymbol{\theta})$. Then, we know that for the prior distribution $p$, the following inequality holds with a probability at least $1 - \kappa$,

$$
\mathbb{E}_{p_{\boldsymbol{\theta}}(\boldsymbol{x})}\left[ \mathbb{E}_{P(X,y)}[F(\boldsymbol{x}; (X, y))] \right] \leq \mathbb{E}_{p_{\boldsymbol{\theta}}(\boldsymbol{x})}\left[ F(\boldsymbol{x}; \mathcal{S}) \right] + \sqrt{\frac{\mathrm{KL}(p_{\boldsymbol{\theta}}\|p) + \log(\frac{M}{\kappa})}{2(M-1)}}.
\tag{117}
$$

Note that for any density $p, q$, we have $\mathrm{KL}(p\|q) \geq 0$. Thus, we know $\mathcal{M}(\boldsymbol{\theta}) \subset \mathcal{C}(\boldsymbol{\theta})$. It follows that

$$
\mathbb{E}_{p_{\boldsymbol{\theta}}(\boldsymbol{x})}\left[ \mathbb{E}_{P(X,y)}[F(\boldsymbol{x}; (X, y))] \right] \leq \mathbb{E}_{p_{\boldsymbol{\theta}}(\boldsymbol{x}) \in \mathcal{C}(\boldsymbol{\theta})}\left[ F(\boldsymbol{x}; \mathcal{S}) \right] + \sqrt{\frac{\mathrm{KL}(p_{\boldsymbol{\theta}}\|p) + \log(\frac{M}{\kappa})}{2(M-1)}}
\tag{118}
$$

$$
\leq \max_{\boldsymbol{\delta} \in \mathcal{C}(\boldsymbol{\theta})} \mathbb{E}_{p_{\boldsymbol{\theta}+\boldsymbol{\delta}}}\left[ F(\boldsymbol{x}; \mathcal{S}) \right] + \max_{\boldsymbol{\delta} \in \mathcal{M}(\boldsymbol{\theta})} \sqrt{\frac{\mathrm{KL}(p_{\boldsymbol{\theta}}\|p_{\boldsymbol{\theta}+\boldsymbol{\delta}}) + \log(\frac{M}{\kappa})}{2(M-1)}}
\tag{119}
$$

$$
\leq \max_{\boldsymbol{\delta} \in \mathcal{C}(\boldsymbol{\theta})} \mathbb{E}_{p_{\boldsymbol{\theta}+\boldsymbol{\delta}}}\left[ F(\boldsymbol{x}; \mathcal{S}) \right] + \sqrt{\frac{\rho^2 + \log(\frac{M}{\kappa})}{2(M-1)}}.
\tag{120}
$$

Note that $F(\boldsymbol{x}; (X, y))$ is convex function w.r.t. $\boldsymbol{x}$, we know that $\mathbb{E}_{P(X,y)}[F(\boldsymbol{x}; (X, y))]$ is a convex function w.r.t. $\boldsymbol{x}$. It follows that

$$
\mathbb{E}_{P(X,y)}[F(\boldsymbol{\mu}; (X, y))] = \mathbb{E}_{P(X,y)}[F(\mathbb{E}_{p_{\boldsymbol{\theta}}(\boldsymbol{x})}[\boldsymbol{x}]; (X, y))]
\tag{121}
$$

$$
\leq \mathbb{E}_{p_{\boldsymbol{\theta}}(\boldsymbol{x})}\left[ \mathbb{E}_{P(X,y)}[F(\boldsymbol{x}; (X, y))] \right].
\tag{122}
$$

Finally, we know that with a probability of at least $1 - \kappa$, the following inequality holds.

$$
\mathbb{E}_{P(X,y)}[F(\boldsymbol{\mu}; \boldsymbol{z}, y)] \leq \max_{\boldsymbol{\delta} \in \mathcal{C}(\boldsymbol{\theta})} \mathbb{E}_{p_{\boldsymbol{\theta}+\boldsymbol{\delta}}}\left[ F(\boldsymbol{x}; \mathcal{S}) \right] + \sqrt{\frac{\rho^2 + \log(\frac{M}{\kappa})}{2(M-1)}}.
\tag{123}
$$

## E PROOF OF TECHNICAL LEMMAS

In this section, we provide the proof of lemmas in Appendix C. Note that Lemma C.1 and Lemma C.2 can be directly obtained by Lemma B.1. and Lemma B.2. in (Ye et al., 2024), respectively.

### E.1 PROOF OF LEMMA C.3

(a): Since we have $\boldsymbol{\Sigma}_{t+1}^{-1} = \boldsymbol{\Sigma}_t^{-1} + 2\beta_t \boldsymbol{G}_t$. We can obtain that

$$\boldsymbol{\Sigma}_t^{-1} + 2b\beta_t \boldsymbol{I} \succeq \boldsymbol{\Sigma}_{t+1}^{-1} \succeq \boldsymbol{\Sigma}_t^{-1} + 2\xi\beta_t \boldsymbol{I}. \tag{124}$$

Summing up it over $t = 0, \ldots, T-1$, we have

$$\boldsymbol{\Sigma}_0^{-1} + 2b\sum_{t=1}^{T}\beta_t \boldsymbol{I} \succeq \boldsymbol{\Sigma}_T^{-1} \succeq \boldsymbol{\Sigma}_0^{-1} + 2\xi\sum_{t=1}^{T}\beta_t \boldsymbol{I}. \tag{125}$$

Therefore, we have

$$\frac{1}{2b\sum_{t=1}^{T}\beta_t \boldsymbol{I} + \boldsymbol{\Sigma}_0^{-1}} \preceq \boldsymbol{\Sigma}_T \preceq \frac{1}{2\xi\sum_{t=1}^{T}\beta_t \boldsymbol{I} + \boldsymbol{\Sigma}_0^{-1}}. \tag{126}$$

(b): We have

$$\|\boldsymbol{\Sigma}_t\|_{\mathrm{F}} \leq \left\|\frac{1}{2\xi\sum_{k=1}^{t}\beta_k \boldsymbol{I} + \boldsymbol{\Sigma}_0^{-1}}\right\|_{\mathrm{F}} \leq \left\|\frac{1}{2\xi\sum_{k=1}^{t}\beta_k \boldsymbol{I}}\right\|_{\mathrm{F}} = \frac{\sqrt{d}}{2\xi\sum_{k=1}^{t}\beta_k}. \tag{127}$$

### E.2 PROOF OF LEMMA C.4

In Algorithm 1 and Algorithm 2, for given $\boldsymbol{\theta}_t$, the perturbation $\boldsymbol{\delta}_t$ satisfies

$$\boldsymbol{\delta}_{\boldsymbol{\mu}_t} = \frac{1}{\lambda}\boldsymbol{\Sigma}_t \boldsymbol{g}_t' = \frac{\rho\boldsymbol{\Sigma}_t \boldsymbol{g}_t'}{\sqrt{\|\boldsymbol{\Sigma}_t \boldsymbol{G}_t'\|_{\mathrm{F}}^2 + 0.5\|\boldsymbol{\Sigma}_t^{\frac{1}{2}}\boldsymbol{g}_t'\|_2^2}} \leq \frac{\rho\sqrt{2}\boldsymbol{\Sigma}_t^{\frac{1}{2}}\boldsymbol{\Sigma}_t^{\frac{1}{2}}\boldsymbol{g}_t'}{\sqrt{\|\boldsymbol{\Sigma}_t^{\frac{1}{2}}\boldsymbol{g}_t'\|_2^2}}. \tag{128}$$

Therefore, we have $\|\boldsymbol{\delta}_{\boldsymbol{\mu}_t}\| \leq \rho\sqrt{2}\|\boldsymbol{\Sigma}_t^{\frac{1}{2}}\|_{\mathrm{F}}$. For $\boldsymbol{\delta}_{\boldsymbol{\Sigma}_t}$, we have

$$\boldsymbol{\delta}_{\boldsymbol{\Sigma}_t} = \frac{2\boldsymbol{\Sigma}_t \boldsymbol{G}_t'}{\lambda\boldsymbol{\Sigma}_t^{-1} - 2\boldsymbol{G}_t'} = \boldsymbol{\Sigma}_t \frac{\frac{2}{\lambda}\boldsymbol{\Sigma}_t \boldsymbol{G}_t'}{\boldsymbol{I} - \frac{2}{\lambda}\boldsymbol{\Sigma}_t \boldsymbol{G}_t'}. \tag{129}$$

Note that $\frac{2}{\lambda} \leq \frac{2\rho}{\|\boldsymbol{\Sigma}\boldsymbol{G}_t'\|_{\mathrm{F}}}$. Therefore, we can obtain that

$$\|\boldsymbol{\delta}_{\boldsymbol{\Sigma}_t}\|_{\mathrm{F}} \leq \|\boldsymbol{\Sigma}_t\|_{\mathrm{F}} \frac{2\rho}{\|\boldsymbol{I} - \frac{2}{\lambda}\boldsymbol{\Sigma}_t \boldsymbol{G}_t'\|_{\mathrm{F}}}. \tag{130}$$

If $\rho < \frac{\sqrt{d}}{2}$, there exist a constant $r > 0$, $\|\boldsymbol{I} - \frac{2}{\lambda}\boldsymbol{\Sigma}_t \boldsymbol{G}_t'\|_{\mathrm{F}} > \frac{1}{r}$ holds. Therefore we have $\|\boldsymbol{\delta}_{\boldsymbol{\Sigma}_t}\|_{\mathrm{F}} \leq 2\rho r\|\boldsymbol{\Sigma}_t\|_{\mathrm{F}}$.

### E.3 PROOF OF LEMMA C.5

We have

$$\mathbb{E}_{\boldsymbol{z}}[\boldsymbol{g}_t'] = \mathbb{E}_{\boldsymbol{z}}[\boldsymbol{\Sigma}_t^{-\frac{1}{2}}\boldsymbol{z}F(\boldsymbol{\mu}_t + \boldsymbol{\Sigma}_t^{\frac{1}{2}}\boldsymbol{z})] - \mathbb{E}_{\boldsymbol{z}}[\boldsymbol{\Sigma}_t^{-\frac{1}{2}}\boldsymbol{z}F(\boldsymbol{\mu}_t)] \tag{131}$$

$$= \mathbb{E}_{\boldsymbol{z}}[\boldsymbol{\Sigma}_t^{-\frac{1}{2}}\boldsymbol{z}F(\boldsymbol{\mu}_t + \boldsymbol{\Sigma}_t^{\frac{1}{2}}\boldsymbol{z})] \tag{132}$$

$$= \mathbb{E}_{\boldsymbol{x}\sim\mathcal{N}(\boldsymbol{\mu}_t, \boldsymbol{\Sigma}_t)}[\boldsymbol{\Sigma}_t^{-1}(\boldsymbol{x} - \boldsymbol{\mu}_t)F(\boldsymbol{x})] \tag{133}$$

$$= \nabla_{\boldsymbol{\mu}}\mathbb{E}_{p_{\boldsymbol{\theta}_t}}[F(\boldsymbol{x})], \tag{134}$$

where we reach the conclusion.

### E.4 PROOF OF LEMMA C.6

Denote the diagonal elements of $\boldsymbol{\Sigma}$ and $\widehat{\boldsymbol{\Sigma}}$ by $\boldsymbol{\sigma}$ and $\widehat{\boldsymbol{\sigma}}$, respectively. Then we have

$$\|\boldsymbol{\Sigma}_t^{\frac{1}{2}}\boldsymbol{g}_{t,j}\|^2 = \|\boldsymbol{\sigma}_t^{\frac{1}{2}} \odot \widehat{\boldsymbol{\sigma}}_t^{-\frac{1}{2}} \odot \boldsymbol{z}_j(F(\widehat{\boldsymbol{\mu}}_t + \widehat{\boldsymbol{\sigma}}_t^{\frac{1}{2}} \odot \boldsymbol{z}_j) - F(\widehat{\boldsymbol{\mu}}_t))\|^2. \tag{135}$$

Note that $\widehat{\boldsymbol{\Sigma}}_{t+1}^{-1} = \boldsymbol{\Sigma}_t^{-1} - 2\lambda \boldsymbol{G}_t'$, we have $\boldsymbol{\sigma}_t \leq \widehat{\boldsymbol{\sigma}}_t$. Since we have

$$\mathbb{E}\|\boldsymbol{\Sigma}_t^{\frac{1}{2}}\boldsymbol{g}_t\|_2^2 = \mathbb{E}\left\|\frac{1}{N}\sum_{j=1}^N \boldsymbol{\Sigma}_t^{\frac{1}{2}}\boldsymbol{g}_{t,j}\right\|^2 \leq \frac{1}{N}\sum_{j=1}^N \|\boldsymbol{\Sigma}_t^{\frac{1}{2}}\boldsymbol{g}_{t,j}\|^2. \tag{136}$$

It follows that

$$\mathbb{E}\|\boldsymbol{\Sigma}_t^{\frac{1}{2}}\boldsymbol{g}_t\|_2^2 \leq \frac{1}{N}\sum_{j=1}^N \mathbb{E}\|\widehat{\boldsymbol{\sigma}}_t^{\frac{1}{2}} \odot \widehat{\boldsymbol{\sigma}}_t^{-\frac{1}{2}} \odot \boldsymbol{z}_j(F(\widehat{\boldsymbol{\mu}}_t + \widehat{\boldsymbol{\sigma}}_t^{\frac{1}{2}} \odot \boldsymbol{z}_j) - F(\widehat{\boldsymbol{\mu}}_t))\|^2 \tag{137}$$

$$\leq \frac{1}{N}\sum_{j=1}^N \mathbb{E}\left[\|\boldsymbol{z}_j\|^2(F(\widehat{\boldsymbol{\mu}}_t + \widehat{\boldsymbol{\sigma}}_t^{\frac{1}{2}} \odot \boldsymbol{z}_j) - F(\widehat{\boldsymbol{\mu}}_t))^2\right] \tag{138}$$

$$\leq \frac{1}{N}\sum_{j=1}^N \mathbb{E}\left[L_F^2\|\boldsymbol{z}_j\|^4\|\widehat{\boldsymbol{\sigma}}_t\|_\infty\right]. \tag{139}$$

Note that

$$\widehat{\boldsymbol{\Sigma}}_t = \boldsymbol{\Sigma}_t + \boldsymbol{\delta}_{\boldsymbol{\Sigma}_t} = \boldsymbol{\Sigma}_t + \boldsymbol{\Sigma}_t \frac{\frac{2}{\lambda}\boldsymbol{\Sigma}_t\boldsymbol{G}_t'}{\boldsymbol{I} - \frac{2}{\lambda}\boldsymbol{\Sigma}_t\boldsymbol{G}_t'}. \tag{140}$$

Note that $\frac{2}{\lambda} \leq \frac{2\rho}{\|\boldsymbol{\Sigma}\boldsymbol{G}_t'\|_F}$. Then if $\rho < \frac{\sqrt{d}}{2}$, there exist a constant $r' > 0$, the inequality $\widehat{\boldsymbol{\sigma}}_t \leq \boldsymbol{\sigma}_t + 2\rho r'\boldsymbol{\sigma}_t$ holds. Therefore, we have

$$\|\widehat{\boldsymbol{\sigma}}_t\|_\infty \leq (1 + 2\rho r')\|\boldsymbol{\sigma}_t\|_\infty \leq \frac{1 + 2\rho r'}{\|\boldsymbol{\sigma}_0^{-1}\|_{min} + 2(\sum_{k=1}^t \beta_k)\xi}, \tag{141}$$

where $\|\cdot\|_{min}$ denotes the minimum element in the input. Noticed that $\mathbb{E}_{\boldsymbol{z}}[\|\boldsymbol{z}\|^2] \leq d + 4$ as shown in (Nesterov & Spokoiny, 2017), we obtain

$$\mathbb{E}\|\boldsymbol{\Sigma}_t^{\frac{1}{2}}\boldsymbol{g}_t\|_2^2 \leq \frac{L_F^2(1 + 2\rho r')(d + 4)^2}{2\xi(\sum_{k=1}^t \beta_k)}, \tag{142}$$

where we reach the conclusion.

### E.5 PROOF OF LEMMA C.7

We have

$$\mathbb{E}[\boldsymbol{g}_t'] = \mathbb{E}[\boldsymbol{\Sigma}_t^{-\frac{1}{2}}\boldsymbol{z}F(\boldsymbol{\mu}_t + \boldsymbol{\Sigma}_t^{\frac{1}{2}}\boldsymbol{z}; \mathcal{B})] - \mathbb{E}[\boldsymbol{\Sigma}_t^{-\frac{1}{2}}\boldsymbol{z}F(\boldsymbol{\mu}_t; \mathcal{B})] \tag{143}$$

$$= \mathbb{E}_{\boldsymbol{z}}[\boldsymbol{\Sigma}_t^{-\frac{1}{2}}\boldsymbol{z}F(\boldsymbol{\mu}_t + \boldsymbol{\Sigma}_t^{\frac{1}{2}}\boldsymbol{z}; \mathcal{D})] - \mathbb{E}_{\boldsymbol{z}}[\boldsymbol{\Sigma}_t^{-\frac{1}{2}}\boldsymbol{z}F(\boldsymbol{\mu}_t; \mathcal{D})] \tag{144}$$

$$= \mathbb{E}_{\boldsymbol{z}}[\boldsymbol{\Sigma}_t^{-\frac{1}{2}}\boldsymbol{z}F(\boldsymbol{\mu}_t + \boldsymbol{\Sigma}_t^{\frac{1}{2}}\boldsymbol{z})] \tag{145}$$

$$= \mathbb{E}_{\boldsymbol{x}\sim\mathcal{N}(\boldsymbol{\mu}_t, \boldsymbol{\Sigma}_t)}[\boldsymbol{\Sigma}_t^{-1}(\boldsymbol{x} - \boldsymbol{\mu}_t)F(\boldsymbol{x})] \tag{146}$$

$$= \nabla_{\boldsymbol{\mu}}\mathbb{E}_{p_{\boldsymbol{\theta}_t}}[F(\boldsymbol{x})], \tag{147}$$

where the second equality is due to Proposition 4.5.

### E.6 PROOF OF LEMMA C.8

Denote the diagonal elements of $\boldsymbol{\Sigma}$ and $\widehat{\boldsymbol{\Sigma}}$ by $\boldsymbol{\sigma}$ and $\widehat{\boldsymbol{\sigma}}$, respectively. Then we have

$$\|\boldsymbol{\Sigma}_t^{\frac{1}{2}}\boldsymbol{g}_{t,j}\|^2 = \|\boldsymbol{\sigma}_t^{\frac{1}{2}} \odot \widehat{\boldsymbol{\sigma}}_t^{-\frac{1}{2}} \odot \boldsymbol{z}_j(F(\widehat{\boldsymbol{\mu}}_t + \widehat{\boldsymbol{\sigma}}_t^{\frac{1}{2}} \odot \boldsymbol{z}_j; \mathcal{B}) - F(\widehat{\boldsymbol{\mu}}_t); \mathcal{B}))\|^2 \tag{148}$$

Note that $\widehat{\boldsymbol{\Sigma}}_{t+1}^{-1} = \boldsymbol{\Sigma}_t^{-1} - 2\lambda \boldsymbol{G}_t'$, we have $\boldsymbol{\sigma}_t \leq \widehat{\boldsymbol{\sigma}}_t$. Since we have

$$\mathbb{E}\|\boldsymbol{\Sigma}_t^{\frac{1}{2}}\boldsymbol{g}_t\|_2^2 = \mathbb{E}\left\|\frac{1}{N}\sum_{j=1}^N \boldsymbol{\Sigma}_t^{\frac{1}{2}}\boldsymbol{g}_{t,j}\right\|^2 \leq \frac{1}{N}\sum_{j=1}^N \|\boldsymbol{\Sigma}_t^{\frac{1}{2}}\boldsymbol{g}_{t,j}\|^2. \tag{149}$$

It follows that

$$\mathbb{E}\|\boldsymbol{\Sigma}_t^{\frac{1}{2}}\boldsymbol{g}_t\|_2^2 \leq \frac{1}{N}\sum_{j=1}^{N}\mathbb{E}\|\widehat{\boldsymbol{\sigma}}_t^{\frac{1}{2}} \odot \widehat{\boldsymbol{\sigma}}_t^{-\frac{1}{2}} \odot \boldsymbol{z}_j(F(\widehat{\boldsymbol{\mu}}_t + \widehat{\boldsymbol{\sigma}}_t^{\frac{1}{2}} \odot \boldsymbol{z}_j; \mathcal{B}) - F(\widehat{\boldsymbol{\mu}}_t; \mathcal{B}))\|^2 \tag{150}$$

$$\leq \frac{1}{N}\sum_{j=1}^{N}\mathbb{E}\left[\|\boldsymbol{z}_j\|^2(F(\widehat{\boldsymbol{\mu}}_t + \widehat{\boldsymbol{\sigma}}_t^{\frac{1}{2}} \odot \boldsymbol{z}_j; \mathcal{B}) - F(\widehat{\boldsymbol{\mu}}_t; \mathcal{B}))^2\right] \tag{151}$$

$$\leq \frac{1}{N}\sum_{j=1}^{N}\mathbb{E}\left[\|\boldsymbol{z}_j\|^2(\frac{1}{3}L_F^2\|\widehat{\boldsymbol{\sigma}}_t^{\frac{1}{2}} \odot \boldsymbol{z}_j\|^2 + \frac{2}{3}\varepsilon^2)\right], \tag{152}$$

where the third inequality is due to Proposition 4.5. Since $\widehat{\boldsymbol{\Sigma}}_t = \boldsymbol{\Sigma}_t + \boldsymbol{\Sigma}_t\frac{\frac{2}{\lambda}\boldsymbol{\Sigma}_t\boldsymbol{G}_t'}{\boldsymbol{I}-\frac{2}{\lambda}\boldsymbol{\Sigma}_t\boldsymbol{G}_t'}$, and $\frac{2}{\lambda} \leq \frac{2\rho}{\|\boldsymbol{\Sigma}\boldsymbol{G}_t'\|_{\mathrm{F}}}$. Then if $\rho < \frac{\sqrt{d}}{2}$, there exist a constant $r' > 0$, the inequality $\widehat{\boldsymbol{\sigma}}_t \leq \boldsymbol{\sigma}_t + 2\rho r'\boldsymbol{\sigma}_t$ holds. Therefore, we have

$$\|\widehat{\boldsymbol{\sigma}}_t\|_\infty \leq (1 + 2\rho r')\|\boldsymbol{\sigma}_t\|_\infty \leq \frac{1 + 2\rho r'}{\|\boldsymbol{\sigma}_0^{-1}\|_{min} + 2(\sum_{k=1}^{t}\beta_k)\xi}, \tag{153}$$

where $\|\cdot\|_{min}$ denotes the minimum element in the input. Noticed that $\mathbb{E}_{\boldsymbol{z}}[\|\boldsymbol{z}\|^2] \leq d + 4$ as shown in (Nesterov & Spokoiny, 2017), we obtain

$$\mathbb{E}\|\boldsymbol{\Sigma}_t^{\frac{1}{2}}\boldsymbol{g}_t\|_2^2 \leq \frac{1}{N}\sum_{j=1}^{N}\mathbb{E}\left[\frac{(1 + 2\rho r')L_F^2}{3}\|\boldsymbol{\sigma}_t\|_\infty \times \|\boldsymbol{z}_j\|^4 + \frac{2\varepsilon^2}{3}\|\boldsymbol{z}_j\|^2\right] \tag{154}$$

$$\leq \frac{L_F^2(1 + 2\rho r')(d + 4)^2}{6\xi(\sum_{k=1}^{t}\beta_k)} + \frac{2(d + 4)\varepsilon^2}{3}, \tag{155}$$

where we reach the conclusion.

## F UPDATED RULE UNDER TRANSFORMATION

To avoid the scaling problem, we can employ monotonic transformation for the objective $F(\boldsymbol{x})$, i.e. $h(F(\boldsymbol{x}_j)) = \frac{F(\boldsymbol{x}_j) - \hat{\mu}}{\hat{\sigma}}$, where $\hat{\mu}$ and $\hat{\sigma}$ denote mean and stand deviation of function values $F(\boldsymbol{x}_j)$, $j = 1, \ldots, N$. Then by applying this rescaling strategy, the update rule for $\boldsymbol{\mu}_t$ and $\boldsymbol{\Sigma}_t$ in $t$-th iteration can be written as

$$\boldsymbol{\mu}_{t+1} = \boldsymbol{\mu}_t - \frac{\beta_t}{N}\sum_{j=1}^{N}\boldsymbol{\Sigma}_t\widehat{\boldsymbol{\Sigma}}_t^{-1}(\boldsymbol{x}_j - \widehat{\boldsymbol{\mu}}_t)\frac{F(\boldsymbol{x}_j) - \hat{\mu}^t}{\hat{\boldsymbol{\sigma}}^t}, \tag{156}$$

$$\boldsymbol{\Sigma}_{t+1}^{-1} = \boldsymbol{\Sigma}_t^{-1} + \frac{\beta_t}{N}\sum_{j=1}^{N}\mathrm{diag}\left[\widehat{\boldsymbol{\Sigma}}_t^{-1}\left[\mathrm{diag}\left((\boldsymbol{x}_j - \widehat{\boldsymbol{\mu}}_t)(\boldsymbol{x}_j - \widehat{\boldsymbol{\mu}}_t)^\top\widehat{\boldsymbol{\Sigma}}_t^{-1}\right)\frac{F(\boldsymbol{x}_j) - \hat{\mu}^t}{\hat{\boldsymbol{\sigma}}^t}\right]\right], \tag{157}$$

where $\widehat{\boldsymbol{\Sigma}}_t = \boldsymbol{\Sigma}_t + \boldsymbol{\delta}_{\boldsymbol{\Sigma}_t}$, $\widehat{\boldsymbol{\mu}}_t = \boldsymbol{\mu}_t + \boldsymbol{\delta}_{\boldsymbol{\mu}_t}$, and $\boldsymbol{x}_j$ denotes the $j$-th sample sampled from the distribution $\mathcal{N}(\boldsymbol{x} \mid \widehat{\boldsymbol{\mu}}_t, \widehat{\boldsymbol{\Sigma}}_t)$.

## G ADDITIONAL MATERIALS FOR SECTION 6

In this section, we provide additional experiments and more implementation details of Section 6.

### G.1 SYNTHETIC PROBLEMS

The four numerical benchmark test functions employed in Section 6.1 are listed as follows:

$$F(\boldsymbol{x}) = \sum_{i=1}^{d} 10^{\frac{2(i-1)}{d-1}} \boldsymbol{x}_i^2, \tag{158}$$

$$F(\boldsymbol{x}) = \sum_{i=1}^{d} 10^{\frac{2(i-1)}{d-1}} |\boldsymbol{x}_i|^{\frac{1}{2}}, \tag{159}$$

$$F(\boldsymbol{x}) = \sqrt{\sum_{i=1}^{d} |x_i|^{2+4\frac{i-1}{d-1}}}, \tag{160}$$

$$F(\boldsymbol{x}) = \sin^2(\pi\omega_1) + \sum_{i=1}^{d-1}(\omega_i - 1)^2(1 + 10\sin^2(\omega_i\pi + 1)) + (\omega_d - 1)^2(1 + \sin^2(2\pi\omega_d)), \tag{161}$$

$$\text{where } \omega_i = 1 + \frac{x_i - 1}{4}, i \in \{1, \ldots, d\}.$$

Test functions (158)-(161) are called the ellipsoid function, $l_{\frac{1}{2}}$-ellipsoid function, different powers function and Levy function, respectively.

**Implementation Details.** For all the methods, we initialize $\boldsymbol{\mu}_0$ from the uniform distribution $\mathrm{Uni}[0,1]$, and set $\boldsymbol{\Sigma}_0 = \boldsymbol{I}$. For the INGO, BES, and SABO methods, we use a fixed step size of $\beta = 0.1$. According to our assumption in Theorem 4.3, we set $\rho = 100/\sqrt{T+1}$ for the proposed SABO method. We set the spacing $c = 1$ for the BES method and employ the default hyperparameter setting from He et al. (2020) for the MMES method. We then assess these methods using varying dimensions, i.e., $d \in \{200, 500, 1000\}$. For $d = 200$, we assess these methods using varying sample sizes, i.e., $N \in \{10, 50, 100\}$. The mean value of $\mathcal{E}$ over 3 independent runs is reported.

**Results.** Figure 2 and 3 show the results on four test functions with problem dimensions $d = 200$ and $d = 1000$, respectively. Figure 4 and 5 show the results on 200-dimensional test functions with sample size $N = 10$ and $N = 100$, respectively. Combining these results with the result from Figure 1, we observe consistent performance for the proposed ASBO method. It achieves a similar convergence result to the INGO method, as they have the same theoretical convergent rate. In some cases, i.e., Figure 3 (d) and Figure 4 (d), it converges slightly faster than INGO. The CMA-ES method and MMES method both work for ellipsoid and different powers functions, but fail in $l_{\frac{1}{2}}$-ellipsoid and Levy functions. In most cases, they converge slower than INGO and SABO. With a large sample size, i.e., $N = 100$, the CMA-ES method can maintain a fast converge rate according to Figure 4 (a). The BES method fails to achieve high precision in all test functions. It diverges in ellipsoid, $l_{\frac{1}{2}}$-ellipsoid, Levy functions, and only achieves a low precision in the different power functions. These results demonstrate the superiority of the SABO method in optimizing high-dimensional problems, and verify our theoretical convergence results in Section 4.

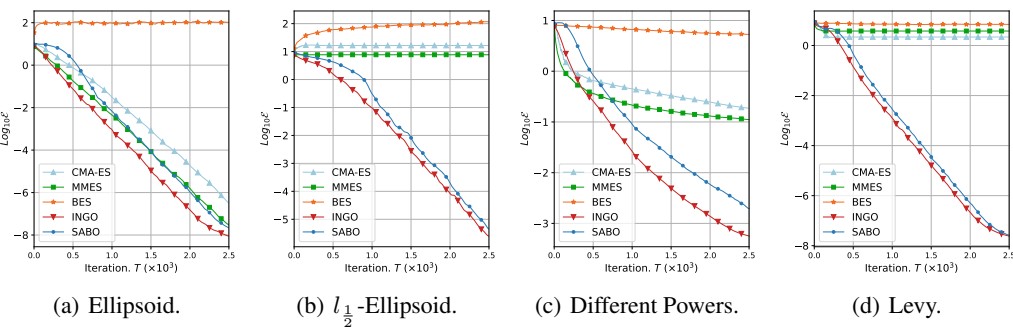

| (a) Ellipsoid. | (b) $l_{\frac{1}{2}}$-Ellipsoid. | (c) Different Powers. | (d) Levy. |

Figure 2: Results on the four test functions with problem dimension $d = 200$ and $N = 50$.

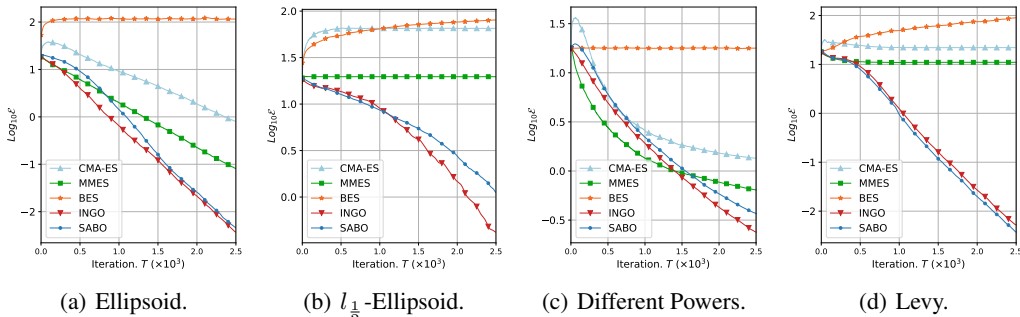

(a) Ellipsoid.  (b) $l_{\frac{1}{2}}$-Ellipsoid.  (c) Different Powers.  (d) Levy.

Figure 3: Results on the four test functions with problem dimension $d = 1000$ and $N = 50$.

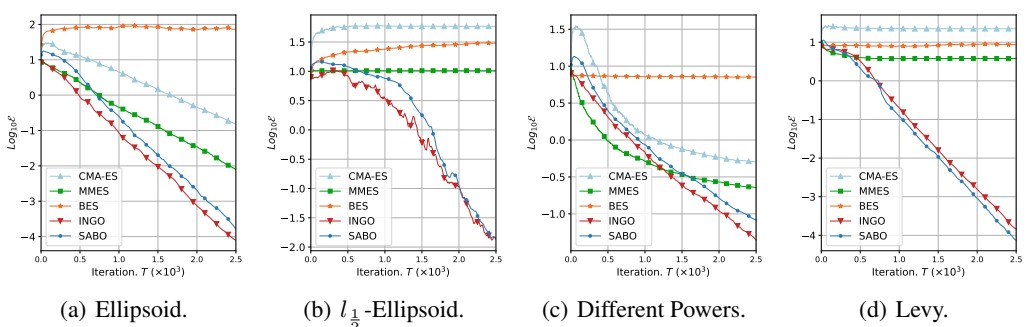

(a) Ellipsoid.  (b) $l_{\frac{1}{2}}$-Ellipsoid.  (c) Different Powers.  (d) Levy.

Figure 4: Results on the four test functions with problem dimension $d = 200$ and $N = 10$.

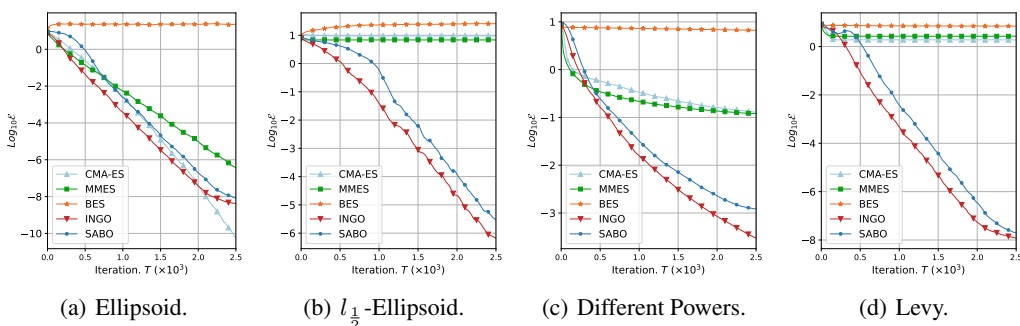

(a) Ellipsoid.  (b) $l_{\frac{1}{2}}$-Ellipsoid.  (c) Different Powers.  (d) Levy.

Figure 5: Results on the four test functions with problem dimension $d = 200$ and $N = 100$.

## H  ADDITIONAL RESULTS ON SYNTHETIC PROBLEMS

We conduct additional experiments to compare the proposed SABO method and GFM method Lin et al. (2022). For GFM, we employ its default hyperparameter setting from Lin et al. (2022). We set its smoothing parameter $\delta = 0.2$ and conduct experiments on the step-size over $\{0.001, 0.0005, 0.0001\}$. The initial point is set the same as SABO.

The results are shown in Figure 6 with problem dimension $d = 200$ and $N = 50$. The GFM method can converge slowly for the Ellipsoid problem. It fails in the $l_{\frac{1}{2}}$-Ellipsoid problem and cannot achieve high precision for Different Powers and Levy problem. This shows that it could be challenging for GFM to optimize non-smooth or high-dimensional test functions without adaptively updating mean and covariance. The proposed SABO method takes advantage of the second-order information, which gains great acceleration for solving these problems.

Table 2: Performance (%) on *SST-2*, *AG's News*, *MRPC*, *RTE*, *SNLI*, and *Yelp P.* datasets. We report the mean and standard deviation over 3 random seeds. The best result across all groups is highlighted in **bold** and the best result in each group is marked with underlined.

| Methods | SST-2 | AG's News | MRPC | RTE | SNLI | Yelp P. |
|---|---|---|---|---|---|---|
| Zero-shot | 79.82 | 76.96 | 67.40 | 51.62 | 38.82 | 89.64 |
| *Dimension $d = 200$* | | | | | | |
| GIBO | $83.53_{\pm0.15}$ | $75.79_{\pm0.08}$ | $79.21_{\pm0.09}$ | $53.07_{\pm0.29}$ | $38.73_{\pm0.09}$ | $89.63_{\pm0.03}$ |
| TurBo | $83.30_{\pm0.19}$ | $79.01_{\pm1.68}$ | $69.59_{\pm8.51}$ | $46.57_{\pm2.30}$ | $40.27_{\pm0.69}$ | $90.16_{\pm0.19}$ |
| **SABO** | $\underline{87.88}_{\pm0.53}$ | $\underline{82.22}_{\pm0.41}$ | $\underline{79.35}_{\pm0.12}$ | $\mathbf{53.67}_{\pm0.17}$ | $\underline{40.72}_{\pm0.15}$ | $\underline{91.50}_{\pm0.13}$ |
| *Dimension $d = 500$* | | | | | | |
| GIBO | $83.49_{\pm0.09}$ | $75.70_{\pm0.05}$ | $79.03_{\pm0.08}$ | $52.95_{\pm0.17}$ | $38.71_{\pm0.16}$ | $89.65_{\pm0.02}$ |
| TurBo | $84.52_{\pm0.65}$ | $80.03_{\pm1.97}$ | $75.30_{\pm2.34}$ | $48.01_{\pm0.59}$ | $38.82_{\pm0.34}$ | $90.20_{\pm0.45}$ |
| **SABO** | $\underline{87.31}_{\pm0.38}$ | $\underline{82.65}_{\pm0.59}$ | $\underline{79.62}_{\pm0.07}$ | $\underline{53.55}_{\pm0.17}$ | $\mathbf{42.29}_{\pm2.48}$ | $\underline{91.83}_{\pm0.16}$ |
| *Dimension $d = 1000$* | | | | | | |
| GIBO | $83.45_{\pm0.11}$ | $75.67_{\pm0.10}$ | $79.15_{\pm0.0}$ | $52.95_{\pm0.17}$ | $38.87_{\pm0.18}$ | $89.65_{\pm0.04}$ |
| TurBo | $85.90_{\pm0.95}$ | $82.36_{\pm0.21}$ | $77.30_{\pm0.86}$ | $50.30_{\pm1.11}$ | $39.87_{\pm1.07}$ | $90.14_{\pm0.20}$ |
| **SABO** | $\mathbf{87.96}_{\pm0.83}$ | $\mathbf{82.77}_{\pm0.41}$ | $\mathbf{79.68}_{\pm0.23}$ | $\underline{53.31}_{\pm0.17}$ | $\underline{40.32}_{\pm0.27}$ | $\mathbf{91.96}_{\pm0.41}$ |

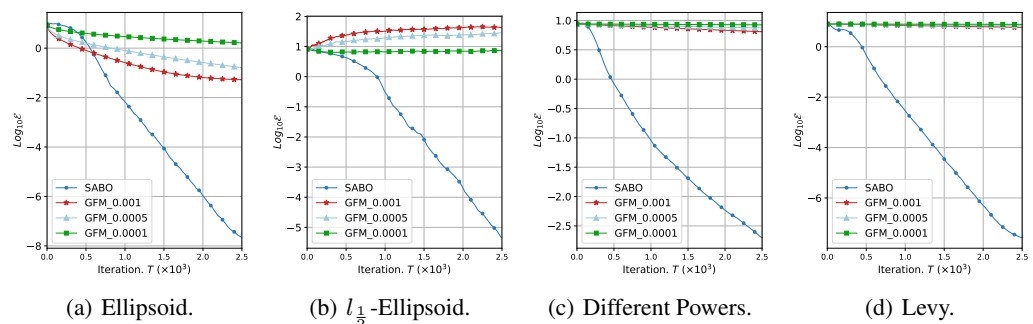

(a) Ellipsoid.  (b) $l_{\frac{1}{2}}$-Ellipsoid.  (c) Different Powers.  (d) Levy.

Figure 6: Results on the four test functions with problem dimension $d = 200$ and $N = 50$.

## H.1 Additional Results on Black-box Prompt Fine-tuning

We conduct an additional experiment on the black-box prompt fine-tuning task to compare the proposed SABO method with high-dimensional BO methods, i.e., TuRBO (Eriksson et al., 2019) and GIBO (Nguyen et al., 2022) methods discussed in Section 5. We employ the default setting of TuRBO-1 from He et al. (2020) for the TuRBO method, and the default setting from Nguyen et al. (2022) for the GIBO method. The results on six benchmark datasets are reported in Table 2. According to the results, the SABO method consistently outperforms these two baselines, highlighting its effectiveness.

## H.2 Synthetic Image Classification Problem

We conduct experiments on a synthetic image classification problem to empirically evaluate the performance of the proposed mini-batch SABO method. Specifically, we apply black-box optimization methods to train a model to classify the images accurately. Moreover, following Foret et al. (2021), we conduct additional experiments on the noisy label setting. Particularly, we train the model on a corrupted version of the *Fashion-MNIST* dataset, where some of its training labels are randomly flipped according to different noise rates, while the testing set is clean. To construct problems with different dimensions, we first adopt UMAP (McInnes et al., 2018) to reduce the dimension of *Fashion-MNIST* to 8 while preserving the class discriminability (Sagawa & Hino, 2022), and then employ a fixed randomly initialized matrix $P \in \mathbb{R}^{8 \times \tilde{d}}$ to project the extracted features to a

Table 3: Performance (%) on *Fashion-MNIST* dataset with noise labels. The best result across all groups is highlighted in **bold** and the best result in each group is marked with underlined.

| Methods | noise=0% | noise=20% | noise=40% | noise=60% | noise=80% |
|---|---|---|---|---|---|
| *Dimension $d = 100$* | | | | | |
| CMA-ES | $76.68\pm_{0.44}$ | $77.25\pm_{1.26}$ | $75.71\pm_{1.15}$ | $70.81\pm_{1.01}$ | $57.86\pm_{3.65}$ |
| MMES | $76.20\pm_{1.45}$ | $74.68\pm_{0.33}$ | $74.46\pm_{0.32}$ | $71.12\pm_{2.24}$ | $65.92\pm_{1.30}$ |
| BES | $76.26\pm_{0.21}$ | $73.60\pm_{0.50}$ | $67.60\pm_{0.72}$ | $55.14\pm_{6.26}$ | $45.65\pm_{4.35}$ |
| INGO | $76.68\pm_{0.90}$ | $75.16\pm_{0.55}$ | $71.26\pm_{0.74}$ | $61.70\pm_{3.89}$ | $46.75\pm_{1.35}$ |
| SABO | $\underline{\mathbf{78.10}}\pm_{0.90}$ | $\underline{77.41}\pm_{0.68}$ | $\underline{\mathbf{77.10}}\pm_{1.69}$ | $\underline{73.41}\pm_{0.33}$ | $\underline{\mathbf{66.45}}\pm_{0.43}$ |
| *Dimension $d = 1000$* | | | | | |
| CMA-ES | $76.75\pm_{0.54}$ | $74.85\pm_{0.06}$ | $73.21\pm_{0.52}$ | $67.32\pm_{0.84}$ | $50.00\pm_{0.98}$ |
| MMES | $75.85\pm_{0.43}$ | $72.39\pm_{3.59}$ | $73.73\pm_{0.50}$ | $72.51\pm_{1.47}$ | $56.60\pm_{2.72}$ |
| BES | $76.18\pm_{0.63}$ | $72.45\pm_{0.44}$ | $66.64\pm_{2.09}$ | $58.74\pm_{1.50}$ | $46.99\pm_{4.30}$ |
| INGO | $74.73\pm_{0.62}$ | $74.87\pm_{1.72}$ | $69.75\pm_{0.73}$ | $63.18\pm_{4.04}$ | $41.80\pm_{1.36}$ |
| SABO | $\underline{76.87}\pm_{1.41}$ | $\underline{\mathbf{77.73}}\pm_{0.85}$ | $\underline{74.97}\pm_{1.12}$ | $\underline{\mathbf{73.81}}\pm_{1.26}$ | $\underline{61.96}\pm_{0.08}$ |

$\widetilde{d}$-dimensional space. After preprocessing, a linear layer is used as a classifier. Therefore, the total number of the trainable parameters is $d = 10 \times \widetilde{d}$.

**Datasets.** *Fashion-MNIST* (Xiao et al., 2017) is a image classifications dataset. It contains $60,000$ training samples and $10,000$ test samples, each representing a $28 \times 28$-pixel grayscale image of fashion items from 10 different classes.

**Implementation Details.** For a fair comparison, we set the same population size, number of batch samples, and the initialization for CMA-ES (Hansen, 2006), INGO (Lyu & Tsang, 2021), BES (Gao & Sener, 2022), and SABO. The population size and the number of batch samples are set to $N = 100$ and $M = 2048$, respectively. The Gaussian distributions are initialized with $\boldsymbol{\mu}_0 = \mathbf{0}$ and $\boldsymbol{\Sigma}_0 = 0.5\boldsymbol{I}$. For BES, INGO and SABO methods, we search the learning rate $\beta$ over $\{0.1, 0.5, 1, 5\}$. Moreover, we perform grid-search on $\rho$ over $\{100, 500, 1000, 5000\}$ for SABO. We employ the default hyperparameter setting from He et al. (2020) for the MMES method. For the BES method, we perform grid-search on the spacing $c$ over $\{0.1, 1, 10\}$. The cross-entropy loss is used as the training objective. All experiments are repeatedly run with three independent seeds and the mean and standard deviation are reported.

**Results.** Table 3 shows experimental results on *Fashion-MNIST* dataset with different $\widetilde{d}$ and different noise rates. We can see that the SABO method consistently outperforms all baselines across different noise rates and dimensions, highlighting its effectiveness. These results show that the proposed SABO method can achieve good robustness performance and demonstrate the effectiveness of the proposed SABO method in improving model generalization performance.

# I  DISCUSSION WITH OFFLINE BLACK-BOX OPTIMIZATION

The typical black-box optimization problem we studied in this work can also be called the online black-box optimization problem. Since we have access to the objective function $F$, the problem can be solved in an **online** iterative manner, where in each iteration the solver proposes new $\boldsymbol{x}$ and queries the objective function for feedback in order to inform better solution proposals at the next iteration.

The offline black-box optimization (Chen et al., 2022; Qi et al., 2022) is different from the research line of standard online black-box optimization. In offline black-box optimization, access to the true objective $F$ is not available. Instead, the offline black-box optimization algorithm is provided access to a static dataset $\mathcal{D} = \{\boldsymbol{x}_i, F(\boldsymbol{x}_i)\}$ of the variable $\boldsymbol{x}_i$ and its corresponding objective value $F(\boldsymbol{x}_i)$. Therefore, the basic settings of offline black-box optimization and online black-box optimization are different.

Moreover, the challenges of offline black-box optimization and online black-box optimization are also different. Offline black-box optimization focuses on producing query candidates by a surrogate model trained with a prior static dataset (Trabucco et al., 2021; Kumar & Levine, 2020). The goal is to produce a good query set based on the fixed model at one time without considering query feedback update, exploration and exploitation balance in the long term. Along this line, Kumar & Levine (2020) proposed a model-based offline optimization by training a conditional generative model that conditions the objective value. In Fannjiang & Listgarten (2020), the authors formulated the problem as a non-zero-sum game and proposed an alternating ascent-descent algorithm for model-based offline optimization. Trabucco et al. (2021) proposed the conservative objective models, which presents a technique similar to adversarial training that avoids overestimation of out-of-distribution inputs. In contrast to offline black-box optimization, standard online black-box optimization needs to balance exploration and exploitation, which focuses on long-term convergence performance. As a result, offline black-box optimization is not suitable for our online prompt fine-tuning tasks.

## J    DISCUSSIONS ABOUT THE SOCIETAL IMPACT AND LIMITATIONS

This work only focuses on black-box optimization in deep learning, so it has no negative societal impact. The main theoretical analysis in this work focuses on convex black-box functions. It is technically challenging to analyze non-convex cases considering both the black-box nature and the sharpness-aware properties, and we leave this as one of our future work. Additionally, the SABO method employs a standard Monte Carlo sampling for gradient approximation. Other sampling techniques might be more efficient, but those are out of the scope of this work. We will study it in the future.

