# OpenReview forum: "Sharpness-Aware Black-Box Optimization"
_ICLR.cc/2025/Conference — ICLR 2025 Poster_

### Official Review · Reviewer_vnYS · 2024-10-29

**Soundness:** 2
**Presentation:** 3
**Contribution:** 2
**Rating:** 5
**Confidence:** 4

**Summary:**

This paper studies Sharpness-Aware Minimization in the context of black-box optimization. It introduces Sharpness-Aware Black-Box Optimization (SABO), which is the first to design a sharpness-aware minimization strategy specifically for black-box optimization. The key contributions include the formulation of the SABO algorithm, convergence results, generalization bounds, and extensive empirical validation through experiments on both synthetic benchmark functions and black-box prompt fine-tuning tasks.

**Strengths:**

- The paper is well-written and easy to follow.
- The proposal of integrating SAM techniques into black-box optimization is interesting. They show clearly how to achieve the update algorithm. The paper provides proofs for the convergence rate and generalization bounds.
- Comprehensive experiments are conducted on synthetic benchmarks and real-world tasks (black-box prompt fine-tuning for large language models). Results show that SABO outperforms other methods.

**Weaknesses:**

- The paper lacks evidence that SABO can find flat minima, even though this is their motivation for improving model generalization.
- The paper lacks a strong motivation for why we need Sharpness-Aware Minimization for black-box optimization, aside from the fact that it may help improve generalization. Could the authors identify the problems that existing black-box optimization methods encounter and demonstrate how SABO addresses these issues?
- Assumption 4.1 is narrow because it includes both L-smoothness and H-Lipschitz conditions; typically, we only assume one of them. For example, the simple function $f(x) = x^2$ satisfies L-smoothness but does not satisfy H-Lipschitz.
- In Theorem 4.3, the assumption is too strong for real applications in neural networks. The result in Lin et al. [2022] is more general.
- Theorem 4.6 is not strong because the assumptions are overly restrictive, which limits the practical applicability of this theorem.
- The empirical convergence results of SABO in synthetic problems do not show much improvement over INGO.
- The empirical results in Table 1 are not very significant, showing only around a 0-1\% improvement, even for datasets with low accuracy such as RTE and SNLI.

Tianyi Lin, Zeyu Zheng, and Michael Jordan. Gradient-free methods for deterministic and stochastic nonsmooth nonconvex optimization. Advances in Neural Information Processing Systems, 35:26160–26175, 2022.

**Questions:**

- Could you provide a table showing how selecting the neighborhood size $\rho$ affects both convergence and performance?
- What specific $\rho$ would you recommend to ensure stability?
- Could the authors include a comparison of computational usage against other methods in Table 1?
- Could the authors provide a comparison of your theoretical and empirical results with the paper Lin et al. [2022]?
- See the Weakness part.

Tianyi Lin, Zeyu Zheng, and Michael Jordan. Gradient-free methods for deterministic and stochastic nonsmooth nonconvex optimization. Advances in Neural Information Processing Systems, 35:26160–26175, 2022.

---

> ### Author Response · Authors · 2024-11-22
> **Reply to Reviewer xCwT (1/3)**
>
> >**Q1**. The paper lacks evidence that SABO can find flat minima, even though this is their motivation for improving model generalization.
>
> **A1**： The main goal of this work is to improve the model generalization performance. The original SAM minimizes the worst-case loss in a local ball in terms of the $l_2$-norm of the parameter. They can find flat minima w.r.t. the parameter space. Our method is inspired by their optimization strategy. The proposed SABO minimizes the worst-case (Gaussian-smooth Expectation) loss in a local ball of parameter's distribution in terms of KL divergence. Intuitively, it can be viewed as adversarial training with an adversary perturbing the distribution of parameters in a local ball w.r.t. KL-divergence. In the general case, the adversary in our SABO can perturb both the mean $\boldsymbol{\mu}$ and the covariance $\boldsymbol{\Sigma}$  of the parameter,  which has a larger degree of freedom to perturb and the adversarial training may lead to a more robust solution. So we can find the flat minima w.r.t. the search distribution space (Lines 69-72), which is different from SAM. It is hard to directly show the flat region w.r.t the distribution space. However, empirical results on both image classification problems and black-box prompt tuning problems have demonstrated that the proposed SABO can improve the model generalization performance.
>
> ---
>
> >**Q2**. The paper lacks a strong motivation for why we need Sharpness-Aware Minimization for black-box optimization, aside from the fact that it may help improve generalization. Could the authors identify the problems that existing black-box optimization methods encounter and demonstrate how SABO addresses these issues?
>
> **A2**： Our paper provides a strong motivation for using SAM in black-box optimization (Lines 36-53). Improving model generalization performance is crucial for all learning tasks. Previous studies have shown that directly optimizing the training loss can lead to a suboptimal model with poor generalization performance [R2]. For standard black-box optimization methods in learning tasks, most of them directly optimize the training loss. Therefore, this motivates us to learn from successful techniques in white-box optimization, such as SAM, which can also be considered a min-max optimization strategy, to design proper algorithms for black-box optimization. Therefore, after applying the stochastic gradient approximation method, we propose a sharpness-aware formulation for the black-box problems.
>
> ---
>
> >**Q3**. Assumption 4.1 is narrow because it includes both L-smoothness and H-Lipschitz conditions; typically, we only assume one of them. For example, the simple function $f(x)=x^2$ satisfies L-smoothness but does not satisfy H-Lipschitz.
>
> **A3**：Thanks for your insightful comments. We realize that the properties of $L$-smoothness and $H$-Lipschitz function $J(\theta)$ are not fully used in our proofs. We can weaken the assumptions by removing the original Assumption 4.1 and only assume that (1) the function $J(\theta)$ satisfies that $\nabla_{\mu}J(\theta)$ is $L$-Lipschitz w.r.t. $\theta$, (2) $||\nabla_{\Sigma=\Sigma_t} J(\theta)||_{F}\le H$ holds for the sequence $\Sigma_t$ generated by SABO. These assumputions is weaker than the original assumputions and we put the updated Assumption 4.1 and Theorem 4.3 in the updated version.
>
> ---
>
> >**Q4**. In Theorem 4.3, the assumption is too strong for real applications in neural networks. The result in Lin et al. [2022] is more general.
>
> **A4**： Lin et al. 2022 [R1] employ the isotropic uniform spherical distribution as the sampling distribution, which follows along the direction of standard zeroth-order optimization analysis.  In contrast,  our method needs to update the covariance dynamically; this introduces an additional challenge to analyzing the convergence rate compared with a fixed variance isotropic distribution. The adaptively updated covariance takes advantage of the second-order information, which gains great acceleration for non-isotropic (e.g., ill-conditioned) problems.
>
> ---
>
> >**Q5**. Theorem 4.6 is not strong because the assumptions are overly restrictive, which limits the practical applicability of this theorem.
>
> **A5**： As our reply to **Q3**, we weaken the current assumptions. Since our approach needs to update the covariance matrix dynamically, this introduces an additional challenge to analyzing the convergence rate compared with a fixed variance distribution and further leads to additional assumptions.
>
> ---

---

> ### Author Response · Authors · 2024-11-22
> **Reply to Reviewer xCwT (2/3)**
>
> >**Q6**. The empirical convergence results of SABO in synthetic problems do not show much improvement over INGO.
>
> **A6**： Although the proposed method focuses on improving model generalization performance in training tasks, verifying its the reliability and convergent properties of the proposed optimization approach is also important. Therefore, we conduct experiments on four challenging synthetic problems to verify our convergent result. Since the proposed SABO method has the same theoretical convergent order as INGO. So, it is reasonable that they have similar convergence results in numerical experiments. In addition, the similar performance of SABO and INGO in the numerical experiment and the better performance of SABO in the training task further illustrate the effectiveness of the proposed method in improving the model generalization performance.
>
> ---
>
> >**Q7**. The empirical results in Table 1 are not very significant, showing only around a 0-1% improvement, even for datasets with low accuracy such as RTE and SNLI.
>
> **A7**： The proposed SABO method consistently achieves better performance on six datasets with three different settings. In contrast, all baseline methods cannot achieve consistently good performance in all datasets. For example, compared with the CMA-ES method, a representative black-box optimization method, the proposed SABO method has 4-5% improvement on the MRPC and SNLI datasets and 0-2% improvement on the other four datasets with $d=200$. These results can demonstrate the effectiveness of our proposed algorithm.
>
> ---
>
> >**Q8**. Could you provide a table showing how selecting the neighborhood size $\rho$ affects both convergence and performance?
>
> **A8**： We conduct sensitivity analysis experiment on $\rho$ and the results on the SST-2 dataset are reported in the following table.  According to the results, we can see that the test accuracy fluctuates between 86\% and 88\% by varying $\rho$, which implies that the performance of the proposed BASO method is not so sensitive to $\rho$.
>
>  | $\rho$ | SST-2 (d=200) |
>  |--------|---------------|
>  | 0.1    | 86.74         |
>  | 0.5    | 86.62         |
>  | 1      | 86.93         |
>  | 5      | 86.81         |
>  | 10     | 86.32         |
>  | 50     | 87.88         |
>  | 100    | 86.55         |
>  | 500    | 86.85         |
>
> ---
>
> >**Q9**. What specific $\rho$ would you recommend to ensure stability?
>
> **A9**： To ensure training stability, we empirically found that a low perturbation performs stably. The proposed SABO is similar to SAM; it can be seen as a min-max optimization or adversarial training strategy. If the perturbation is extremely large, the algorithm may lead to bad results. Meanwhile, if the perturbation is too small, the algorithm's performance will tend towards standard black-box optimization, which directly optimizes the training loss and thus cannot improve the model's generalization. Empirically, we find that we need to perform gird search for the perturbation $\rho$ to achieve good performance, which is similar to other variants of SAM [R3,R4].
>
> ---
>
> >**Q10**. Could the authors include a comparison of computational usage against other methods in Table 1?
>
> **A10**：The analysis of computational complexity of the proposed SABO method can be found in Section 3.2. Empirically, we report the time cost per iteration (min/itr) with different sample sizes $N$ in the SST-2 dataset when $d=500$. Since our method requires calculating the perturbated distribution in each iteration, like SAM and its variants [R4,R5], this approach will cause additional computational costs, which is still of the same order as other methods. We will add this computational usage discussion.
>
>  |        | N = 50 | N = 100 |
>  |--------|--------|---------|
>  | CMA-ES | 0.128  | 0.290   |
>  | INGO   | 0.137  | 0.284   |
>  | BES    | 0.125  | 0.265   |
>  | MMES   | 0.121  | 0.255   |
>  | BASO   | 0.280  | 0.572   |
>
> ---

---

> ### Author Response · Authors · 2024-11-22
> **Reply to Reviewer xCwT (3/3)**
>
> >**Q11**. Could the authors provide a comparison of your theoretical  and empirical   with the paper Lin et al. [2022]?
>
> **A11**： Theoretically, it is hard to compare the theoretical results between SABO and GFM [R1] since the analysis of our work focuses on the convergence properties w.r.t the global optimal of convex functions. In contrast, [R1] focuses on analyzing the convergence properties w.r.t the local stationary solution.
>
> Empirically, we compare SABO and GFM [R1] in both synthetic benchmark problems and black-box prompt fine-tuning tasks.
>
> 1. We compare our SABO  with GFM [R1] on the four synthetic benchmark problems. To show the results, we put the corresponding Figures (Figure 6) in Appendix I in the updated version. For [R1], we employ its default hyperparameter setting. We set its smoothing parameter $\delta=0.2$ and conduct experiments on the step-size over {0.001,0.0005,0.0001}. The popsize and initial point are set as same as SABO. The results are shown in Figure 6 with problem dimension $d=200$ and $N=50$. The GFM method can converge slowly for the Ellipsoid problem. It fails in the $l_{\frac{1}{2}}$-Ellipsoid problem and cannot achieve high precision for Different Powers and Levy problem. This shows that it could be challenging for GFM to optimize non-smooth or high-dimensional test functions without adaptively updating mean and covariance. The proposed SABO method takes advantage of the second-order information, which gains great acceleration for solving these problems.
>
> 2. We conduct additional experiments on the black-box prompt fine-tuning tasks with three datasets. For GFM [R1], we employ its default hyperparameter setting. We set its smoothing parameter $\delta=0.2$ and conduct a gird search to its step-size over {0.001,0.0005,0.0001}. The popsize and initial point are set as same as other baseline methods. The results are shown in the following table. The results show that the proposed SABO method has better performance.
>
> We will complete these experiments and add these comparisons.
>
> | Method | SST2 | MRPC | SNLI |
> | ------ | ---- | ---- | ---- |
> | GFM ($d$=200)    | $83.49_{\pm 0.18}$ | $79.23_{\pm 0.20}$ | $38.68_{\pm0.13}$|
> | SABO ($d$=200)   | $87.88_{\pm 0.53}$ | $79.35_{\pm 0.12}$ | $40.72_{\pm0.15}$|
> | GFM ($d$=500)    | $83.45_{\pm 0.06}$ | $79.21_{\pm 0.09}$ | $38.65_{\pm0.23}$|
> | SABO ($d$=500)   | $87.31_{\pm 0.38}$ | $79.62_{\pm 0.07}$ | $42.29_{\pm2.48}$|
> | GFM  ($d$=1000)   | $83.10_{\pm 0.11}$ | $79.03_{\pm 0.08}$| $38.54_{\pm0.13}$ |
> | SABO ($d$=1000)   | $87.96_{\pm 0.83}$ |$79.68_{\pm 0.23}$ | $40.32_{\pm0.27}$ |
>
> ---
>
> **Ref**:
>
> [R1] Lin et al. Gradient-free methods for deterministic and stochastic nonsmooth nonconvex optimization. Neurips 2022.
>
> [R2] Foret et al. Sharpness-Aware Minimization for Efficiently Improving Generalization. ICLR 2021.
>
> [R3] Jiang et al. An Adaptive Policy to Employ Sharpness-Aware Minimization. ICLR 2023
>
> [R4] Kim et al. Fisher SAM: Information geometry and sharpness aware minimisation. ICML 2022.
>
> [R5] Du et al. Efficient sharpness-aware minimization for improved training of neural networks. ICLR, 2022.

---

> > ### Comment · Reviewer_vnYS · 2024-11-26
> > **Thanks for your response**
> >
> > Thank you for your response. Most of my questions have been addressed, and the newly added experiments on GFM are valuable. I have slightly increased my score to reflect and encourage this improvement.
> >
> > However, I still find the motivation for SABO insufficient. A deeper analysis of the practical challenges in black-box optimization and how SABO addresses these issues would make the work more compelling. Currently, SABO appears to be a variant of SAM applied to black-box optimization, rather than a truly innovative solution for this domain.

---

> > > ### Author Response · Authors · 2024-11-27
> > >
> > > We are glad to see that our responses have addressed most of the reviewer's concerns. We appreciate the reviewer's positive feedback and increasing the score.
> > >
> > > For the follow-up concern, the proposed SABO method focuses on improving generalization performance for training and fine-tuning tasks, and it is not a standard black-box optimization algorithm that particularly designed for highly-complex black-box problems. Our method conducts perturbation on the search distribution space instead of the parameter space, which makes this sharpness-aware training strategy applicable to black-box objectives and further improves the model generalization performance. Moreover, our method offers a valuable alternative perspective in the field of black-box optimization and may serve as a source of inspiration for other researchers in this area.

---

### Official Review · Reviewer_WkhT · 2024-10-31

**Soundness:** 3
**Presentation:** 3
**Contribution:** 3
**Rating:** 6
**Confidence:** 4

**Summary:**

The paper introduces the Sharpness-Aware Black-Box Optimization (SABO) algorithm, enhancing model generalization in black-box optimization tasks by incorporating sharpness-aware minimization principles. Unlike traditional methods that may lead to poor generalization by focusing solely on training loss, SABO optimizes a Gaussian parameter distribution, finding "flat" minima that improve robustness. Through reparameterization and stochastic gradient approximation, SABO adapts Sharpness-Aware Minimization (SAM) for use without gradient information, making it suitable for black-box settings. Theoretical results demonstrate SABO's convergence and generalization bounds, and empirical evaluations on synthetic problems and prompt fine-tuning tasks show that SABO outperforms existing methods in high-dimensional settings, positioning it as a powerful tool for black-box optimization in machine learning.

**Strengths:**

- This paper is novel in that it extends sharpness-aware minimization (SAM) principles, typically used in gradient-based settings, to black-box optimization where gradients are unavailable.
- This paper is supported by rigorous theoretical analysis, including convergence rates and generalization bounds in both full-batch and mini-batch settings.
- The empirical results highlight SABO’s practical value and potential to improve generalization in real-world applications where direct gradient access is unavailable.

**Weaknesses:**

- Due to its reliance on iterative updates to a Gaussian distribution, along with stochastic gradient approximations and KL divergence constraints, this method may incur higher computational costs per iteration compared to some existing black-box optimization methods, especially in high-dimensional settings. A discussion of the time complexity for various black-box optimization algorithms would strengthen the paper.

- This paper introduces additional hyperparameters (e.g., neighborhood size for KL divergence) that require careful tuning to achieve optimal performance. Discussing the sensitivity of these hyperparameters would improve the analysis.

- Although the proposed method is evaluated on prompt fine-tuning tasks, the backbone model used in the experiments may not be sufficiently representative. Evaluating it on a larger model, such as LLaMA-7B, would provide a stronger assessment of its effectiveness.

**Questions:**

Please see weakness

---

> ### Author Response · Authors · 2024-11-22
> **Reply to Reviewer WkhT**
>
> Thank you for your thoughtful review and constructive comments.
>
> >**Q1**. Due to its reliance on iterative updates to a Gaussian distribution, along with stochastic gradient approximations and KL divergence constraints, this method may incur higher computational costs per iteration compared to some existing black-box optimization methods, especially in high-dimensional settings. A discussion of the time complexity for various black-box optimization algorithms would strengthen the paper.
>
> **A1**： The analysis of computational complexity can be found in Line 248. For the proposed BASO method with a diagonal covariance matrix, it requires $\mathcal{O}(Nd)$ time to calculate the perturbation distribution in each iteration. Then it takes $\mathcal{O}(Nd)$ time to update the distribution $p_{\theta_t}$. Therefore, the computational cost of the BASO method in each iteration is of order $\mathcal{O}(Nd)$. For the INGO, BES and MMES methods, their computational cost is also of order $\mathcal{O}(Nd)$. Therefore, the proposed SABO method does not increase the order of the computational cost compared with various black-box optimization algorithms. The increased computational cost of the proposed method mainly comes from calculating the perturbation perturbations, which is similar to the SAM.
>
> ---
>
> >**Q2**. This paper introduces additional hyperparameters (e.g., neighborhood size for KL divergence) that require careful tuning to achieve optimal performance. Discussing the sensitivity of these hyperparameters would improve the analysis.
>
> **A2**：We conduct sensitivity analysis experiment on $\rho$ and the results on the SST-2 dataset are reported in the following table.
> According to the results, we can see that the test accuracy fluctuates between 86\% and 88\% by varying $\rho$, which implies that the performance of the proposed BASO method is not so sensitive to $\rho$.
>
>  | $\rho$ | SST-2 (d=200) |
>  |--------|---------------|
>  | 0.1    | 86.74         |
>  | 0.5    | 86.62         |
>  | 1      | 86.93         |
>  | 5      | 86.81         |
>  | 10     | 86.32         |
>  | 50     | 87.88         |
>  | 100    | 86.55         |
>  | 500    | 86.85         |
>
> ---
>
> >**Q3**. Although the proposed method is evaluated on prompt fine-tuning tasks, the backbone model used in the experiments may not be sufficiently representative. Evaluating it on a larger model, such as LLaMA-7B, would provide a stronger assessment of its effectiveness.
>
> **A3**： We appreciate the suggestion and agree that evaluating the proposed method on larger models, such as LLaMA-7B, could provide a more comprehensive assessment of its effectiveness. In our current experiments, we selected the backbone model based on computational resource constraints and the need to focus on validating the core ideas of our approach, i.e., improving the model generalization performance. Despite this, the model we used has been shown to be effective and relevant for black-box prompt fine-tuning tasks within similar research contexts. Testing on larger models like LLaMA-7B is indeed a valuable direction for future work, as it could help verify the scalability and robustness of our method in more demanding settings. We plan to explore this in follow-up studies.

---

### Official Review · Reviewer_xCwT · 2024-11-03

**Soundness:** 3
**Presentation:** 4
**Contribution:** 3
**Rating:** 10
**Confidence:** 2

**Summary:**

This paper combines ideas from sharpness aware minimization (SAM) in machine learning with black-box optimization problems to propose a new algorithm called Sharpness-Aware Black-box Optimization (SABO). The goal of SABO is to improve the generalization performance of models by finding flat minima in the loss landscape, which are believed to lead to better generalization. The authors provide theoretical analysis of SABO's convergence rate and generalization bound, and they demonstrate its effectiveness through experiments on synthetic problems and black-box prompt fine-tuning tasks.

**Strengths:**

This is an interesting paper that combines ideas from sharpness aware minimization (SAM) with black box optimization. SAM has already proven to be a very useful technique to improve generalization in training ML models and the authors present an intuitive motivation for applying these ideas to BBO. SABO is a creative combination of SAM techniques and black box optimization methods.

This paper has a strong theoretical analysis of SABO, including convergence rate and generalization error bounds. These theoretical results are then empirically validated on synthetic and real-worl black-box prompt fine-tuning models.

The paper is very well-written and logically structured. The authors do a good job motivating their approach and everything is presented in a clear understandable manner.

**Weaknesses:**

Overall, I see no major weaknesses for this paper and some of these points raised here are covered in the questions below which might clear up these apparent weaknesses. For example, in eqn (8) there is a dependence on the parameter $\rho$. The performance of SABO would be sensitive to the neighborhood size $\rho$ in (8) and tuning this parameter would be critical for successful application of this technique. It would be nice to see methods which can automatically adapt this parameter in a useful way to reduce dependence on manual tuning.

The authors do not mention in detail the increased computational cost of the sharpness-aware updates in SABO. One way to decrease computational cost, the authors assume the covariance matrix $\Sigma$ is diagonal. This seems like a fairly naive assumption and one I would assume is too simplistic in general. To address this problem in SAM for traditional ML settings, variants of SAM have been introduced. Could similar techniques be applied here?

The authors focus on Gaussian distributions for the search distribution. Is that a fair simplification? How effective or sensitive is SABO with other types of distributions that are likely to occur for parameters; e.g. heavy-tail distributions, or other posteriors that are more informative.

**Questions:**

- in the setup on line 177, you claim that for computational consideration you can assume the search distribution to be Gaussian. Is that a fair simplification? How sensitive is SABO to this choice of distribution and would it make more sense to sample the local parameter space in some way that captures the nature of the posterior rather than with a naive Gaussian?
 - related to above (line 180), is it fair to assume that the covariance matrix is diagonal. This seems optimistic and is essentially assuming no correlation between parameters, correct? And then your perturbations $\delta$ also have diagonal covariance. I understand that, as you point out in line 195, removing this assumption would make the problem computationally substantially more difficult.  Theoretically, how much of a sacrifice does this assumption make? and are there any other means you can use to not lose this amount of information?

---

> ### Author Response · Authors · 2024-11-22
> **Reply to Reviewer xCwT (1/2)**
>
> Thank you for your thoughtful review and constructive comments.
>
> >**Q1**. Overall, I see no major weaknesses for this paper and some of these points raised here are covered in the questions below which might clear up these apparent weaknesses. For example, in eqn (8) there is a dependence on the parameter $\rho$. The performance of SABO would be sensitive to the neighborhood size $\rho$ in (8) and tuning this parameter would be critical for successful application of this technique. It would be nice to see methods which can automatically adapt this parameter in a useful way to reduce dependence on manual tuning.
>
> **A1**： For the SAM-based methods, the setting of the hyperparameter $\rho$ is important. Most SAM-based work [R1,R2] uses the grid-search method to determine $\rho$, and our work also follows their approach. To automatically adapt this hyperparameter, we could apply a learn-to-learn strategy, a.k.a meta-learning, and then, we can evaluate the performance of $\rho$ in a validation dataset in each iteration and adapt it accordingly. However, this will lead to a bi-level black-box optimization problem, and we need specially designed algorithms to solve it. Nevertheless, this could be an interesting extension direction of our work, and we will explore it in the future.
>
> ---
>
> >**Q2**. The authors do not mention in detail the increased computational cost of the sharpness-aware updates in SABO. One way to decrease computational cost, the authors assume the covariance matrix $\Sigma$ is diagonal. This seems like a fairly naive assumption and one I would assume is too simplistic in general. To address this problem in SAM for traditional ML settings, variants of SAM have been introduced. Could similar techniques be applied here?
>
> **A2**： We employ the diagonal covariance update scheme to approximate the full covariance matrix, because we need to calculate the inversion of the covariance matrix to obtain the stochastic gradients (Line 192 in our paper) and the computational cost of calculating the inversion of a non-diagonal $d$-dimensional matrix is very high (i.e. $\mathcal{O}(d^3)$). Therefore, due to the consideration of computational cost, we assume that the covariance matrix $\Sigma$ is a diagonal matrix. Then, for the proposed method, the computational cost of calculating the perturbated distribution is of order $\mathcal{O}(Nd)$, which directly reduces the order of the computational cost and is much more affordable.
>
> Compared with standard black-box optimization methods, the proposed SABO increases the computational cost as we need to calculate the perturbation, which is similar to the standard SAM. To reduce the computational cost, some SAM-based methods like [R2] use the gradient norm to determine whether to do the perturbation in each iteration. We can employ a similar strategy in the black-box optimization case to avoid calculating the perturbation in each iteration. We will regard it as a future work.
>
>
> ---
>
> >**Q3**. The authors focus on Gaussian distributions for the search distribution. Is that a fair simplification? How effective or sensitive is SABO with other types of distributions that are likely to occur for parameters; e.g. heavy-tail distributions, or other posteriors that are more informative.
>
> **A3**： Using Gaussian distributions for the search distribution is simple and effective. This approach has been wildly used in black-box optimization [R3,R4]. By using the Gaussian distribution, the adaptively updated covariance can take advantage of the second-order information, which gains great acceleration for non-isotropic (e.g., ill-conditioned) problems (as we shown in Figure 1).
>
> One advantage of using Gaussian distribution in our work is that we can obtain a close form update formulation for the search distribution by solving the problem (15). If we use heavy-tail distributions or other posteriors, we may not obatin a close-form update formulation.
>
> ---

---

> ### Author Response · Authors · 2024-11-22
> **Reply to Reviewer xCwT (2/2)**
>
> >**Q4**. in the setup on line 177, you claim that for computational consideration you can assume the search distribution to be Gaussian. Is that a fair simplification? How sensitive is SABO to this choice of distribution and would it make more sense to sample the local parameter space in some way that captures the nature of the posterior rather than with a naive Gaussian?
>
> **A4**： As our reply to Q3, using Gaussian distributions for the search distribution is simple and effective and can lead to a close-form update formulation (i.e., Eq. (16)). Therefore, it is a reasonable simplification. Other distributions may not have this property, so it is hard to adaptively update those search distributions.
>
> ---
>
> >**Q5**. related to above (line 180), is it fair to assume that the covariance matrix is diagonal. This seems optimistic and is essentially assuming no correlation between parameters, correct? And then your perturbations $\delta$ also have diagonal covariance. I understand that, as you point out in line 195, removing this assumption would make the problem computationally substantially more difficult. Theoretically, how much of a sacrifice does this assumption make? and are there any other means you can use to not lose this amount of information?
>
> **A5**： We employ the diagonal covariance update scheme to approximate the full covariance matrix. This is a reasonable choice to **balance** the computational cost and optimization properties. On the one hand, compared with the full-matrix update, we lose some information, but we reduce the computational cost from $\mathcal{O}(Nd^3)$ to $\mathcal{O}(Nd)$. This is a huge decrease and makes our algorithm applicable to high-dimensional problems. On the other hand, compared with the ES [R5] and BES [R6] methods, which use a fixed standard Gaussian distribution with an identity covariance matrix, our method can take advantage of the second-order information by adaptively updating the covariance matrix. So, our method can gain great acceleration for non-isotropic (e.g., ill-conditioned) problems.
>
> It is hard to theoretically explain how much information is lost in the whole optimization process. However, exploring how to reduce information loss remains important. Thank you for your valuable comments.
>
> ---
>
> **Ref**:
>
> [R1] Jiang et al. An Adaptive Policy to Employ Sharpness-Aware Minimization. ICLR 2023
>
> [R2] Kim et al. Fisher SAM: Information geometry and sharpness aware minimisation. ICML 2022.
>
> [R3] Wierstra et al. Natural evolution strategies. JMLR. 2014.
>
> [R4] Lyu et al. Black-box optimizer with stochastic implicit natural gradient. ECML-PKDD, 2021
>
> [R5] Salimans et al. Evolution strategies as a scalable alternative to reinforcement learning. arXiv.
>
> [R6] Gao et al. Generalizing gaussian smoothing for random search, ICML, 2022.

---

> > ### Comment · Reviewer_xCwT · 2024-11-27
> >
> > Thanks to the authors for the additional detail and answering my questions.

---

> > > ### Author Response · Authors · 2024-11-28
> > >
> > > Thanks for the reviewer's feedback. We appreciate your continual and positive support, as well as your high recognition of our contributions.

---

### Official Review · Reviewer_SM7t · 2024-11-04

**Soundness:** 2
**Presentation:** 3
**Contribution:** 2
**Rating:** 6
**Confidence:** 3

**Summary:**

The authors strive to introduce SAM to the black-box optimization algorithm, and propose a new algorithm called Sharpness-Aware Black-box Optimization (SABO). SABO reparameterizes the objective function using its expectation over a Gaussian distribution and iteratively updates the parameterized distribution based on approximated stochastic gradients of the maximum objective value within a small neighborhood. The authors provide theoretical guarantees for SABO's convergence rate and generalization bounds. Also, they give empirical experiments to validate its effectiveness in improving model generalization performance on black-box prompt fine-tuning tasks and synthetic numerical problems.

**Strengths:**

1. The paper is clearly written, well organized and easy to follow.
2. I think the whole topic is intriguing and worthy to probe, and the authors give some interesting insights.
3. It is valuable to find that the proposed method could be effective in prompt tuning.

**Weaknesses:**

1. My primary concern is the motivation regarding the introduction of SAM in black-box optimization. In other words, can we safely use the proposed method in black-box optimization? The implicit premise of SAM relies on the smoothness of the mapping function. However, black-box optimization may deal with highly complex, non-smooth, or potentially noisy objective landscapes, scenarios much more complicated than optimizing neural networks. In my opinion, it is crucial for the authors to clarify the scope of which minimizing the maximization of the neighborhood region is appropriate for black-box optimization.

2. I noticed that in the algorithm, the authors use two different samples to separately calculate the ascent and descent gradients. Could the authors give some explanations here given that we always keep the same batch in SAM?

3. The essence of SAM involves two main steps: first, a maximization step using gradient ascent based on the current parameters with a step size of rho/||g||; and second, a minimization step using gradient descent based on the maximized parameters with a step size equal to the learning rate. What are the results of applying this procedure in the current black-box optimization?

4. The convergence analysis mostly relies on the assumption of convexity of F(x), which I think could be less useful.

5. Some minor mistakes: (1) Line 121, SAM is proposed by [Foretet al., 2021] not [Kwon et al., 2021].
(2) Eq 6. nabla_F(x + epsilon_t) should be rewrite as nabla_F(x)|x = epsilon_t

**Questions:**

See Weakness

**Details Of Ethics Concerns:**

I have not found any discussions about the limitations and potential negative societal impact. But in my opinion, this may not be a problem, since the work only focuses on the optimization in deep learning. Still, it is highly encouraged to add corresponding discussions.

---

> ### Author Response · Authors · 2024-11-22
> **Reply to Reviewer SM7t (1/2)**
>
> Thank you for your thoughtful review and constructive comments.
>
> >**Q1**. My primary concern is the motivation regarding the introduction of SAM in black-box optimization. In other words, can we safely use the proposed method in black-box optimization? The implicit premise of SAM relies on the smoothness of the mapping function. However, black-box optimization may deal with highly complex, non-smooth, or potentially noisy objective landscapes, scenarios much more complicated than optimizing neural networks. In my opinion, it is crucial for the authors to clarify the scope of which minimizing the maximization of the neighborhood region is appropriate for black-box optimization.
>
> **A1**：
>
> 1. Our work focuses on improving the model generalization for training problems while we cannot use backpropagation to compute the gradients. So, we use the SAM strategy to design optimization algorithm as SAM has been proved to benefit the model generalization performance. This problem is essential in many settings, such as Languaged-Model-as-a-Service (LMaaS). The proposed algorithm is not specially designed to maintain good theoretical properties in solving highly complex, non-smooth black-box optimization problems.
>
> 2. Our theoretical result clarifies the scope of which SABO is appropriate for black-box optimization. We only assume the $J(\theta)$ is smooth and do not assume that $F(x)$ is smooth. Therefore, the proposed method can be used in solving non-smooth black-box optimization problems.
>
> 3. Empirically, our results also show that the proposed method can be used in solving non-smooth black-box optimization problems. The $l_{0.5}$-Ellipsoid function is a non-smooth function and our method can still converge on it.
>
> Therefore, we can safely use the proposed method for black-box optimization. Particularly. The proposed black-box optimization algorithm can improve the model generalization for training problems.
>
> ---
>
> >**Q2**. I noticed that in the algorithm, the authors use two different samples to separately calculate the ascent and descent gradients. Could the authors give some explanations here given that we always keep the same batch in SAM?
>
> **A2**：In algorithm 1, the model parameter samples $x_i$ are sampled from the optimized distribution of the model parameters, and they are used to approximate the gradient. Since the model parameter distribution changes after gradient ascent, we need to resample these points according to the new parameter distribution. In SAM, they directly calculate the gradient w.r.t. the model parameter by backpropagation, so they do not need these sample processes. In algorithm 2, the resampling process for the training data is not necessary. Whether using the same training data batch to calculate the ascent and descent gradients in one iteration will not affect our theoretical result.
>
> ---
>
> >**Q3**. The essence of SAM involves two main steps: first, a maximization step using gradient ascent based on the current parameters with a step size of rho/\|g\|; and second, a minimization step using gradient descent based on the maximized parameters with a step size equal to the learning rate. What are the results of applying this procedure in the current black-box optimization?
>
> **A3**：Indeed, SAM is a general optimization approach in optimization. However, SAM requires the gradient w.r.t. the optimized variable $x$ to determine the perturbation. The CMA-ES, INGO, and proposed SABO methods reparameterize the original objective as
>
>  $J(\theta) = \mathbb{E}_{p({\theta})}[F(x)]$.
>
> The CMA-ES method is not a stochastic gradient-based method so we cannot directly apply SAM. The INGO and the proposed SABO method uses the approximated gradient to update the distribution $p_{\theta}$. However, those approximated gradients are not w.r.t the optimized variable $x$ and INGO and SABO updates the inverse of the covariance matrix. Therefore, we cannot directly combine SAM with black-box optimization methods like SABO and INGO.
>
> Following the suggestion of applying the procedure of SAM with some black-box optimization methods, we combine SAM with BES method. BES a first-order black-box optimization method that directly approximates the gradient w.r.t. $x$. Because of the accessibility of the gradient w.r.t. $x$, we can directly combine SAM with BES. The results on the SST-2 and AG’s News datasets with $d=200$ are reported in the following table. Though applying SAM improves the performance of the BES method, the proposed BASO method outperforms those two baselines.
>
>  |                 | SST-2                           | AG’s News                       |
>  |-----------------|---------------------------------|---------------------------------|
>  | BES (d=200)     | $83.52 _{\pm 0.11}$             | $75.44 _{\pm 0.31}$             |
>  | BES+SAM (d=200) | $83.83_{\pm0.09}$               | $76.00 _{\pm 0.12}$             |
>  | BASO (d=200)    | $\underline{87.88} _{\pm 0.53}$ | $\underline{82.22} _{\pm 0.41}$ |
>
> ---

---

> ### Author Response · Authors · 2024-11-22
> **Reply to Reviewer SM7t (2/2)**
>
> >**Q4**. The convergence analysis mostly relies on the assumption of convexity of $F(x)$, which I think could be less useful.
>
> **A4**：In the area of Evolution Strategies and (Gaussian-smooth) gradient-approximation-based black-box optimization, the convex assumption is very common [R1, R2, R3, R4] for theoretical analysis of the second-order method (covariance matrix adaptive method).
>
> For example, Beyer (2014) [R1] analyzed the convergence of Natural Evolution Strategies and [R2] analyzed with infinitesimal step size using an ordinary differential equation under the assumption that the objective function is quadratic and strongly convex. In [R3], the authors analyzed the converge rate of the covariance adaptive method under the convex assumption. In the theoretical analysis of [R4], the authors employed the strongly convex assumption and additional Lipschitz continuous assumption of the Hessian matrix.
>
> In the area of Bayesian optimization, the theoretical analysis does not require the convex assumption. However, it requires the assumption of the smoothness of the function in the RKHS [R5, R6, R7], and usually, the resulting converge rate is slower than the one in the (Gaussian-smooth) gradient-approximation-based method with the convex assumption.
>
> ---
>
> >**Q5**. Some minor mistakes: (1) Line 121, SAM is proposed by [Foretet al., 2021] not [Kwon et al., 2021]. (2) Eq 6. nabla_F(x + epsilon_t) should be rewrite as nabla_F(x)|x = epsilon_t
>
> **A5**：Thanks for your insightful comments. We've corrected these mistakes in the updated version.
>
> ---
>
> >**Q6**. I have not found any discussions about the limitations and potential negative societal impact. But in my opinion, this may not be a problem, since the work only focuses on the optimization in deep learning. Still, it is highly encouraged to add corresponding discussions.
>
> **A6**：Thanks for your suggestion. We write a new paragraph to discuss the limitations and potential negative societal impact. We put the following paragraph at Appendix H in the updated version.
>
> "This work only focuses on black-box optimization in deep learning, so it has no negative societal impact. The main theoretical analysis in this work focuses on convex black-box functions. It is technically challenging to analyze non-convex cases considering both the black-box nature and the sharpness-aware properties, and we leave this as one of our future work. Additionally, the SABO method employs a standard Monte Carlo sampling for gradient approximation. Other sampling techniques might be more efficient, but those are out of the scope of this work. We will study it in the future."
>
> ---
>
> **Ref**:
>
> [R1] H. Beyer. Convergence analysis of evolutionary algorithms that are based on the paradigm of information geometry. Evolutionary Computation, 2014.
>
> [R2] Wierstra et al. Natural evolution strategies. JMLR. 2014.
>
> [R3] Lyu et al. Black-box optimizer with implicit natural gradient. ECML-PKDD 2021.
>
> [R4] Haishan Ye. Mirror natural evolution strategies. 2023.
>
> [R5] Srinivas et al. Gaussian process optimization in the bandit setting: no regret and experimental design. ICML 2010.
>
> [R6] Desautels et al. Parallelizing exploration-exploitation tradeoffs in Gaussian process bandit optimization. JMLR 2014.
>
> [R7] Berkenkamp et al. No-regret bayesian optimization with unknown hyperparameters. JMLR 2019.

---

> > ### Comment · Reviewer_SM7t · 2024-11-26
> > **Thanks for your kind response.**
> >
> > Thank you for your response. The rebuttal has addressed my concerns to some extent, and I am considering raising my score to 6. However, for this paper,
> >
> > 1. I am uncertain about the scope of applicability for this method in black-box optimization, especially given the diverse scenarios typically encountered in this field.
> >
> > 2. Most convergence analyses in sharpness-aware training are based on the Lipschitz assumption. Particularly in the context of black-box optimization, relying on an analysis under the convexity assumption may be not quite appropriate, so will limit its practical relevance.

---

> ### Author Response · Authors · 2024-11-26
>
> Thanks for the reviewer's feedback and increasing the score. For the follow-up concerns, we provide additional clarification below:
>
> 1. Our method focuses on improving generalization performance for training neural network with black-box objectives, and it is not particularly designed for highly-complex black-box problems. Whether it can solve high-complex black-box problems needs further investigation.  Nevertheless, our method provides a good alternative perspective for the black-box optimization area, and our approach may be a good inspiration for other researchers in this area.
>
> 2. Standard sharpness-aware training cannot be directly used in black-box optimization. To handle black-box objectives, we need to employ a sampling distribution. Thus, we propose SABO, which conducts sharpness-aware training on the search distribution space instead of the parameter space. This brings a challenge for convergence analysis compared with standard SAM. In addition. Our method updates the covariance dynamically, which can take advantage of the second-order information. This introduces an additional challenge for analyzing the convergence rate compared with a fixed variance isotropic distribution. In this work, we provide a convergence rate for the convex case as the first step for the theoretical analysis of this challenging problem. We leave the theoretical analysis of the non-convex case as one of our future explorations.

---

### Author Response · Authors · 2024-11-22
**General Response**

Dear Reviewers,


We thank all the reviewers for their constructive and valuable comments.

We have answered each reviewer's questions separately. And we hope that our response addressed the reviewers' concerns. We have revised the paper to address the comments raised by reviewers, where the revision is highlighted in blue.

Please check the updated version. Please let us know if there is any further concern or question.

Best,

The authors.

---

### Meta-Review · Area_Chair_bwxo · 2024-12-21

**Metareview:**

This paper extends the idea of Sharpness-Aware Minimization (SAM) to black-box optimization settings, where an objective function is optimized using only function queries. Extending SAM to this setting is non-trivial because the core idea of SAM involves computing the worst-case loss perturbation via a gradient step, but in black-box optimization, only function values can be queried, not gradients. The key idea behind the proposed approach is to consider a probability distribution over the queries, such as a normal distribution centered at each query point, and then work with the expected function value instead of the actual queried value. By differentiating the expected function, one can obtain an approximation of the objective function's gradient using only the available query points. The paper also presents a convergence analysis for the convex case, as well as empirical results on black-box prompt fine-tuning tasks. Given the prevalence of black-box optimization in practical settings like reinforcement learning and prompt fine-tuning, the proposed idea is practically relevant and potentially impactful.

Out of four reviews, three reviewers rated the paper on the accept side (including one strong accept), and one reviewer rated the paper as marginally below the acceptance threshold. The reviewers raised some concerns about the limitations of the theoretical results (specially the convergence proof relying on convexity assumption), as well as some clarifying questions about the connection between the proposed method and SAM and the computational cost of the proposed method. The authors responded to these issues, and as a result, some reviewers increased their initial scores.

In concordance with the reviewers, I believe the paper provides solid and potentially impactful contributions to the field, and thus I recommend acceptance.

**Additional Comments On Reviewer Discussion:**

Out of four reviews, three reviewers rated the paper on the accept side (including one strong accept), and one reviewer rated the paper as marginally below the acceptance threshold. The reviewers raised some concerns about the limitations of the theoretical results (specially the convergence proof relying on convexity assumption), as well as some clarifying questions about the connection between the proposed method and SAM and the computational cost of the proposed method. The authors responded to these issues, and as a result, some reviewers increased their initial scores.

---

### Decision · Program_Chairs · 2025-01-22

Accept (Poster)